# Network dynamics underlying OFF responses in the auditory cortex

**Giulio Bondanelli[1,2], Thomas Deneux[3], Brice Bathellier[3,4], Srdjan Ostojic[1]***

[1]Laboratoire de Neurosciences Cognitives et Computationelles, Département d'études cognitives, ENS, PSL University, INSERM, Paris, France; [2]Neural Computation Laboratory, Center for Human Technologies, Istituto Italiano di Tecnologia (IIT), Genoa, Italy; [3]Départment de Neurosciences Intégratives et Computationelles (ICN), Institut des Neurosciences Paris-Saclay (NeuroPSI), UMR 9197 CNRS, Université Paris Sud, Gif-sur-Yvette, France; [4]Institut Pasteur, INSERM, Institut de l'Audition, Paris, France

**Abstract** Across sensory systems, complex spatio-temporal patterns of neural activity arise following the onset (ON) and offset (OFF) of stimuli. While ON responses have been widely studied, the mechanisms generating OFF responses in cortical areas have so far not been fully elucidated. We examine here the hypothesis that OFF responses are single-cell signatures of recurrent interactions at the network level. To test this hypothesis, we performed population analyses of two-photon calcium recordings in the auditory cortex of awake mice listening to auditory stimuli, and compared them to linear single-cell and network models. While the single-cell model explained some prominent features of the data, it could not capture the structure across stimuli and trials. In contrast, the network model accounted for the low-dimensional organization of population responses and their global structure across stimuli, where distinct stimuli activated mostly orthogonal dimensions in the neural state-space.

**\*For correspondence:**
srdjan.ostojic@ens.fr

## Introduction

Neural responses within the sensory cortices are inherently transient. In the auditory cortex (AC) even the simplest stimulus, for instance a pure tone, evokes neural responses that strongly vary in time following the onset and offset of the stimulus. A number of past studies have reported a prevalence of ON- compared to OFF-responsive neurons in different auditory areas (*Phillips et al., 2002*; *Luo et al., 2008*; *Fu et al., 2010*; *Pollak and Bodenhamer, 1981*). As a result, the transient onset component has been long considered the dominant feature of auditory responses and has been extensively studied across the auditory pathway (*Liu et al., 2019b*; *Kuwada and Batra, 1999*; *Grothe et al., 1992*; *Guo and Burkard, 2002*; *Yu et al., 2004*; *Heil, 1997a*; *Heil, 1997b*), with respect to its neurophysiological basis and perceptual meaning (*Phillips et al., 2002*). In parallel, due to less evidence of OFF-responsive neurons in anaesthetized animals, OFF cortical responses have received comparably less attention. Yet, OFF responses have been observed in awake animals throughout the auditory pathway, and in the mouse AC they arise in 30–70% of the responsive neurons (*Scholl et al., 2010*; *Keller et al., 2018*; *Joachimsthaler et al., 2014*; *Liu et al., 2019a*; *Sollini et al., 2018*).

While the generation of ON responses has been attributed to neural mechanisms based on short-term adaptation, most likely inherited from the auditory nerve fibers (*Phillips et al., 2002*; *Smith and Brachman, 1980*; *Smith and Brachman, 1982*), the mechanisms that generate OFF responses are more diverse and seem to be area-specific (*Xu et al., 2014*; *Kopp-Scheinpflug et al., 2018*). In subcortical regions, neurons in the dorsal cochlear nucleus and in the superior paraolivary nucleus of the brainstem nuclei may generate OFF responses by post-inhibitory rebound, a synaptic

mechanism in which a neuron emits one or more spikes following the cessation of a prolonged hyperpolarizing current (*Suga, 1964*; *Hancock and Voigt, 1999*; *Kopp-Scheinpflug et al., 2011*). In the midbrain inferior colliculus and in the medial geniculate body of the thalamus, OFF responses appear to be mediated by excitatory inputs from upstream areas and potentially boosted by a post-inhibitory facilitation mechanism (*Kasai et al., 2012*; *Vater et al., 1992*; *Yu et al., 2004*; *He, 2003*). Unlike in subcortical areas, OFF responses in AC do not appear to be driven by hyperpolarizing inputs during the presentation of the stimulus, since synaptic inhibition has been found to be only transient with respect to the stimulus duration (*Qin et al., 2007*; *Scholl et al., 2010*). The precise cellular or network mechanisms underlying transient OFF responses in cortical areas therefore remain to be fully elucidated.

Previous studies investigating the transient responses in the auditory system mostly adopted a single-neuron perspective (*Henry, 1985*; *Scholl et al., 2010*; *Qin et al., 2007*; *He, 2002*; *Wang et al., 2005*; *Wang, 2007*). However, in recent years, population approaches to neural data have proven valuable for understanding the role of transients dynamics in various cortical areas (*Buonomano and Maass, 2009*; *Remington et al., 2018*; *Saxena and Cunningham, 2019*). Work in the olfactory system has shown that ON and OFF responses encode the stimulus identity in the dynamical patterns of activity across the neural population (*Mazor and Laurent, 2005*; *Stopfer et al., 2003*; *Broome et al., 2006*; *Friedrich and Laurent, 2001*; *Saha et al., 2017*). In motor and premotor cortex, transient responses during movement execution form complex population trajectories (*Churchland and Shenoy, 2007*; *Churchland et al., 2010*) that have been hypothesized to be generated by a mechanism based on recurrent network dynamics (*Shenoy et al., 2011*; *Hennequin et al., 2014*; *Sussillo et al., 2015*; *Stroud et al., 2018*). In the AC, previous works have suggested a central role of the neural dynamics across large populations for the coding of different auditory features (*Deneux et al., 2016*; *Saha et al., 2017*; *Lim et al., 2016*), yet how these dynamics are generated remains an open question.

Leveraging the observation that the AC constitutes a network of neurons connected in a recurrent fashion (*Linden and Schreiner, 2003*; *Oswald and Reyes, 2008*; *Oswald et al., 2009*; *Barbour and Callaway, 2008*; *Bizley et al., 2015*), in this study we test the hypothesis that transient OFF responses are generated by a recurrent network mechanism broadly analogous to the motor cortex (*Churchland et al., 2006*; *Hennequin et al., 2014*). We first analyzed OFF responses evoked by multiple auditory stimuli in large neural populations recorded using calcium imaging in the mouse AC (*Deneux et al., 2016*). These analyses identified three prominent features of the auditory cortical data: (i) OFF responses correspond to transiently amplified trajectories of population activity; (ii) for each stimulus, the corresponding trajectory explores a low-dimensional subspace; and (iii) responses to different stimuli lie mostly in orthogonal subspaces. We then determined to what extent these features can be accounted for by a linear single-cell or network model. We show that the single-cell model can reproduce the first two features of population dynamics in response to individual stimuli, but cannot capture the structure across stimuli and single trials. In contrast, the network model accounts for all three features. Identifying the mechanisms responsible for these features led to additional predictions that we verified on the data.

## Results

### ON/OFF responses in AC reflect transiently amplified population dynamics

We analyzed the population responses of 2343 cells from the AC of three awake mice recorded using calcium imaging techniques (data from *Deneux et al., 2016*). The neurons were recorded while the mice passively listened to randomized presentations of different auditory stimuli. In this study we consider a total of 16 stimuli, consisting of two groups of intensity modulated UP- or DOWN-ramping sounds. In each group, there were stimuli with different frequency content (either 8 kHz pure tones or white noise [WN] sounds), different durations (1 or 2 s) and different intensity slopes (either 50–85 dB or 60–85 dB and reversed, see *Table 1* and Materials and methods, Section 'The data set').

We first illustrate the responses of single cells to the presentation of different auditory stimuli, focusing on the periods following the onset and offset of the stimulus. The activity of individual

**Table 1.** Stimuli set.

| Stimulus | Direction | Frequency | Duration (s) | Modulation (dB) |
| --- | --- | --- | --- | --- |
| 1 | UP | 8 kHz | 1 s | 50–85 |
| 2 | UP | 8 kHz | 1 s | 60–85 |
| 3 | UP | 8 kHz | 2 s | 50–85 |
| 4 | UP | 8 kHz | 2 s | 60–85 |
| 5 | UP | WN | 1 s | 50–85 |
| 6 | UP | WN | 1 s | 60–85 |
| 7 | UP | WN | 2 s | 50–85 |
| 8 | UP | WN | 2 s | 60–85 |
| 9 | DOWN | 8 kHz | 1 s | 85–50 |
| 10 | DOWN | 8 kHz | 1 s | 85–60 |
| 11 | DOWN | 8 kHz | 2 s | 85–50 |
| 12 | DOWN | 8 kHz | 2 s | 85–60 |
| 13 | DOWN | WN | 1 s | 85–50 |
| 14 | DOWN | WN | 1 s | 85–60 |
| 15 | DOWN | WN | 2 s | 85–50 |
| 16 | DOWN | WN | 2 s | 85–60 |

neurons to different stimuli was highly heterogeneous. In response to a single stimulus, we found individual neurons that were strongly active only during the onset of the stimulus (ON responses), or only during the offset (OFF responses), while other neurons in the population responded to both stimulus onset and offset, consistent with previous analyses (*Deneux et al., 2016*). Importantly, across stimuli some neurons in the population exhibited ON and/or OFF responses only when specific stimuli were presented, showing stimulus-selectivity of transient responses, while others strongly responded at the onset and/or at the offset of multiple stimuli (*Figure 1A*).

Because of the intrinsic heterogeneity of single-cell responses, we examined the structure of the transient ON and OFF responses to different stimuli using a population approach (*Buonomano and Maass, 2009*; *Saxena and Cunningham, 2019*). The temporal dynamics of the collective response of all the neurons in the population can be represented as a sequence of states in a high-dimensional state space, in which the $i$-th coordinate corresponds to the (baseline-subtracted) firing activity $r_i(t)$ of the $i$-th neuron in the population. At each time point, the population response is described by a population activity vector $\mathbf{r}(t)$ which draws a *neural trajectory* in the state space (*Figure 1B left panel*).

To quantify the strength of the population transient ON and OFF responses, we computed the distance of the population activity vector from its baseline firing level (average firing rate before stimulus presentation), corresponding to the norm of the population activity vector $||\mathbf{r}(t)||$ (*Mazor and Laurent, 2005*). This revealed that the distance from baseline computed during ON and OFF responses was larger than the distance computed for the state at the end of stimulus presentation (*Figure 1B right panel*). We refer to this feature of the population transient dynamics as the transient amplification of ON and OFF responses.

To examine what the transient amplification of ON and OFF responses implies in terms of stimulus decoding, we trained a simple decoder to classify pairs of stimuli that differed in their frequency content, but had the same intensity modulation and duration. We found that the classification accuracy was highest during the transient phases corresponding to ON and OFF responses, while it decreased at the end of stimulus presentation (*Figure 1C*). This result revealed a robust encoding of the stimulus features during ON and OFF responses, as previously found in the locust olfactory system (*Mazor and Laurent, 2005*; *Saha et al., 2017*).

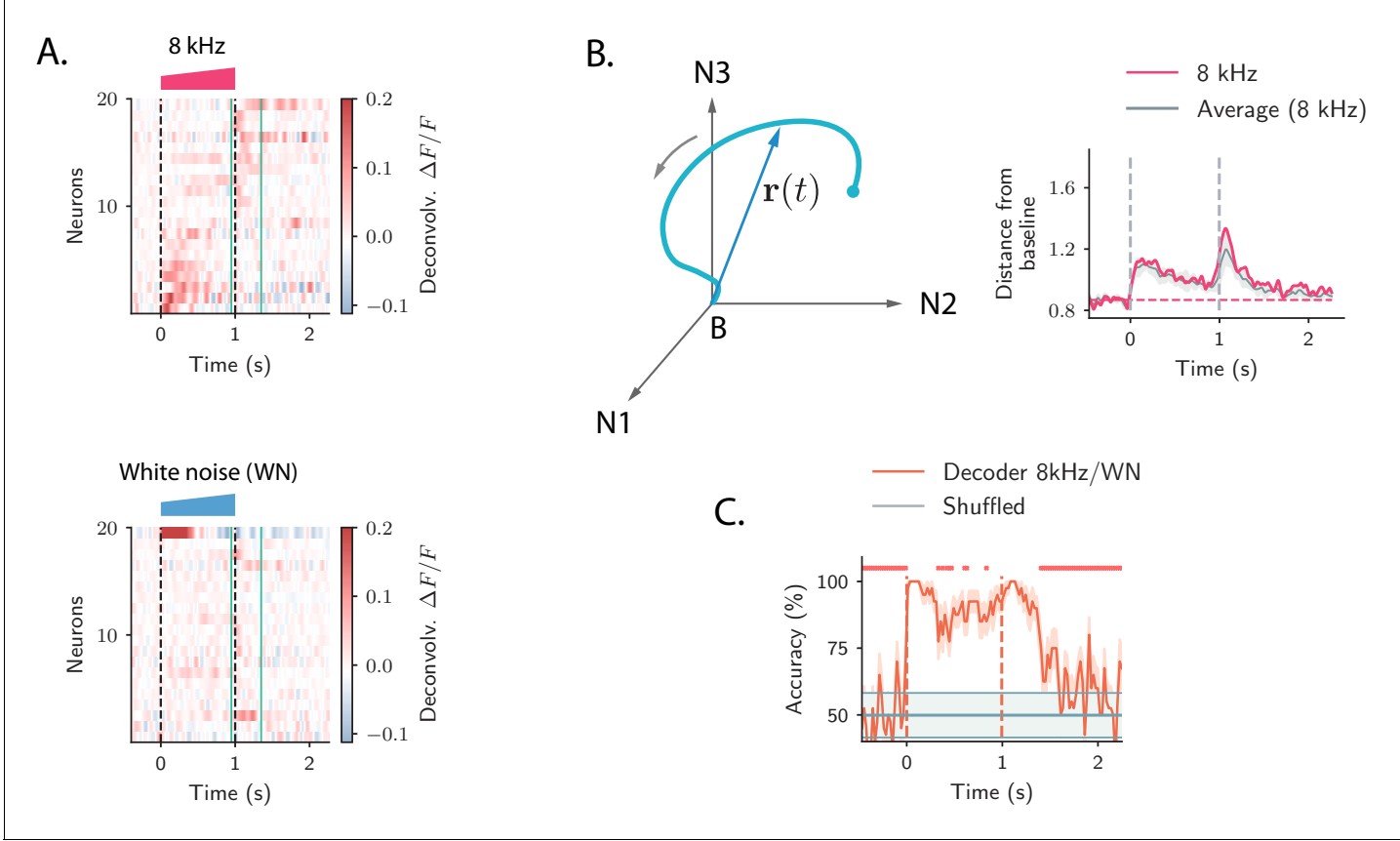

**Figure 1.** Strong transient ON and OFF responses in auditory cortex of mice passively listening to different sounds. (A) *Top*: deconvolved calcium signals averaged over 20 trials showing the activity (estimated firing rate) of 20 out of 2343 neurons in response to a 8 kHz 1 s UP-ramp with intensity range 60–85 dB. We selected neurons with high signal-to-noise ratios (ratio between the peak activity during ON/OFF responses and the standard deviation of the activity before stimulus presentation). Neurons were ordered according to the difference between peak activity during ON and OFF response epochs. *Bottom*: activity of the same neurons as in the top panel in response to a white noise (WN) sound with the same duration and intensity profile. In all panels dashed lines indicate onset and offset of the stimulus, and green solid lines show the temporal region where OFF responses were analyzed (from 50 ms before stimulus offset to 300 ms after stimulus offset). (B) *Left*: cartoon showing the OFF response to one stimulus as a neural trajectory in the state space, where each coordinate represents the firing rate of one neuron (with respect to the baseline B). The length of the dashed line represents the distance between the population activity vector and its baseline firing rate, that is, $\|\mathbf{r}(t)\|$. *Right*: the red trace shows the distance from baseline $\|\mathbf{r}(t)\|$ computed for the population response to the 8 kHz sound in **A**. The gray trace shows the distance from baseline averaged over the 8 kHz sounds of 1 s duration (four stimuli). The gray shading represents ±1 standard deviation. The dashed horizontal line shows the average level of the distance $\|\mathbf{r}(t)\|$ before stimulus presentation (even if baseline-subtracted responses were used, a value of the norm different from zero is expected because of noise in the spontaneous activity before stimulus onset). (C) Accuracy of stimulus classification between a 8 kHz versus WN UP-ramping sounds over time based on single trials (20 trials). The decoder is computed at each time step (spaced by ~50 ms) and accuracy is computed using leave-one-out cross-validation. Orange trace: average classification accuracy over the cross-validation folds. Orange shaded area corresponds to ±1 standard error. The same process is repeated after shuffling stimulus labels across trials at each time step (chance level). Chance level is represented by the gray trace and shading, corresponding to its average and ±1 std computed over time. The red markers on the top indicate the time points where the average classification accuracy is lower than the average accuracy during the ON transient response ($P<0.01$, two-tailed t-test).

## OFF responses rely on orthogonal low-dimensional subspaces

To further explore the structure of the neural trajectories associated with the population OFF responses to different stimuli, we analyzed neural activity using dimensionality reduction techniques (*Cunningham and Yu, 2014*). We focused specifically on responses within the period starting 50 ms before stimulus offset to 300 ms after stimulus offset.

By performing principal component analysis (PCA) independently for the responses to individual stimuli, we found that the dynamics during the transient OFF responses to individual stimuli explored

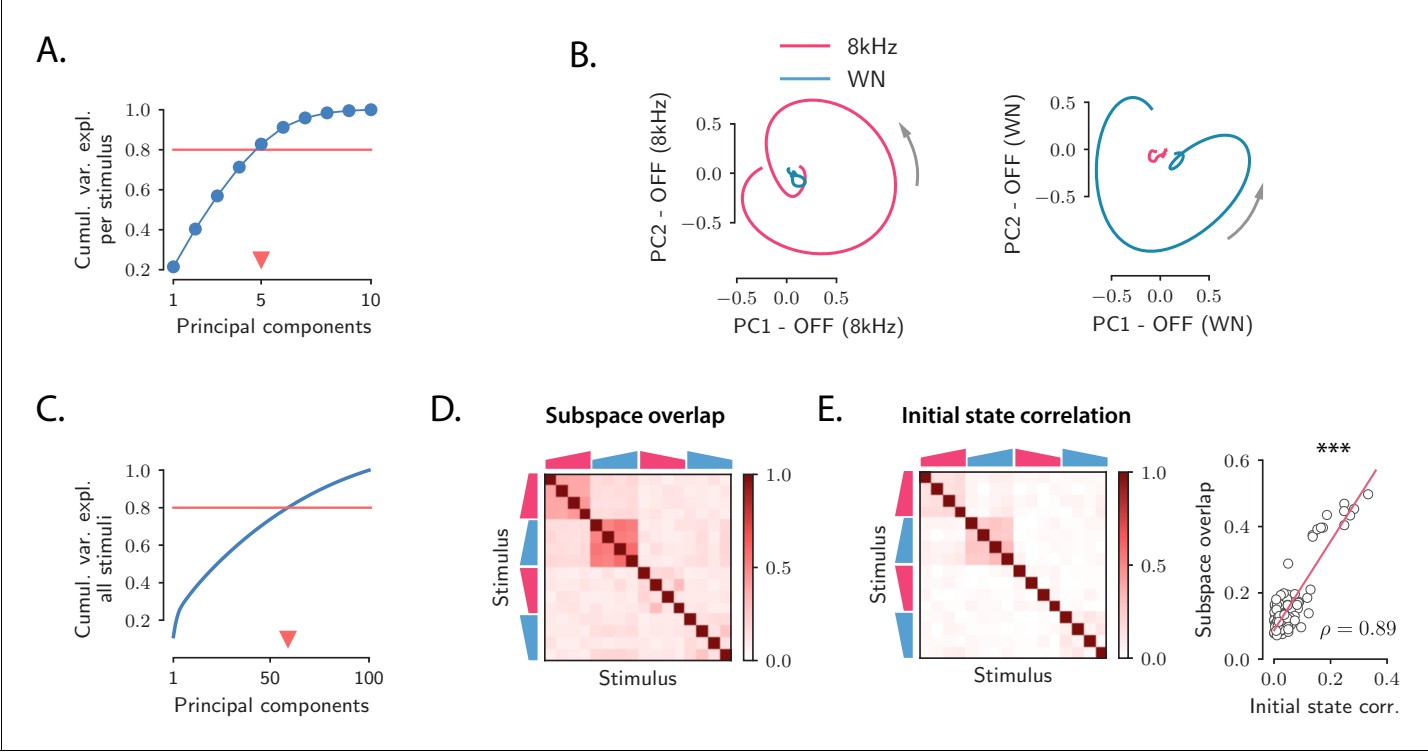

**Figure 2.** Low-dimensional structure of population OFF responses. (A) Cumulative variance explained for OFF responses to individual stimuli, as a function of the number of principal components. The blue trace shows the cumulative variance averaged across all 16 stimuli. Error bars are smaller than the symbol size. The triangular marker indicates the number of PCs explaining 80% (red line) of the total response variance for individual stimuli. (B) *Left*: projection of the population OFF response to the 8 kHz and white noise (WN) sounds on the first two principal components computed for the OFF response to the 8 kHz sound. *Right*: projection of both responses onto the first two principal components computed for the OFF response to the WN sound. PCA was performed on the period from −50 ms to 300 ms with respect to stimulus offset. We plot the response from 50 ms before stimulus offset to the end of the trial duration. (C) Cumulative variance explained for the OFF responses to all 16 stimuli together as a function of the number of principal components. The triangular marker indicates the number of PCs explaining 80% of the total response variance for all stimuli. (D) Overlap between the subspaces defined by the first five principal components of the OFF responses corresponding to pairs of stimuli. The overlap is measured by the cosine of the principal angle between these subspaces (see Materials and methods, Section 'Correlations between OFF response subspaces').
(E) *Left*: correlation matrix between the initial population activity states $\mathbf{r}_0^{(s)} = \mathbf{r}^{(s)}(0)$ at the end of stimulus presentation (50 ms before offset) for each pair of stimuli. *Right*: linear correlation between subspace overlaps (D) and the overlap between initial states $\mathbf{r}_0^{(s)}$ (E left panel) for each stimulus pair. The component of the dynamics along the corresponding initial states was subtracted before computing the subspace overlaps.
The online version of this article includes the following figure supplement(s) for figure 2:

**Figure supplement 1.** Controls for PCA of OFF responses to individual stimuli and across stimuli.
**Figure supplement 2.** Controls for the orthogonality between OFF response subspaces.
**Figure supplement 3.** Relation between ON and OFF responses in the auditory cortex.
**Figure supplement 4.** Overlap between the states at the peak of the transient OFF responses.

only about five dimensions, as 80% of the variance of the OFF responses to individual stimuli was explained on average by the first five principal components (*Figure 2A*; see *Figure 2—figure supplement 1* for cross-validated controls; note that, given the temporal resolution, the maximal dimensionality explaining 80% of the variance of the responses to individual stimuli was 9). The projection of the low-dimensional OFF responses to each stimulus onto the first two principal components revealed circular activity patterns, where the population vector smoothly rotated between the two dominant dimensions (*Figure 2B*).

A central observation revealed by the dimensionality reduction analysis was that the OFF response trajectories relative to stimuli with different frequency content spanned orthogonal low-dimensional subspaces. For instance, the response to the 8 kHz sound was poorly represented on

the plane defined by the two principal components of the response to the WN sound (*Figure 2B*), and vice versa, showing that they evolved in distinct subspaces. To quantify the relationship between the subspaces spanned by the OFF responses to different stimuli, we proceeded as follows. We first computed the first five principal components of the (baseline-subtracted) OFF response $\mathbf{r}^{(s)}(t)$ to each individual stimulus $s$. Therefore, for each stimulus $s$ these dimensions define a five-dimensional subspace. Then, for each pair of stimuli, we computed the relative orientation of the corresponding pair of subspaces, measured by the subspace overlap (*Figure 2D*; see Materials and methods, Section 'Correlations between OFF response subspaces').

This approach revealed an interesting structure between the OFF response subspaces for different stimuli (*Figure 2D* and *Figure 2—figure supplement 2*). Stimuli with different frequency content evoked in general non-overlapping OFF responses reflected in low values of the subspace overlap. Two clusters of correlated OFF responses emerged, corresponding to the 8 kHz UP-ramps and WN UP-ramps of different durations and intensity. Surprisingly, DOWN-ramps evoked OFF responses that were less correlated than UP-ramps, even for sounds with the same frequency content.

The fact that most of the stimuli evoked non-overlapping OFF responses is reflected in the number of principal components that explain 80% of the variance for all OFF responses considered together, which is around 60 (*Figure 2C* and *Figure 2—figure supplement 1*). This number is in fact close to the number of dominant components of the joint response to all 16 stimuli (see *Table 1*) that we would expect if the responses to individual stimuli evolved on uncorrelated subspaces (given by #PC per stimulus $\times$ #stimuli $\approx 80$). Notably this implies that while the OFF responses to individual stimuli span low-dimensional subspaces, the joint response across stimuli shows high-dimensional structure.

We finally examined to what extent the structure observed between the OFF response trajectories $\mathbf{r}^{(s)}(t)$ to different stimuli (*Figure 2D*) could be predicted from the structure of the population activity states reached at the end of stimulus presentation, corresponding to the initial states $\mathbf{r}^{(s)}(0)$. Remarkably, we found that the initial states exhibited structure across stimuli that matched well the structure of OFF response dynamics (*Figure 2E left panel* and *Figure 2—figure supplement 4*), even though the component along the initial states was subtracted from the corresponding OFF response trajectories (*Figure 2E right panel* and *Figure 2—figure supplement 4*). This suggests that initial states contribute to determining the subsequent dynamics of the OFF responses.

## Single-cell model for OFF responses

Our analyses of auditory cortical data identified three prominent features of population dynamics: (i) OFF responses correspond to transiently amplified trajectories; (ii) responses to individual stimuli explore low-dimensional subspaces; (iii) responses to different stimuli lie in largely orthogonal subspaces. We next examined to what extent these three features could be accounted for by a single-cell mechanisms for OFF response generation.

The AC is not the first stage where OFF responses arise. Robust OFF responses are found throughout the auditory pathway, and in subcortical areas the generation of OFF responses most likely relies on mechanisms that depend on the interaction between the inhibitory and excitatory synaptic inputs to single cells (e.g. post-inhibitory rebound or facilitation, see *Kopp-Scheinpflug et al., 2018*). To examine the possibility that OFF responses in AC are consistent with a similar single-cell mechanism, we considered a simplified, linear model that could be directly fit to calcium recordings in the AC. Adapting previously used models (*Anderson and Linden, 2016*; *Meyer et al., 2016*), we assumed that the cells received no external input after stimulus offset, and that the response of neuron $i$ after stimulus offset is specified by a characteristic linear filter, which describes the cell's intrinsic response generated by intracellular or synaptic mechanisms (*Figure 3A*). We moreover assumed that the shape of this temporal response is set by intrinsic properties, and is therefore identical across different stimuli. In the model, each stimulus modulates the response of a single neuron linearly depending on its activity at stimulus offset, so that the (baseline-subtracted) OFF response of neuron $i$ to stimulus $s$ is written as:

$$r_i^{(s)}(t) = r_{0,i}^{(s)} L_i(t) \tag{1}$$

where $r_{0,i}^{(s)}$ is the modulation factor for neuron $i$ and stimulus $s$. We estimated the single-cell

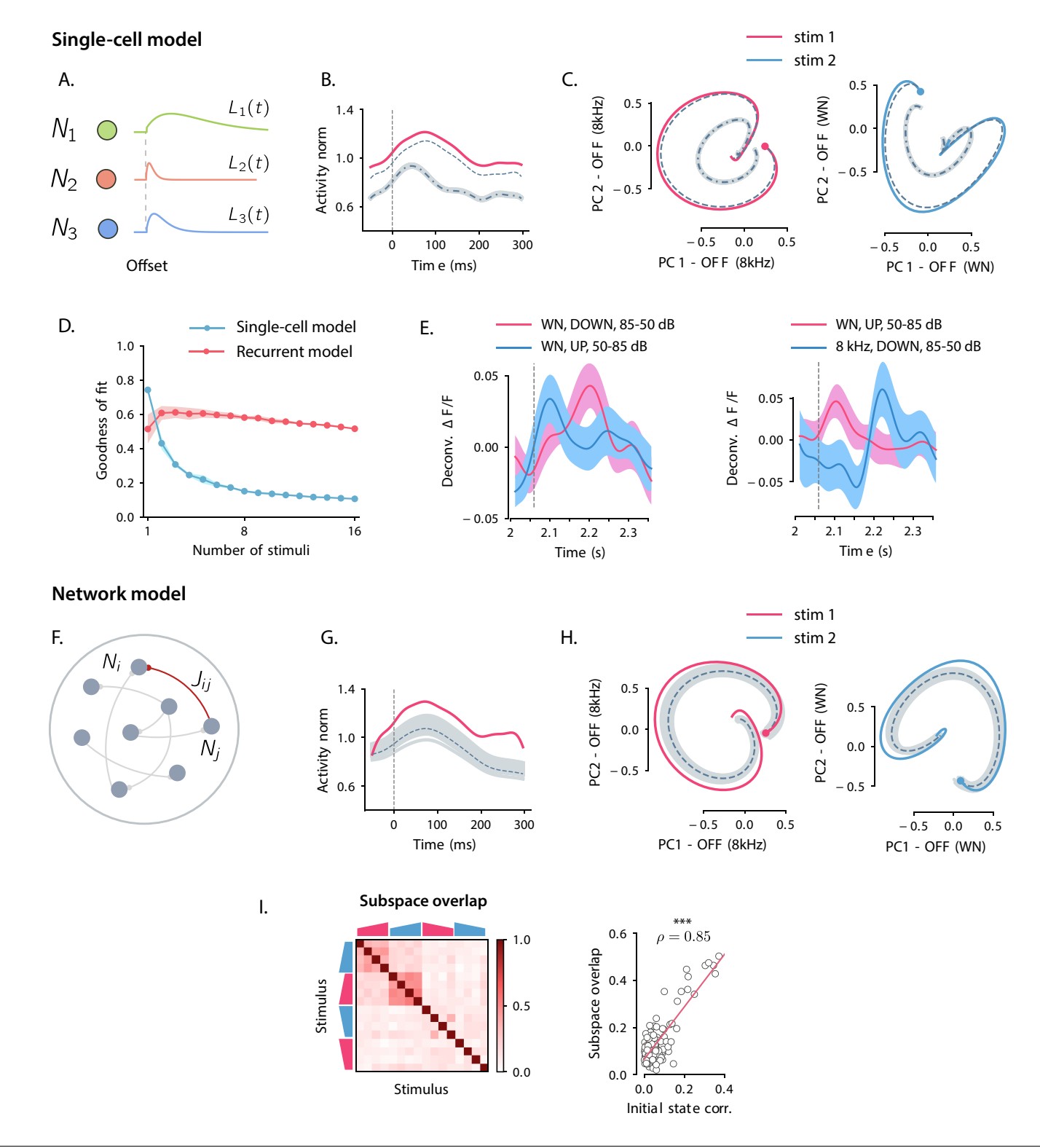

**Figure 3.** Comparison between single-cell and network models for OFF response generation. (**A**) Cartoon illustrating the single-cell model defined in *Equation (1)*. The activity of each unit is described by a characteristic temporal response $L_i(t)$ (colored traces). Across stimuli, only the relative activation of the single units changes, while the temporal shape of the responses of each unit $L_i(t)$ does not vary. (**B**) Distance from baseline of the population activity vector during the OFF response to one example stimulus (red trace; same stimulus as in C *left panel*). The dashed vertical line indicates the time of stimulus offset. Gray dashed lines correspond to the norms of the population activity vectors obtained from fitting the single-cell

*Figure 3 continued on next page*

*Figure 3 continued*

model respectively to the OFF response to a single stimulus (dashed line), and simultaneously to the OFF responses to two stimuli (dash-dotted line; in this example, the two fitted stimuli are the ones considered in panel **C**). In light gray we plot the norm of 100 realizations of the fitted OFF response by simulating different initial states distributed in the vicinity of the fitted initial state (shown only for the simultaneous fit of two stimuli for clarity). Note that in the single-cell model fit (see Materials and methods, Section 'Fitting the single-cell model'), the fitted initial condition can substantially deviate from the initial condition taken from the data. (**C**) Colored traces: projection of the population OFF responses to two distinct example stimuli (same stimuli as in *Figure 2B*) onto the first two principal components of either stimulus. As in panel **B** gray dashed and dash-dotted traces correspond to the projection of the OFF responses obtained when fitting the single-cell model to one or two stimuli at once. (**D**) Goodness of fit (as quantified by coefficient of determination $R^2$) computed by fitting the single-cell model (blue trace) and the network model (red trace) to the calcium activity data, as a function of the number of stimuli included in the fit. Both traces show the cross-validated value of the goodness of fit (10-fold cross-validation in the time domain). Error bars represent the standard deviation over multiple subsamplings of a fixed number of stimuli (reported on the abscissa). Prior to fitting, for each subsampling of stimuli, we reduced the dimensionality of the responses using PCA, and kept the dominant dimensions that accounted for 90% of the variance. (**E**) Examples of the OFF responses to distinct stimuli of two different auditory cortical neurons (each panel corresponds to a different neuron). The same neuron exhibits different temporal response profiles for different stimuli, a feature consistent with the network model (see *Figure 6A,D*), but not with the single-cell model. (**F**) Illustration of the recurrent network model. The variables defining the state of the network are the (baseline-subtracted) firing rates of the units, denoted by $r_i(t)$. The strength of the connection from unit $j$ to unit $i$ is denoted by $J_{ij}$. (**G**) Distance from baseline of the population activity vector during the OFF response to one example stimulus (red trace; same stimulus as in C *left panel*). Gray traces correspond to the norms of the population activity vectors obtained from fitting the network model to the OFF response to a single stimulus and generated using the fitted connectivity matrix. The initial conditions were chosen in a neighborhood of the population activity vector 50 ms before sound offset. 100 traces corresponding to 100 random choices of the initial condition are shown. Dashed trace: average distance from baseline. (**H**) Colored traces: projection of the population OFF responses to two different stimuli (same stimuli as in *Figure 2B*) on the first two principal components. Gray traces: projections of multiple trajectories generated by the network model using the connectivity matrix fitted to the individual stimuli as in **G**. The initial condition is indicated with a dot. 100 traces are shown. Dashed trace: projection of the average trajectory. (**I**) *Left*: overlap between the subspaces corresponding to OFF response trajectories to pairs of stimuli (as in *Figure 2D*) generated using the connectivity matrix fitted to all stimuli at once. *Right*: linear correlation between subspace overlaps and the overlaps between the initial states $\mathbf{r}_0^{(s)}$ for each stimulus pair computed using the trajectories obtained by the fitted network model. The component of the dynamics along the corresponding initial states was subtracted before computing the subspace overlaps.

The online version of this article includes the following figure supplement(s) for figure 3:

**Figure supplement 1.** Goodness of fit for the network and single-cell models adjusted for the number of parameters.

**Figure supplement 2.** Goodness of fit for the original and surrogate data sets.

**Figure supplement 3.** Goodness of fit as a function of PC dimensionality for original and surrogate data sets.

responses $L_i(t)$ from the data by fitting basis functions to the responses of neurons subject to prior normalization by the modulation factors $r_{0,i}^{(s)}$ using linear regression (see Materials and methods, Section 'Single-cell model for OFF response generation'). We first fitted the single-cell model to responses to a single stimulus, and then increased the number of stimuli.

The single-cell model accounted well for the first two features observed in the data: (i) because the single-cell responses $L_i(t)$ are in general non-monotonic, the distance from baseline of the population activity vector displayed amplified dynamics (*Figure 3B*); (ii) at the population level, the trajectories generated by the single-cell model spanned low-dimensional subspaces of at least two dimensions (*Figure 3C*) and provided excellent fits to the trajectories in the data when the response to a single stimulus was considered (*Figure 3C*). However, we found that the single-cell model could not account for the structure of responses across multiple stimuli. Indeed, although fitting the model to OFF responses to a single stimulus led to an excellent match with auditory cortical data (coefficient of determination $R^2 = 0.75$), increasing the number of simultaneously fitted stimuli led to increasing deviations from the data (*Figure 3B,C*), and strongly decreased the goodness of fit (*Figure 3D*). When fitting all stimuli at once, the goodness of fit was extremely poor (coefficient of determination $R^2 = 0.1$), and therefore the single-cell model could not provide useful information about the third feature of the data, the structure of subspaces spanned in response to different stimuli. A simple explanation for this inability to capture structure across stimuli is that in the single-cell model the temporal shape of the response of each neuron is the same across all stimuli, while this was not the case in the data (*Figure 3E*).

## Network model for OFF responses

We contrasted the single-cell model with a network model that generated OFF responses through collective interactions between neurons. Specifically, we studied a linear network of $N$ recurrently coupled linear rate units with time evolution given by:

$$\dot{r}_i = -r_i + \sum_{j=1}^{N} J_{ij} r_j. \tag{2}$$

The quantity $r_i(t)$ represents the deviation of the activity of the unit $i$ from its baseline firing rate, while $J_{ij}$ denotes the effective strength of the connection from neuron $j$ to neuron $i$ (**Figure 3F**). As in the single-cell model, we assumed that the network received no external input after stimulus offset. The only effect of the preceding stimulus was to set the initial pattern of activity of the network at $t = 0$. The internal recurrent dynamics then fully determined the subsequent evolution of the activity. Each stimulus $s$ was thus represented by an initial state $\mathbf{r}_0^{(s)}$ that was mapped onto a trajectory of activity $\mathbf{r}^{(s)}(t)$. We focused on the dynamics of the network following stimulus offset, which we represent as $t = 0$.

To quantify to what extent recurrent interactions could account for auditory cortical dynamics, we fitted the network model to OFF responses using linear regression (see Materials and methods, Section 'Fitting the network model'). For each stimulus, the pattern of initial activity $\mathbf{r}_0^{(s)}$ was taken from the data, and the connectivity matrix $\mathbf{J}$ was fitted to the responses to a progressively increasing number of stimuli.

Qualitatively, we found that the fitted network model reproduced well the first two features of the data, transient amplification and low-dimensional population trajectories (**Figure 3G,H**). Quantitatively, we evaluated the goodness of fit by computing the coefficient of determination $R^2$. While the goodness of fit of the network model was lower than the single-cell model when fitting the response to a single stimulus ($R^2 = 0.52$), for the network model the goodness of fit remained consistently high as the number of simultaneously fitted stimuli was increased (**Figure 3D**, $R^2 = 0.52$ when fitted on the responses to all stimuli). Computing the goodness of fit by taking into account the number of parameters of the network and single-cell models led to the same conclusion (**Figure 3—figure supplement 1**). When fitted to all stimuli at once, the fitted network model captured the structure of the subspace overlaps (**Figure 3I left panel**) and its correlation with the structure of initial conditions (**Figure 3I right panel**). In contrast to the single-cell model, the network model therefore accounted well for the third feature of the data, the structure of responses across stimuli. This can be explained by the fact that, in the network model, the temporal OFF-responses of single cells can in general differ across stimuli (see Figure 6A,D), similar to the activity in the AC (**Figure 3E**).

## Testing the network mechanisms of OFF responses on the data

Having found that the fitted network model reproduced the three main features of the data, we next analyzed this model to identify the mechanisms responsible for each feature. The identified mechanisms provided new predictions that we tested on the effective connectivity matrix $\mathbf{J}$ obtained from the fit to the data.

## Transiently amplified OFF responses

The first feature of the data was that the dynamics were transiently amplified, i.e. the OFF responses first deviated from baseline before eventually converging to it. A preliminary requirement to reproduce this feature is that dynamics are stable, that is, eventually decay to baseline following any initial state. This requirement leads to the usual condition that the eigenvalues of the connectivity matrix $\mathbf{J}$ have real parts less than unity. Provided this basic requirement is met, during the transient dynamics from the initial state at stimulus offset, the distance from baseline can either monotonically decrease, or transiently increase before eventually decreasing. To generate the transient increase, the connectivity matrix needs to belong to the class of non-normal matrices (**Trefethen and Embree, 2005**; **Murphy and Miller, 2009**; **Goldman, 2009**; **Hennequin et al., 2012**; see Materials and methods, Section 'Normal and non-normal connectivity matrices'), but this is not a sufficient condition. More specifically, the largest eigenvalue of the symmetric part defined by $\mathbf{J}_S = (\mathbf{J} + \mathbf{J}^T)/2$ needs to be larger than unity, while the initial state $\mathbf{r}_0$ needs to belong to a subset of amplified patterns

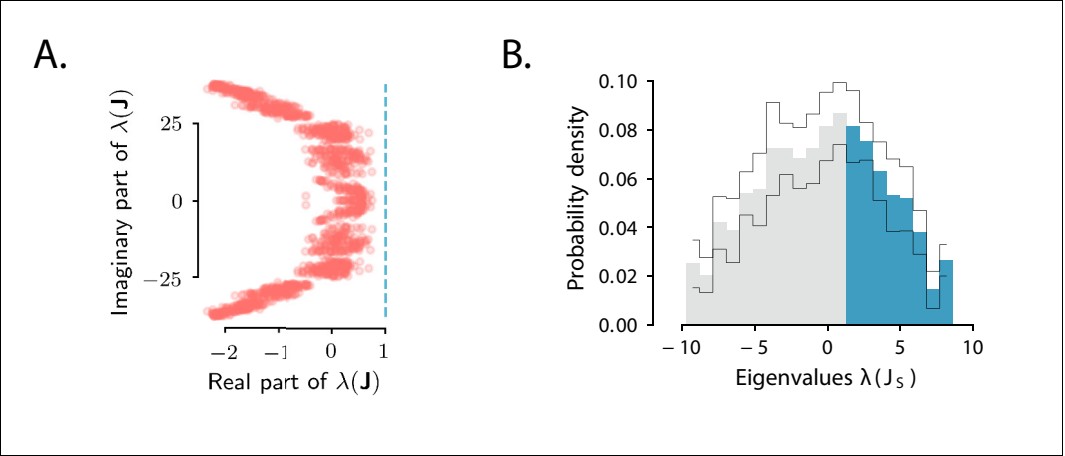

**Figure 4.** Spectra of the connectivity matrix of the fitted network model. (A) Eigenvalues of the connectivity matrix **J** obtained by fitting the network model to the population OFF responses to all 1 s stimuli at once. The dashed line marks the stability boundary given by $\Re\lambda(\mathbf{J}) = 1$. (B) Probability density distribution of the eigenvalues of the symmetric part of the effective connectivity, $\mathbf{J}_S$. Eigenvalues larger than unity ($\lambda(\mathbf{J}_S) = 1$; highlighted in blue) determine strongly non-normal dynamics. In A and B the total dimensionality of the responses was set to 100. The fitting was performed on 20 bootstrap subsamplings (with replacement) of 80% of neurons out of the total population. Each bootstrap subsampling resulted in a set of 100 eigenvalues. In panel A we plotted the eigenvalues of **J** obtained across all subsamplings. In panel B the thin black lines indicate the standard deviation of the eigenvalue probability density across subsamplings. In panels A and B, the temporal window considered for the fit was extended from 350 ms to 600 ms, to include the decay to zero baseline of the OFF responses. The extension of the temporal window was possible only for the 1 s long stimuli (n=8), since the length of the temporal window following stimulus offset for the 2 s stimuli was limited by the length of the neural recordings.

(***Bondanelli and Ostojic, 2020***; see Materials and methods, Section 'Sufficient condition for amplified OFF responses').

To test the predictions derived from the condition for amplified transient responses, that is, that **J** belongs to a specific subset of non-normal matrices, we examined the spectra of the full connectivity **J** and of its symmetric part $\mathbf{J}_S$ (***Bondanelli and Ostojic, 2020***). The eigenvalues of the fitted connectivity matrix had real parts smaller than one, indicating stable dynamics, and large imaginary parts (***Figure 4A***). Our theoretical criterion predicted that for OFF responses to be amplified, the spectrum of the symmetric part $\mathbf{J}_S$ must have at least one eigenvalue larger than unity. Consistent with this predictions, we found that the symmetric part of the connectivity had indeed a large number of eigenvalues larger than one (***Figure 4B***) and could therefore produce amplified responses (***Figure 3G***).

## Low-dimensionality of OFF response trajectories

The second feature of auditory cortical data was that each stimulus generated a population response embedded in an approximately five-dimensional subspace of the full state space. We hypothesized that low-dimensional responses in the fitted network model originated from a low-rank structure in the connectivity (***Mastrogiuseppe and Ostojic, 2018***), implying that the connectivity matrix could be approximated in terms of $R \ll N$ modes, that is, as

$$\mathbf{J} = \mathbf{u}^{(1)}\mathbf{v}^{(1)T} + \mathbf{u}^{(2)}\mathbf{v}^{(2)T} + ... + \mathbf{u}^{(R)}\mathbf{v}^{(R)T}, \tag{3}$$

where each mode was specified by two $N$-dimensional vectors $\mathbf{u}^{(r)}$ and $\mathbf{v}^{(r)}$, which we term the right and left connectivity patterns respectively (***Mastrogiuseppe and Ostojic, 2018***). This set of connectivity patterns is uniquely defined from the singular value decomposition (SVD) of the connectivity matrix **J** (see Materials and methods, Section 'Low-dimensional dynamics').

To test the low-rank hypothesis, we re-fitted the network model while constraining the rank of connectivity matrix (***Figure 5A***, ***Figure 5—figure supplement 1***, ***Figure 5—figure supplement 2***;

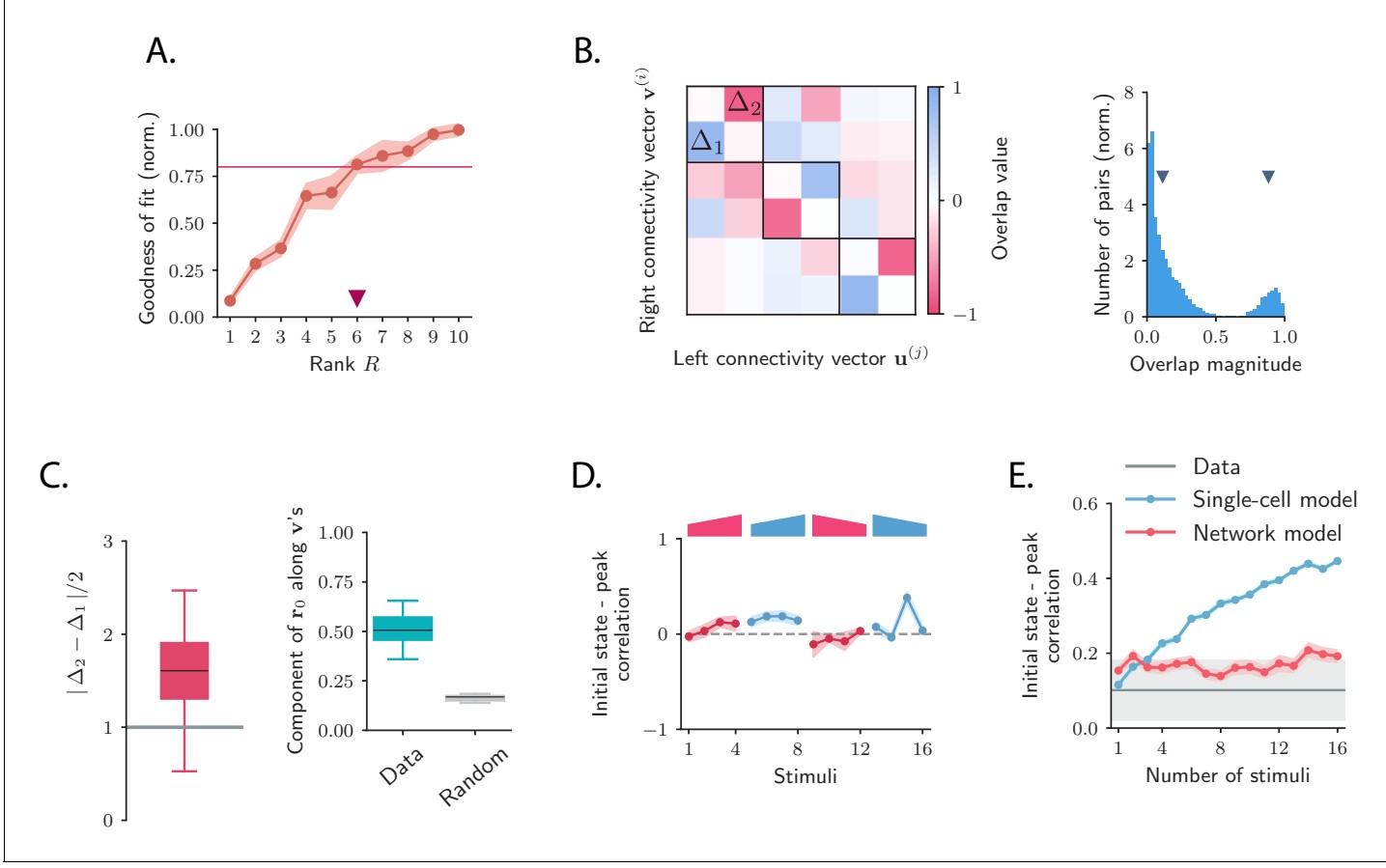

**Figure 5.** Low-dimensional structure of the dynamics of OFF responses to individual stimuli. (A) Goodness of fit (coefficient of determination) as a function of the rank $R$ of the fitted network, normalized by the goodness of fit computed using ordinary least squares ridge regression. The shaded area represents the standard deviation across individual stimuli. For $R = 6$ reduced-rank regression captures more than 80% (red solid line) of the variance explained by ordinary least squares regression. (B) *Left*: overlap matrix $\mathbf{J}^{ov}$ consisting of the overlaps between left and right connectivity patterns of the connectivity $\mathbf{J}$ fitted on one example stimulus. The color code shows strong (and opposite in sign) correlation values between left and right connectivity patterns within pairs of nearby modes. Weak coupling is instead present between different rank-2 channels (highlighted by dark boxes). *Right*: histogram of the absolute values of the correlation between right and left connectivity patterns, across all stimuli, and across 20 random subsamplings of 80% of the neurons in the population for each stimulus. Left and right connectivity vectors are weakly correlated, except for the pairs corresponding to the off-diagonal couplings within each rank-2 channel. The two markers indicate the average values of the correlations within each group. When fitting individual stimuli the rank parameter $R$ and the number of principal components are set respectively to 6 and 100. (C) *Left*: absolute value of the difference between $\Delta_1$ and $\Delta_2$ (see panel **B**) divided by 2, across stimuli. For a rank-R channel (here $R = 6$) comprising $R/2$ rank-2 channels, the maximum difference $|\Delta_1 - \Delta_2|/2$ across the $R/2$ rank-2 channels is considered. Large values of this difference indicate that the dynamics of the corresponding rank-R channel are amplified. *Right*: component of the initial state $\mathbf{r}_0$ on the left connectivity vectors $\mathbf{v}^{(2;k)}$, (i.e. $(\sum_k \alpha_2^{(k)2})^{1/2}$ in *Equation (75)*), obtained from the fitted connectivity $\mathbf{J}$ to individual stimuli (in green). The component of the initial condition along the left connectivity vectors $\mathbf{v}^{(2;k)}$ (green box) is larger than the component of random vectors along the same vectors (gray box). In both panels the boxplots show the distributions across all stimuli of the corresponding average values over 20 random subsamplings of 80% of the neurons. The rank parameter and the number of principal components are the same as in **B**. (D) For each stimulus, we show the correlation between the state at the end of stimulus presentation and the state at the peak of the OFF response, defined as the time of maximum distance from baseline. Error bars represent the standard deviation computed over 2000 bootstrap subsamplings of 50% of the neurons in the population (2343 neurons). (E) Correlation between initial state and peak state obtained from fitting the single-cell (in green) and network models (in red) to a progressively increasing number of stimuli. For each fitted response the peak state is defined as the population vector at the time of maximum distance from baseline of the original response. The colored shaded areas correspond to the standard error over 100 subsamplings of a fixed number of stimuli (reported on the abscissa) out of 16 stimuli. For each subsampling of stimuli, the correlation is computed for one random stimulus.

The online version of this article includes the following figure supplement(s) for figure 5:

**Figure supplement 1.** Selection of hyperparameters in rank-reduced ridge regression.

**Figure supplement 2.** Model recovery for simulated OFF responses using a low-rank network model.

**Figure supplement 3.** Controls for the orthogonality between initial and peak state.

see Materials and methods, Section 'Fitting the network model'). Progressively increasing the rank $R$, we found that $R = 6$ was sufficient to capture more than 80% of the variance explained by the network model when fitting the responses to individual stimuli (*Figure 5A*).

The dynamics in the obtained low-rank network model are therefore fully specified by a set of patterns over the neural population: patterns corresponding to initial states $\mathbf{r}_0$ determined by the stimuli, and patterns corresponding to connectivity vectors $\mathbf{u}^{(r)}$ and $\mathbf{v}^{(r)}$ for $r = 1, \ldots, R$ that determine the network dynamics. Recurrent neural networks based on such connectivity directly generate low-dimensional dynamics: for a given stimulus $s$ the trajectory of the dynamics lies in the subspace spanned by the initial state $\mathbf{r}_0^{(s)}$ and the set of right connectivity patterns $\{\mathbf{u}^{(r)}\}_{r=1,\ldots,R}$ (*Mastrogiuseppe and Ostojic, 2018*; *Bondanelli and Ostojic, 2020*; *Schuessler et al., 2020*; *Beiran et al., 2020*). More specifically, the transient dynamics can be written as (see Materials and methods, Section 'Low-dimensional dynamics')

$$\mathbf{r}(t) = e^{-t}\mathbf{r}_0 + e^{-t}\mathbf{U}\mathbf{J}^{ov-1}\big[\exp(t(\mathbf{J}^{ov} - \mathbf{I}))\big]\big(\mathbf{V}^T\mathbf{r}_0\big), \tag{4}$$

where $\mathbf{U}$ and $\mathbf{V}$ are $N \times R$ matrices that contain respectively the $R$ right and left connectivity vectors as columns, and $\mathbf{J}^{ov}$ is the $R \times R$ matrix of overlaps $\mathbf{J}^{ov}_{kl} = \mathbf{v}^{(k)T}\mathbf{u}^{(l)}$ between left and right connectivity vectors. This overlap matrix therefore fully determines the dynamics in the network (see Materials and methods, Section 'Low-dimensional dynamics').

Inspecting the overlap matrix $\mathbf{J}^{ov}$ obtained from the fitted connectivity matrix revealed a clear structure, where left and right vectors were strongly correlated within pairs, but uncorrelated between different pairs (*Figure 5B left panel*). This structure effectively defined a series of $R/2$ rank-2 channels that were orthogonal to each other. Within individual rank-2 channels, strong correlations were observed only across the two modes (e.g. between $\mathbf{v}^{(1)}$ and $\mathbf{u}^{(2)}$, $\mathbf{v}^{(2)}$ and $\mathbf{u}^{(1)}$, etc.; *Figure 5B*), so that the connectivity matrix corresponding to each rank-2 channel can be written as

$$\mathbf{J}_2 = \Delta_1 \mathbf{v}^{(2)}\mathbf{v}^{(1)T} - \Delta_2 \mathbf{v}^{(1)}\mathbf{v}^{(2)T}, \tag{5}$$

where $\Delta_1$ and $\Delta_2$ are two scalar values. Rank-2 matrices of this type have purely imaginary eigenvalues given by $\pm i\sqrt{\Delta_1\Delta_2}$, reflecting the strong imaginary components in the eigenspectrum of the full matrix (*Figure 4A*). These imaginary eigenvalues lead to strongly rotational dynamics in the plane defined by $\mathbf{v}^{(1)}$ and $\mathbf{v}^{(2)}$ (*Figure 6C*; see Materials and methods, Section 'Dynamics of a low-rank rotational channel'), qualitatively similar to the low-dimensional dynamics in the data (*Figure 2B*). Such rotational dynamics however do not necessarily imply transient amplification. In fact, a rank-2 connectivity matrix as in *Equation (5)* generates amplified dynamics only if two conditions are satisfied (*Figure 6B*; see Materials and methods, Section 'Dynamics of a low-rank rotational channel'): (i) the difference $|\Delta_2 - \Delta_1|/2$ is greater than unity; (ii) the initial state $\mathbf{r}_0$ overlaps strongly with the left connectivity patterns $\mathbf{v}^{(r)}$. A direct consequence of these constraints is that when dynamics are transiently amplified, the population activity vector at the peak of the transient dynamics lies along a direction that is largely orthogonal to the initial state at stimulus offset (*Figure 6C,E*, see Materials and methods, Section 'Correlation between initial and peak state').

The theoretical analyses of the model therefore provide a new set of predictions that we directly tested on the connectivity fitted to the data. We first computed the differences $|\Delta_2 - \Delta_1|/2$ for each channel using the SVD of the fitted matrix $\mathbf{J}^{(s)}$, and found that they were sufficiently large to amplify the dynamics within each rank-$R$ channel (*Figure 5C left panel*). We next examined for each stimulus the component of the initial state $\mathbf{r}_0$ on the $\mathbf{v}^{(r)}$'s and found that it was significantly larger than the component on the $\mathbf{v}^{(r)}$'s of a random vector, across all stimuli (*Figure 5C right panel*). Finally, computing the correlation between the peak state and the initial state at the end of stimulus presentation, we found that this correlation took values lower than values predicted by chance for almost all stimuli, consistent with the prediction of the network model (*Figure 5D*, *Figure 5—figure supplement 3*). The single-cell model instead predicts stronger correlations between initial and peak states, in clear conflict with the data (*Figure 5E*).

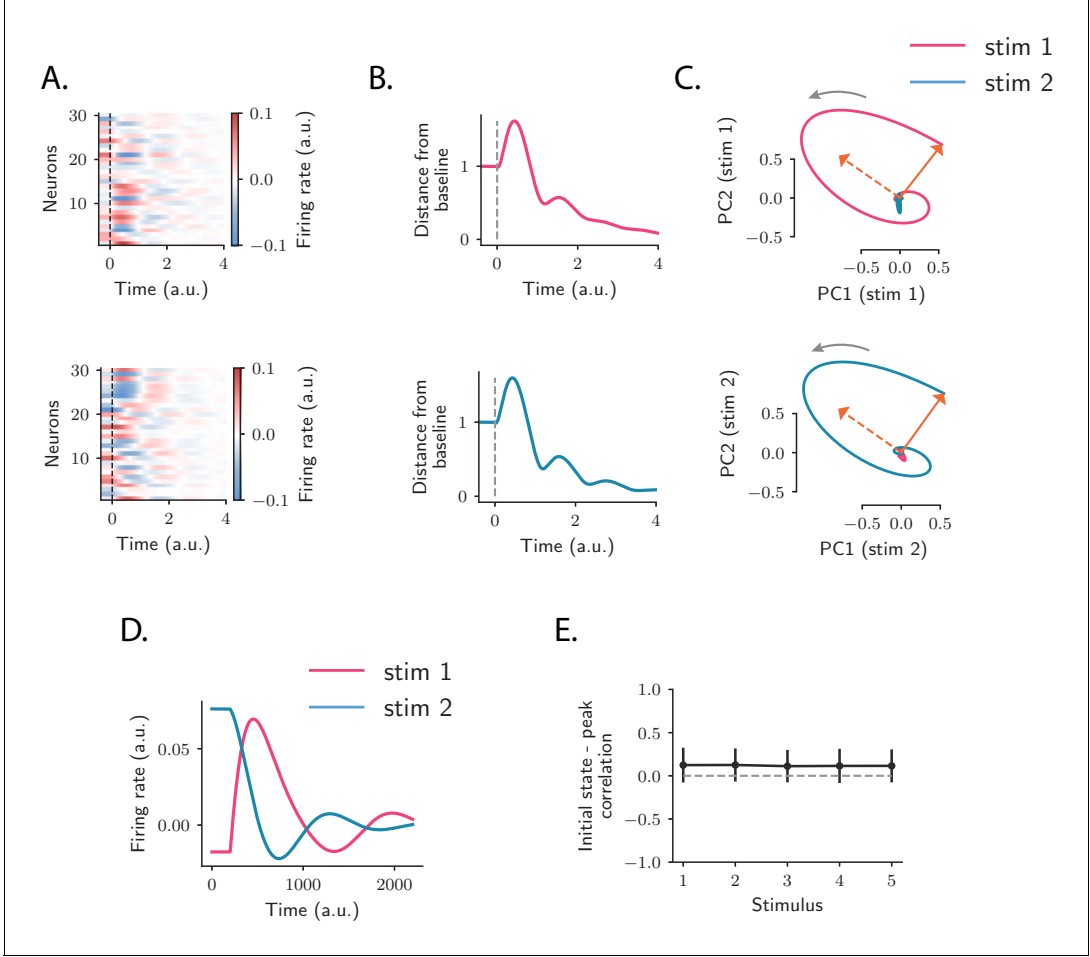

**Figure 6.** Population OFF responses in a network model with low-rank rotational structure. (**A**) Single-unit OFF responses to two distinct stimuli generated by orthogonal rank-2 rotational channels (see Materials and methods, Section 'Dynamics of a low-rank rotational channel'). The two stimuli were modeled as two different activity states $\mathbf{r}_0^{(1)}$ and $\mathbf{r}_0^{(2)}$ at the end of stimulus presentation ($t = 0$). We simulated a network of 1000 units. The connectivity consisted of the sum of $P = 20$ rank-2 rotational channels of the form given by **Equation (5, 6)** ($R_1 = R_2 = ... = R_P = 2$ in **Equation (6)** and $\Delta_1 = ||\mathbf{u}^{(1;s)}|| = 1$, $\Delta_2 = ||\mathbf{u}^{(2;s)}|| = 7$ for all $s = 1, ..., P$). Here the two stimuli were chosen along two orthogonal vectors $\mathbf{v}^{(2;s=1)}$ and $\mathbf{v}^{(2;s=2)}$. Dashed lines indicate the time of stimulus offset. (**B**) Distance from baseline of the population activity vector during the OFF responses to the two stimuli in **A**. For each stimulus, the amplitude of the offset responses (quantified by the peak value of the distance from baseline) is controlled by the difference between the lengths of the right connectivity vectors of each rank-2 channel, that is, $|\Delta_2 - \Delta_1|/2$. (**C**) Projection of the population OFF responses to the two stimuli onto the first two principal components of either stimuli. The projections of the vectors $\mathbf{v}^{(2;s=1)}$ and $-\mathbf{v}^{(1;s=1)}$ (resp. $\mathbf{v}^{(2;s=2)}$ and $-\mathbf{v}^{(1;s=2)}$) on the subspace defined by the first two principal components of stimulus 1 (resp. 2) are shown respectively as solid and dashed arrows. The subspaces defined by the vector pairs ($\mathbf{v}^{(1;s=1)}, \mathbf{v}^{(2;s=1)}$) and ($\mathbf{v}^{(1;s=2)}, \mathbf{v}^{(2;s=2)}$) are mutually orthogonal, so that the OFF responses to stimuli 1 and 2 evolve in orthogonal dimensions. (**D**) Firing rate of one example neuron in response to two distinct stimuli: in the recurrent network model the time course of the activity of one unit is not necessarily the same across stimuli. (**E**) Correlation between the initial state (i.e. the state at the end of stimulus presentation) and the state at the peak of the OFF responses, for five example stimuli. Error bars represent the standard deviation computed over 100 bootstrap subsamplings of 50% of the units in the population.

## Orthogonal OFF response trajectories across stimuli

The third feature observed in the data was that population responses evoked by different stimuli were orthogonal for many of the considered stimuli. Orthogonal trajectories for different stimuli can be reproduced in the model under the conditions that (i) initial patterns of activity corresponding to different stimuli are orthogonal, (ii) the connectivity matrix is partitioned into a set of orthogonal low-rank terms, each individually leading to transiently amplified dynamics, and (iii) each stimulus activates one of the low-rank terms. Altogether, for $P$ stimuli leading to orthogonal responses, the connectivity matrix can be partitioned in $P$ different groups of modes:

$$\mathbf{J} = \sum_{s=1}^{P} \mathbf{J}^{(s)}, \tag{6}$$

with

$$\mathbf{J}^{(s)} = \sum_{r=1}^{R_s} \mathbf{u}^{(r;s)} \mathbf{v}^{(r;s)T}, \tag{7}$$

where the vectors $\mathbf{v}^{(r;s)}$ have unit norm. The set of modes indexed by $s$ correspond to the $s$-th stimulus, so that the pattern of initial activity $\mathbf{r}_0^{(s)}$ evoked by stimulus $s$ overlaps only with those modes. Moreover, modes corresponding to different groups are orthogonal, and generate dynamics that span orthogonal subspaces (*Figure 2D*). Each term in *Equation (6)* can therefore be interpreted as an independent transient coding channel that can be determined from the OFF-response to stimulus $s$ alone.

We therefore examined whether the fitted connectivity $\mathbf{J}$ consisted of independent low-rank coding channels (*Equation (6)*), a constraint that would generate orthogonal responses to different stimuli as observed in the data. If $\mathbf{J}$ consisted of fully independent channels as postulated in *Equation (6)*, then it could be equivalently estimated by fitting the recurrent model to the OFF responses to each stimulus $s$ independently, leading to one matrix $\mathbf{J}^{(s)}$ for each stimulus. The hypothesis of fully independent channels then implies that (i) the connectivity vectors for the different connectivity matrices are mutually orthogonal, (ii) the full connectivity matrix can be reconstructed by summing the matrices $\mathbf{J}^{(s)}$ estimated for individual stimuli. To test these two predictions, we estimated the connectivity matrices $\mathbf{J}^{(s)}$ (the rank parameter for each stimulus was set to $R = 5$). We then computed the overlaps between the connectivity vectors corresponding to different pairs of stimuli (see Materials and methods, Section 'Analysis of the transient channels') and compared them to the overlaps between the subspaces spanned in response to the same pairs of stimuli (see *Figure 2D*). We found a close match between the two quantities (*Figure 7A,B*): pairs of stimuli with strong subspace overlaps corresponded to high connectivity overlaps, and pairs of stimuli with low subspace overlaps corresponded to low connectivity overlaps. This indicated that some of the stimuli, but not

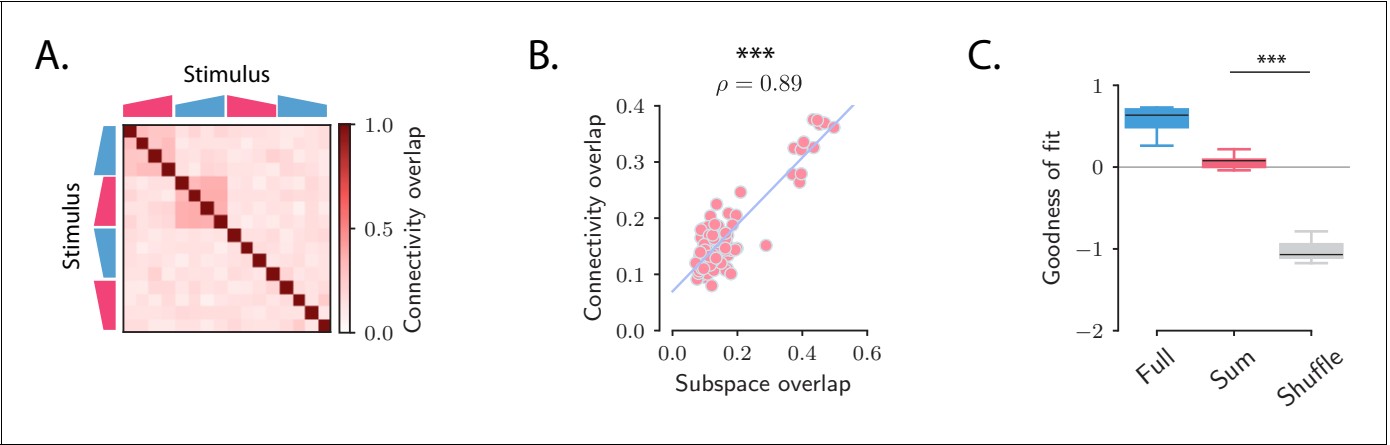

**Figure 7.** Structure of the transient channels across stimuli. (A) Overlaps between the transient channels corresponding to individual stimuli (*connectivity overlaps*), as quantified by the overlap between the right connectivity vectors $\mathbf{u}^{(r;s)}$ of the connectivities $\mathbf{J}^{(s)}$ fitted to each individual stimulus $s$ (see Materials and methods, Section 'Analysis of the transient channels'). The ridge and rank parameters, and the number of principal components have been respectively set to $\lambda = 5$, $R = 5$ and $\#PC = 100$ (the choice of $R = 5$ maximized the coefficient of determination between the connectivity overlaps, A, and the subspace overlaps, *Figure 2D*). (B) Scatterplot showing the relationship between the subspace overlaps (see *Figure 2D*) and connectivity overlaps (panel A) for each pair of stimuli. (C) Goodness of fit computed by predicting the population OFF responses using the connectivity $\mathbf{J}_{\text{Full}}$ (Full) or the connectivity given by the sum of the individual channels $\mathbf{J}_{\text{Sum}}$ (Sum). These values are compared with the goodness of fit obtained by shuffling the elements of the matrix $\mathbf{J}_{\text{Sum}}$ (Shuffle). Box-and-whisker plots show the distributions of the goodness of fit computed for each cross-validation partition (10-fold). For $\mathbf{J}_{\text{Full}}$ the ridge and rank parameters, and the number of principal components were set to $\lambda = 2$, $R = 70$ and $\#PC = 100$ (*Figure 5—figure supplement 1*).

all, corresponded to orthogonal coding channels. We further reconstructed the OFF responses using the matrix $\mathbf{J}_{\mathrm{Sum}} = \sum_s \mathbf{J}^{(s)}$. We then compared the goodness of fit computed using the matrices $\mathbf{J}_{\mathrm{Sum}}$, $\mathbf{J}_{\mathrm{Full}}$ and multiple controls where the elements of the matrix $\mathbf{J}_{\mathrm{Sum}}$ were randomly shuffled. While the fraction of variance explained when using the matrix $\mathbf{J}_{\mathrm{Sum}}$ was necessarily lower than the one computed using $\mathbf{J}_{\mathrm{Full}}$, the model with connectivity $\mathbf{J}_{\mathrm{Sum}}$ yielded values of the goodness of fit significantly higher than the ones obtained from the shuffles (*Figure 7C*). These results together indicated that the full matrix $\mathbf{J}_{\mathrm{Full}}$ can indeed be approximated by the sum of the low-rank channels represented by the $\mathbf{J}^{(s)}$.

## Amplification of single-trial fluctuations

So far we examined the activity obtained by averaging for each neuron the response to the same stimulus over multiple trials. We next turned to trial-to-trial variability of simultaneously recorded neurons and compared the structure of the variability in the data with the network and single-cell models. Specifically, we examined the variance of the activity along a direction $\mathbf{z}_0$ of state-space (*Figure 8A*; *Hennequin et al., 2012*), defined as:

$$\mathrm{var}(\mathbf{z}_0; t) = \langle \left( \mathbf{z}_0^T \mathbf{r}(t) - \mathbf{z}_0^T \langle \mathbf{r}(t) \rangle \right)^2 \rangle, \tag{8}$$

where $\langle \cdot \rangle$ denotes the averaging across trials. We compared the variance along the direction $\mathbf{r}_{\mathrm{ampl}}$ corresponding to the maximum amplification (distance from baseline) of the trial-averaged dynamics with variance along a random direction $\mathbf{r}_{\mathrm{rand}}$.

Inspecting the variability during the OFF-responses in the AC data revealed two prominent features: (i) fluctuations across trials are amplified along the same direction $\mathbf{r}_{\mathrm{ampl}}$ of state-space as trial-averaged dynamics, but not along random directions (*Figure 8B,C,D*); (ii) cancelling cross-correlations by shuffling trial labels independently for each cell strongly reduces the amplification of the variance (*Figure 8C,D*). This second feature demonstrates that the amplification of variance is due to noise-correlations across neurons that have been previously reported in the AC (*Rothschild et al., 2010*). Indeed the variance along a direction $\mathbf{z}_0$ at time $t$ can be expressed as

$$\mathrm{var}(\mathbf{z}_0; t) = \mathbf{z}_0^T \mathbf{C}(t) \mathbf{z}_0, \tag{9}$$

where $\mathbf{C}(t)$ represents the covariance matrix of the population activity at time $t$. Shuffling trial labels independently for each cell effectively cancels the off-diagonal elements of the covariance matrix and leaves only the diagonal elements that correspond to single-neuron variances. The comparison between simultaneous and shuffled data (*Figure 8C,D*) demonstrates that an increase in single-neuron variance is not sufficient to explain the total increase in variance along the amplified direction $\mathbf{r}_{\mathrm{ampl}}$.

Simulations of the fitted models and a mathematical analysis show that the network model reproduces both features of trial-to-trial variability observed in the data (*Figure 8E,F*), based on a minimal assumption that variability originates from independent noise in the initial condition at stimulus offset (see Materials and methods, Section 'Structure of single-trial responses in the network model'). Surprisingly, the single-cell model also reproduces the first feature, the amplification of fluctuations along the same direction of state-spate as trial-averaged dynamics, although in that model the different neurons are not coupled and noise-correlations are absent (*Figure 8G,H*). Instead, the single-cell model fails to capture the second feature, as it produces amplified variability in both simultaneous and shuffled activity. This demonstrates that the amplification of variability in the single-cell model is due to an amplification of single-cell variances, that is, the diagonal elements of covariance matrix (see Materials and methods, Section 'Structure of single-trial responses in the single-cell model'). Somewhat counter-intuitively, the disagreement between the single-cell model and the data is not directly due to the lack of noise-correlations in that model, but due to the fact that the single-cell model does not predict accurately the variance of single-neuron activity during the OFF-responses, and therefore fails to capture variance in shuffled activity.

In summary, the network model accounts better than the single-cell model for the structure of single-trial fluctuations present in the AC data.

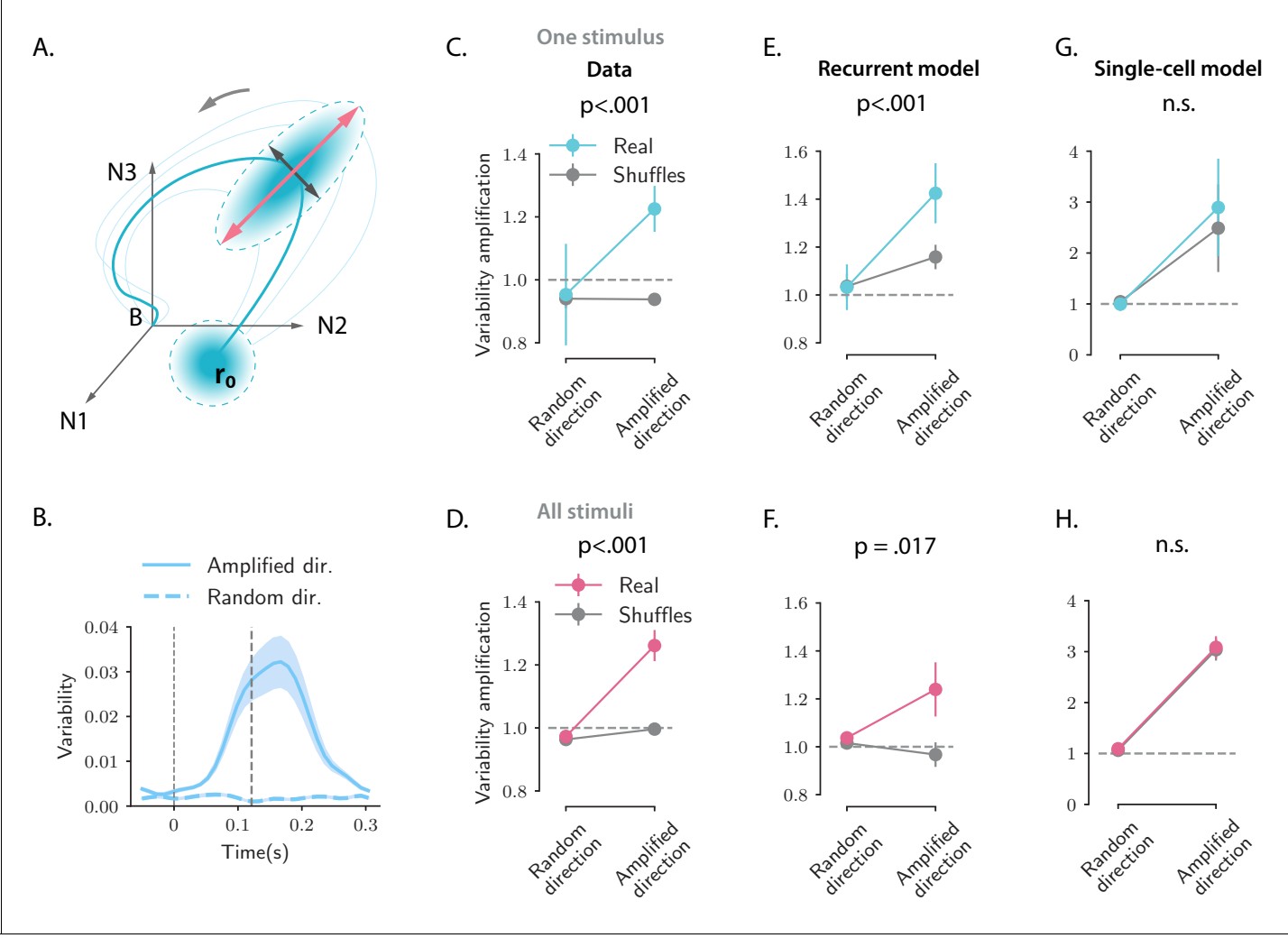

**Figure 8.** Structure of trial-to-trial variability generated by the network and single-cell models, compared to the auditory cortical data. (A) Cartoon illustrating the structure of single-trial population responses along different directions in the state-space. The thick and thin blue lines represent respectively the trial-averaged and single-trial responses. Single-trial responses start in the vicinity of the trial-averaged activity $\mathbf{r}_0$. Both the network and single-cell mechanisms dynamically shape the structure of the single-trial responses, here represented as graded blue ellipses. The red and black lines represent respectively the amplified and random directions considered in the analyses. (B) Time course of the variability computed along the amplified direction (solid trace) and along a random direction (dashed trace) for one example stimulus and one example session (287 simultaneously recorded neurons). In this and all the subsequent panels the amplified direction is defined as the population activity vector at the time when the norm of the trial-averaged activity of the pseudo-population pooled over sessions and animals reaches its maximum value (thick dashed line). Thin dashed lines denote stimulus offset. Shaded areas correspond to the standard deviation computed over 20 bootstrap subsamplings of 19 trials out of 20. (C, E, and G) Variability amplification (VA) computed for the amplified and random directions on the calcium activity data (panel C), on trajectories generated using the network model (panel E) and the single-cell model (panel G); (see Materials and methods, Section 'Single-trial analysis of population OFF responses'), for one example stimulus (same as in B). The network and single-cell models were first fitted to the trial-averaged responses to individual stimuli, independently across all recordings sessions (13 sessions, 180 ± 72 neurons). 100 single-trial responses were then generated by simulating the fitted models on 100 initial conditions drawn from a random distribution with mean $\mathbf{r}_0$ and covariance matrix equal to the covariance matrix computed from the single-trial initial states of the original responses (across neurons for the single-cell model, across PC dimensions for the recurrent model). Results did not change by drawing the initial conditions from a distribution with mean $\mathbf{r}_0$ and isotropic covariance matrix (i.e. proportional to the identity matrix, as assumed for the theoretical analysis in Materials and methods, Section 'Single-trial analysis of population OFF responses'). In the three panels, the values of VA were computed over 50 subsamplings of 90% of the cells (or PC dimensions for the recurrent model) and 50 shuffles. Error bars represent the standard deviation over multiple subsamplings, after averaging over all sessions and shuffles. Significance levels were evaluated by first computing the difference in VA between amplified and random directions ($\Delta$VA) and then computing the p-value on the difference between $\Delta$VA(Real) and $\Delta$VA(Shuffles) across subsamplings (two-sided independent t-test). For the network model, the VA is higher for the amplified direction than for a random direction, and this effect is significantly stronger for the real than for the shuffled responses. Instead, for the single-cell model the values of the VA computed on the real responses are not significantly different from the ones computed on the shuffled responses. (D, F,

*Figure 8 continued on next page*

*Figure 8 continued*

and H) Values of the VA computed as in panels **C, E, and G** pooled across all 16 stimuli. Error bars represent the standard error across stimuli. Significance levels were evaluated computing the p-value on the difference between $\Delta\mathrm{VA}(\mathrm{Real})$ and $\Delta\mathrm{VA}(\mathrm{Shuffles})$ across stimuli (two-sided Wilcoxon signed-rank test). The fits of the network and single-cell models of panels **E, G, F, and H** were generated using ridge regression ($\lambda = 0.5$ for both models).

## Discussion

Adopting a population approach, we showed that strong OFF responses observed in auditory cortical neurons form transiently amplified trajectories that encode individual stimuli within low-dimensional subspaces. A geometrical analysis revealed a clear structure in the relative orientation of these subspaces, where subspaces corresponding to different auditory stimuli were largely orthogonal to each other. We found that a simple, linear single-neuron model captures well the average response to individual stimuli, but not the structure across stimuli and across individual trials. In contrast, we showed that a simple, linear recurrent network model accounts for all the properties of the population OFF responses, notably their global structure across multiple stimuli and the fluctuations across single trials. Our results therefore argue for a potential role of network interactions in shaping OFF responses in AC. Ultimately, future work could combine both single-cell and network mechanisms in a network model with more complex intrinsic properties of individual neurons (*Beiran and Ostojic, 2019*; *Muscinelli et al., 2019*).

In this study, we focused specifically on the responses following stimulus offset. Strong transients during stimulus onset display similar transient coding properties (*Mazor and Laurent, 2005*) and could be generated by the same network mechanism as we propose for the OFF responses. However, during ON-transients, a number of additional mechanisms are likely to play a role, in particular single-cell adaptation, synaptic depression, or feed-forward inhibition. Indeed, recent work has shown that ON and OFF trajectories elicited by a stimulus are orthogonal to each other in the state-space (*Saha et al., 2017*), and this was also the case in our data set (*Figure 2—figure supplement 3*). Linear network models instead produce ON and OFF responses that are necessarily correlated and cannot account for the observed orthogonality of ON and OFF responses for a given stimulus. Distinct ON and OFF response dynamics could also result from intrinsic nonlinearities known to play a role in the AC (*Calhoun and Schreiner, 1998*; *Rotman et al., 2001*; *Sahani and Linden, 2003*; *Machens et al., 2004*; *Williamson et al., 2016*; *Deneux et al., 2016*).

A major assumption of both the single-cell and network models we considered is that the AC does not receive external inputs after the auditory stimulus is removed. Indeed, the observation that the structure of the dynamics across stimuli could be predicted by the structure of their initial states (*Figure 2D,E* and *Figure 2—figure supplement 4*) indicates that autonomous dynamics at least partly shape the OFF responses. The comparisons between data and models moreover show that autonomous dynamics are in principle sufficient to reproduce a number of features of OFF responses in the absence of any external drive. However, neurons in the AC do receive direct afferents from the medial geniculate body of the thalamus, and indirect input from upstream regions of the auditory pathway, where strong OFF responses have been observed. Thus, in principle, OFF responses observed in AC could be at least partly inherited from upstream auditory regions (*Kopp-Scheinpflug et al., 2018*). Disentangling the contributions of upstream inputs and recurrent dynamics is challenging if one has access only to auditory cortical activity (but see *Seely et al., 2016* for an interesting computational approach). Ultimately, the origin of OFF responses in AC needs to be addressed by comparing responses between related areas, an approach recently adopted in the context of motor cortical dynamics (*Lara et al., 2018*). A direct prediction of our model is that the inactivation of recurrent excitation in auditory cortical areas should weaken OFF responses (*Li et al., 2013*). However, recurrency in the auditory system is not present only within the cortex but also between different areas along the pathway (*Ito and Malmierca, 2018*; *Winer et al., 1998*; *Lee et al., 2011*). Therefore OFF responses could be generated at a higher level of recurrency and might not be abolished by inactivation of AC.

The dimensionality of the activity in large populations of neurons in the mammalian cortex is currently the subject of debate. A number of studies have found that neural activity explores low-dimensional subspaces during a variety of simple behavioral tasks (*Gao et al., 2017*). In contrast, a

recent study in the visual cortex has shown that the response of large populations of neurons to a large number of visual stimuli is instead high-dimensional (*Stringer et al., 2019*). Our results provide a potential way of reconciling these two sets of observations. We find that the population OFF responses evoked by individual auditory stimuli are typically low-dimensional, but lie in orthogonal spaces, so that the dimensionality of the responses increases when considering an increasing number of stimuli. Note that in contrast to *Stringer et al., 2019*, we focused here on the temporal dynamics of the population response. The dimensionality of these dynamics is in particular limited by the temporal resolution of the recordings (*Figure 2—figure supplement 1*).

The analyses we performed in this study were directly inspired by an analogy between OFF responses in the sensory areas and neural activity in the motor cortices (*Churchland and Shenoy, 2007*; *Churchland et al., 2010*). Starting at movement onset, single-neuron activity recorded in motor areas exhibits strongly transient and multiphasic firing lasting a few hundreds of milliseconds. Population-level dynamics alternate between at least two dimensions, shaping neural trajectories that appear to rotate in the state-space. These results have been interpreted as signatures of an underlying dynamical system implemented by recurrent network dynamics (*Churchland et al., 2012*; *Shenoy et al., 2011*), where the population state at movement onset provides the initial condition able to generate the transient dynamics used for movement generation. Computational models have explored this hypothesis (*Sussillo et al., 2015*; *Hennequin et al., 2014*; *Stroud et al., 2018*) and showed that the complex transient dynamics observed in motor cortex can be generated in network models with strong recurrent excitation balanced by fine-tuned inhibition (*Hennequin et al., 2014*). Surprisingly, fitting a linear recurrent network model to auditory cortical data, we found that the arrangement of the eigenvalues of the connectivity matrix was qualitatively similar to the spectrum of this class of networks (*Hennequin et al., 2014*), suggesting that a common mechanism might account for the responses observed in both areas.

The perceptual significance of OFF responses in the auditory pathway is still matter of ongoing research. Single-cell OFF responses observed in the auditory and visual pathways have been postulated to form the basis of duration selectivity (*Brand et al., 2000*; *Alluri et al., 2016*; *He, 2002*; *Aubie et al., 2009*; *Duysens et al., 1996*). In the auditory brainstem and cortex, OFF responses of single neurons exhibit tuning in the frequency-intensity domain, and their receptive field has been reported to be complementary to the receptive field of ON responses (*Henry, 1985*; *Scholl et al., 2010*). The complementary spatial organization of ON and OFF receptive fields may result from two distinct sets of synaptic inputs to cortical neurons (*Scholl et al., 2010*), and has been postulated to form the basis for higher-order stimulus features selectivity in single cells, such as frequency-modulated (FM) sounds (*Sollini et al., 2018*) and amplitude-modulated (AM) sounds (*Deneux et al., 2016*), both important features of natural sounds (*Sollini et al., 2018*; *Nelken et al., 1999*). Complementary tuning has also been observed between cortical ON and OFF responses to binaural localization cues, suggesting OFF responses may contribute to the encoding of sound-source location or motion (*Hartley et al., 2011*). At the population level, the proposed mechanism for OFF response generation may provide the basis for encoding complex sequences of sounds. Seminal work in the olfactory system has shown that sequences of odors evoked specific transient trajectories that depend on the history of the stimulation (*Broome et al., 2006*; *Buonomano and Maass, 2009*). Similarly, within our framework, different combinations of sounds could bring the activity at the end of stimulus offset to different regions of the state-space, setting the initial condition for the subsequent OFF responses. If the initial conditions corresponding to different sequences are associated with distinct transient coding channels, different sound sequences would evoke transient trajectories along distinct dimensions during the OFF responses, therefore supporting the robust encoding of complex sound sequences.

# Materials and methods

## Data analysis
### The data set
#### Neural recordings

Neural data was recorded and described in previous work (*Deneux et al., 2016*). We analyzed the activity of 2343 neurons in mouse AC recorded using two-photon calcium imaging while mice were

passively listening to different sounds. Each sound was presented 20 times. Data included recordings from three mice across 13 different sessions. Neural recordings in the three mice comprised respectively 1251, 636, and 456 neurons. Recordings in different sessions were performed at different depths, typically with a 50 μm difference (never less than 20 μm). Since the soma diameter is ~15 μm, this ensured that different cells were recorded in different sessions. We analyzed the trial-averaged activity of a pseudo-population of neurons built by pooling neural activity across all recording sessions and all animals. The raw calcium traces (imaging done at 31.5 frames per second) were smoothed using a Gaussian kernel with standard deviation $\sigma = 32$ ms. We then subtracted the baseline firing activity (i.e. the average neural activity before stimulus presentation) from the activity of each neuron, and used the baseline-subtracted neural activity for the analyses.

## The stimuli set

The stimuli consisted of a randomized presentation of 16 different sounds, 8 UP-ramping sounds, and 8 DOWN-ramping sounds. For each type, sounds had different frequency content (either 8 kHz or WN), different durations (1 or 2 s), and different combinations of onset and offset intensity levels (for UP-ramps either 50–85 dB or 60–85 dB, while for DOWN-ramps 85–50 dB or 85–60 dB). The descriptions of the stimuli are summarized in *Table 1*.

## Decoding analysis

To assess the accuracy of stimulus discrimination (8 kHz vs. WN sound) on single-trials, we trained and tested a linear discriminant classifier (*Bishop, 2006*) using cross-validation. For each trial, the pseudo-population activity vectors (built by pooling across sessions and animals) were built at each 50 ms time bin. We used leave-one-out cross-validation. At each time bin we used 19 out of 20 trials for the training set, and tested the trained decoder on the remaining trial. The classification accuracy was computed as the average fraction of correctly classified stimuli over all 20 cross-validation folds.

At each time $t$ the decoder for classification between stimuli $s_1$ and $s_2$ was trained using the trial-averaged pseudo-population vectors $\mathbf{c}_{1t}$ and $\mathbf{c}_{2t}$. These vectors defined the decoder $\mathbf{w}_t$ and the bias term $\mathbf{b}_t$ as:

$$\mathbf{w}_t = \mathbf{c}_{1t} - \mathbf{c}_{2t}, \qquad \mathbf{b}_t = \frac{\mathbf{c}_{1t} + \mathbf{c}_{2t}}{2} \tag{10}$$

A given population vector $\mathbf{x}$ was classified as either stimulus $s_1$ or stimulus $s_2$ according to the value of the function $y(\mathbf{x}) = \mathbf{w}_t^T \mathbf{x} - \mathbf{b}_t$:

$$\begin{cases} \text{if } y(\mathbf{x}) > 0 \text{ then } \mathbf{x} \text{ is classified as stimulus } s_1 \\ \text{if } y(\mathbf{x}) < 0 \text{ then } \mathbf{x} \text{ is classified as stimulus } s_2 \end{cases} \tag{11}$$

Random performance was evaluated by training and testing the classifier using cross-validation on surrogate data sets built by shuffling stimulus single-trial labels at each time bin. We performed the same analysis using more sophisticated decoder algorithms, that is, Linear Discriminant Analysis (LDA), Quadratic Discriminant Analysis (QDA), and C-SVM decoders. For none of these decoder algorithms was the cross-validated accuracy substantially better than the naive classifier reported in *Figure 1C* (and for the QDA algorithm the accuracy was substantially worse; not shown).

## Principal component analysis

To perform PCA on the population responses to $C$ stimuli $s_{i_1}, ..., s_{i_C}$ we considered the matrix $\mathbf{X} \in \mathbb{R}^{N \times TC}$, where $N$ is the number of neurons and $T$ is the number of time steps considered. The matrix $\mathbf{X}$ contained the population OFF responses to the stimuli $s_{i_1}, ..., s_{i_C}$, centered around the mean over times and stimuli. If we denote by $\lambda_i$ the $i$-th eigenvalue of the covariance matrix $\mathbf{X}\mathbf{X}^T/(TC)$, the percentage of variance explained by the $i$-th principal component is given by:

$$VAR(i) = \lambda_i / \sum_{j=1}^{N} \lambda_j \tag{12}$$

while the cumulative percentage of variance explained by the first $M$ principal components (*Figure 2A,C*) is given by:

$$CUMVAR(M) = \sum_{j=1}^{M} \lambda_j \Big/ \sum_{j=1}^{N} \lambda_j. \tag{13}$$

In **Figure 2A**, for each stimulus $s$ we consider the matrix $\mathbf{X}^{(s)} \in \mathbb{R}^{N \times T}$ containing the population OFF responses to stimulus $s$. We computed the cumulative variance explained as a function of the number of principal components $M$ and then averaged over all stimuli.

## Cross-validated PCA

We used cross-validation to estimate the variance of OFF responses to individual stimuli and across stimuli attributable to the stimulus-related component, discarding the component due to trial-to-trial variability (**Figure 2—figure supplement 1**). Specifically, we applied to the calcium activity data the method of cross-validated PCA (cvPCA) developed in **Stringer et al., 2019**. This method provides an unbiased estimate of the stimulus-related variance by computing the covariance between a training and a test set of responses (e.g. two different trials or two different sets of trials) to an identical collection of stimuli. Let $\mathbf{X}^{(train)}$ and $\mathbf{X}^{(test)}$ be the $N \times TC$ matrices containing the mean-centered responses to the same $C$ stimuli. We consider the training and test responses to be two distinct trials. We first perform ordinary PCA on the training responses $\mathbf{X}^{(train)}$ and find the principal component $\mathbf{u}_i^{(train)}$ ($i = 1, ..., N$). We then evaluate the cross-validated PCA spectrum $\{\lambda_i\}$ as:

$$\lambda_i = \frac{1}{C} \mathbf{u}_i^{(\text{train})T} \mathbf{X}^{(\text{test})T} \mathbf{X}^{(\text{train})} \mathbf{u}_i^{(\text{train})}. \tag{14}$$

We repeat the procedure for all pairs of trials $(i,j)$ with $i \neq j$ and average the result over pairs. The cross-validated cumulative variance is finally computed as in **Equation (13)**.

## Correlations between OFF response subspaces

To quantify the degree of correlation between pairs of OFF responses corresponding to two different stimuli, termed *subspace overlap*, we computed the cosine of the principal angle between the corresponding low-dimensional subspaces (**Figure 2D**, **Figure 3I**). In general, the principal angle $\theta_P$ between two subspaces $U$ and $V$ corresponds to the largest angle between any two pairs of vectors in $U$ and $V$ respectively, and it is defined by **Bjorck and Golub, 1973**; **Knyazev and Argentati, 2002**:

$$\cos\theta_P = \max_{\mathbf{u} \in U, \mathbf{v} \in V} \mathbf{u}^T \mathbf{v} \tag{15}$$

To compute the correlations between the OFF responses to stimuli $s_1$ and $s_2$ we first identified the first $K = 5$ principal components of the response to stimulus $s_1$ and organized them in a $N \times K$ matrix $\mathbf{Q}(s_1)$. We repeated this for stimulus $s_2$, which yielded a matrix $\mathbf{Q}(s_2)$. Therefore the columns of $\mathbf{Q}(s_1)$ and $\mathbf{Q}(s_2)$ define the two subspaces on which the responses to stimuli $s_1$ and $s_2$ live. The cosine of the principal angle between these two subspaces is given by **Bjorck and Golub, 1973**; **Knyazev and Argentati, 2002**:

$$\cos\theta_P(s_1, s_2) = \sigma_{\max}\big(\mathbf{Q}(s_1)^T \mathbf{Q}(s_2)\big), \tag{16}$$

where $\sigma_{\max}(\mathbf{A})$ denotes the largest singular value of a matrix $\mathbf{A}$. We note that this procedure directly relates to canonical correlation analysis (CCA; see **Hotelling, 1936**; **Uurtio et al., 2018**). In particular the first principal angle corresponds to the first canonical weight between the subspaces spanned by the columns of $\mathbf{Q}(s_1)$ and $\mathbf{Q}(s_2)$ (**Golub and Zha, 1992**; **Bjorck and Golub, 1973**).

## Controls for subspace overlaps and initial state-peak correlations

In this section we describe the controls that we used to evaluate the statistical significance of the measures of orthogonality between subspaces spanned by neural activity during OFF responses to different stimuli (**Figure 2D**, **Figure 3I**), and between initial and peak activity vectors for a single stimulus (**Figure 5D**). We assessed the significance of the orthogonality between OFF response subspaces across stimuli using two separate controls, which aim at testing for two distinct null hypotheses (**Figure 2D**, **Figure 3I**, **Figure 2—figure supplement 2**). The first hypothesis is that small

subspace overlaps (i.e. low correlations) between OFF responses to different stimuli may be due to the high number of dimensions of the state-space in which they are embedded. To test for this hypothesis we compared the subspace overlap computed on the trial-averaged activity with the subspace overlaps computed on the trial-averaged activity where the stimulus labels had been shuffled across trials for each pair of stimuli. We shuffled stimulus labels multiple times, resulting in one value of the subspace overlap for each shuffle. For each pair of stimuli, significance levels were then computed as the fraction of shuffles for which the subspace overlap was lower than the subspace overlap for the real data (lower tail test; *Figure 2—figure supplement 2A*).

Alternatively, small subspace overlaps could be an artifact of the trial-to-trial variability present in the calcium activity data. In fact, for a single stimulus, maximum values of the correlation between two different trials were of the order of 0.2 (*Deneux et al., 2016*). To test for the possibility that small subspace overlaps may be due to trial-to-trial variability, for each pair of stimuli we computed the values of the subspace overlaps by computing the trial-averaged activity on only half of the trials (10 trials), subsampling the set of 10 trials multiple times for each stimulus. This yielded a set of values $\cos\theta_{\mathrm{real}}(s_1, s_2, n)$, where $s_1$ and $s_2$ are the specific stimuli considered and $n = 1, ..., N_{\mathrm{shuffle}}$. We then computed the subspace overlaps between the trial-averaged responses to the same stimulus, but averaged over two different sets of 10 trials each, over multiple permutations of the trials, resulting in a set of values $\cos\theta_{\mathrm{shuffle}}(s, n)$, where $s \in \{s_1, s_2\}$ is the specific stimulus considered and $n = 1, ..., N_{\mathrm{shuffle}}$. For each pair of stimuli $s_1$ and $s_2$, significance levels were computed using two-tailed t-test and taking the maximum between the p-values given by $p(\cos\theta_{\mathrm{real}}(s_1, s_2, :), \cos\theta_{\mathrm{shuffle}}(s_1, :))$ and $p(\cos\theta_{\mathrm{real}}(s_1, s_2, :), \cos\theta_{\mathrm{shuffle}}(s_2, :))$ for those stimulus pairs for which $\langle\cos\theta_{\mathrm{real}}(s_1, s_2, :)\rangle_n < \langle\cos\theta_{\mathrm{shuffled}}(s_1, :)\rangle_n$ and $\langle\cos\theta_{\mathrm{real}}(s_1, s_2, :)\rangle_n < \langle\cos\theta_{\mathrm{shuffled}}(s_2, :)\rangle_n$, where the bar symbol indicated the mean over shuffles (*Figure 2—figure supplement 2B*).

The same null hypotheses have been used to test the significance of the orthogonality between initial state and peak state for individual stimuli (*Figure 5D*, *Figure 5—figure supplement 3*). A procedure analogous to the one outlined above was employed. Here, instead of the subspace overlaps, the quantity of interest is the correlation (scalar product) between the initial and peak state. For the first control shuffled data are obtained by shuffling the labels 'initial state' and 'peak state' across trials. Significance levels were evaluated as outlined above for the first control (*Figure 5—figure supplement 3A*). To test for the second hypothesis, we computed correlations between activity vectors defined at the same time point, but averaged over two different sets of 10 trials each. For each trial permutations we computed these correlations for all time points belonging to the OFF response (35 time points) and average over time points. Significance level was then assessed as outlined above for the second control (*Figure 5—figure supplement 3B*).

## Single-cell model for OFF response generation

In the next section we describe the procedure used to fit the single-cell model to the auditory cortical OFF responses. We consider the single-cell model given by *Equation (1)*, where $r_{0,i}^{(s)} L_i(0)$ represents the initial state of the response of unit $i$ to stimulus $s$. Without loss of generality we assume that the dynamic range of the temporal filters, defined as $|\max_t L_i(t) - \min_t L_i(t)|$, is equal to unity, so that $r_{0,i}^{(s)}$ represents the firing rate range of neuron $i$ for stimulus $s$. If that was not true, i.e. if the single-neuron responses were given by $r_i^{(s)}(t) = r_{0,i}^{(s)} K_i(t)$ with $\alpha_i = |\max_t K_i(t) - \min_t K_i(t)| \neq 1$ we could always write the responses as $r_i^{(s)}(t) = \tilde{r}_{0,i}^{(s)} \tilde{L}_i(t)$, where $\tilde{r}_{0,i}^{(s)} = r_{0,i}^{(s)} \alpha_i$ and $\tilde{L}_i(t) = K_i(t)/\alpha_i$, formally equivalent to *Equation (1)*.

## Fitting the single-cell model

To fit the single-cell OFF responses $r_i^{(s)}(t)$, we expressed the single-cell responses on a set of basis functions (*Pillow et al., 2008*).

$$r_i^{(s)}(t) = \sum_{j=1}^{N_{\mathrm{basis}}} a_{ij}^{(s)} f_j(t),\tag{17}$$

where the shape and the number of basis function $N_{\mathrm{basis}}$ are predetermined. We choose the functions $f_i(t)$ to be Gaussian functions centered around a value $\bar{t}_i$ and with a given width $w_i$, that is,

$f_i(t) = \exp(-(t - \bar{t}_i)^2 / 2w_i^2)$. The problem then consists in finding the coefficients $a_{ij}^{(s)}$ that best approximate *Equation (17)*. By dividing the left- and right-hand side of *Equation (17)* by the firing rate range $r_{0,i}^{(s)}$ we obtain:

$$\frac{r_i^{(s)}(t)}{r_{0,i}^{(s)}} = \sum_{j=1}^{N_{\text{basis}}} b_{ij}^{(s)} f_j(t), \quad b_{ij}^{(s)} = a_{ij}^{(s)} / r_{0,i}^{(s)}. \tag{18}$$

In general the coefficients $b_{ij}^{(s)}$ could be fitted independently for each stimulus. However, the single-cell model assumes that the coefficients $b_{ij}$ do not change across stimuli (see *Equation (1)*). Therefore, to find the stimulus-independent coefficients $b_{ij}$ that best approximate *Equation (18)* across a given set of stimuli, we minimize the mean squared error given by:

$$\text{MSE} = \sum_{i,s} \int \mathrm{d}t \left( \frac{r_i^{(s)}(t)}{r_{0,i}^{(s)}} - \sum_{j=1}^{N_{\text{basis}}} b_{ij} f_j(t) \right)^2 \tag{19}$$

The minimization of the mean squared error can be solved using linear regression techniques. Suppose we want to fit the population responses to $C$ different stimuli simultaneously. Let $\mathbf{R}^{(C)}$ be the matrix of size $N \times TC$ obtained by concatenating the $N \times T$ matrices $(\mathbf{r}_{\text{norm}}^{(s)})_{it} = r_i^{(s)}(t)/r_{0,i}^{(s)}$ ($i = 1,...,N$, $t = 1,...,T$, $s = 1,...,C$) corresponding to the normalized responses to the $C$ stimuli. Let $\mathbf{F}^{(C)}$ be the $N_{\text{basis}} \times TC$ matrix obtained by concatenating $C$ times the $N_{\text{basis}} \times T$ matrix $(\mathbf{f})_{it} = f_i(t)$. Let $\mathbf{B}$ be the $N \times N_{\text{basis}}$ matrix given by $\mathbf{B}_{ij} = b_{ij}$. Then *Equation (18)* can be written as:

$$\mathbf{R}^{(C)} = \mathbf{B}\mathbf{F}^{(C)}, \tag{20}$$

which can be solved using linear regression.

In order to avoid normalizing by very small values, we fit only the most responding neurons in the population, as quantified by their firing rate range. We set $w_i = w = 35\,ms$ for all $i = 1,...,N_{\text{basis}}$, and we set the number of basis functions to $N_{\text{basis}} = 10$, corresponding to the minimal number of basis functions sufficient to reach the highest cross-validated goodness of fit.

## The network model

We consider a recurrent network model of $N$ randomly coupled linear rate units. Each unit $i$ is described by the time-dependent variable $r_i(t)$, which represents the difference between the firing rate of neuron $i$ at time $t$ and its baseline firing level. The equation governing the temporal dynamics of the network reads:

$$\tau \dot{r}_i = -r_i + \sum_{j=1}^{N} J_{ij} r_j, \tag{21}$$

where $\tau$ represents the time constant (fixed to unity), and $J_{ij}$ is the effective synaptic strength from neuron $j$ to neuron $i$. The system has only one fixed point corresponding to $r_i = 0$ for all $i$. To have stable dynamics, we require that the real part of the eigenvalues of the connectivity matrix $\mathbf{J}$ is smaller than unity, that is, $(\Re\lambda(\mathbf{J}))_{\max} < 1$. We represent each stimulus as the state of the system reached at the end of stimulus presentation, which we denote by the vector $\mathbf{r}_0$. We moreover assume that during the OFF response the network receives no external input. The OFF response to the stimulus associated with the state $\mathbf{r}_0$ therefore corresponds to the network dynamics starting from the initial condition $\mathbf{r}_0$.

## Fitting the network model

In this section we describe the procedure we used for fitting a linear recurrent network model (*Equation (21)*) to the auditory cortical data using ridge and reduced-rank ridge regression. To fit the network model to the responses to $C$ stimuli, we first concatenate the $C$ matrices $\mathbf{X}^{(s)} \in \mathbb{R}^{T \times D}$ (where $\mathbf{X}_{t,i}^{(s)} = r_i^{(s)}(t)$, with $s = 1,...,C$ and $i = 1,...,D$) containing the neural responses to stimulus $s$ across $T$ timesteps and $D$ neurons (or PC dimensions) along the first dimension, obtaining a matrix

$\mathbf{X} \in \mathbb{R}^{CT \times D}$ contains the neural activity across $C$ stimuli, $T$ time steps, and $D$ neurons or PC dimensions. We then fit the network model $\dot{\mathbf{X}} = \mathbf{X}(\mathbf{J} - \mathbf{I})$ by first computing from the data the velocity of the trajectory as

$$\dot{\mathbf{X}} = \frac{\mathbf{X}(t_{i+1}) - \mathbf{X}(t_i)}{t_{i+1} - t_i}. \tag{22}$$

Both ridge and reduced-rank ridge regression aim at minimizing the mean squared error $||\dot{\mathbf{X}} - \mathbf{X}(\mathbf{J} - \mathbf{I})||^2$ subject to different constraints, which are described below. Since ridge regression involves computationally expensive matrix inversion, we first reduced the dimensionality of the original data set by using PCA. Unless otherwise stated, we kept a number of PC components equal to K = 100 for the fit of individual stimuli (*Figure 3G,H*, *Figure 5A–C*, *Figure 8*, *Figure 7*) and for the fit of all stimuli together (*Figure 3I*, *Figure 4*, *Figure 7C*, *Figure 3—figure supplement 2*, *Figure 3—figure supplement 3*, *Figure 5—figure supplement 1*; K = 100 principal components sufficed to explain more than 90% of the variance of the responses to all stimuli together). As a result, the data matrix $\mathbf{X} \in \mathbb{R}^{TC \times K}$ contains the activity across $T$ time bins and across all $K$ dimensions for a number $C$ of stimuli.

## Ridge regression

Ridge regression aims at determining the connectivity matrix $\mathbf{J}$ that minimizes the mean squared error $||\dot{\mathbf{X}} - \mathbf{X}(\mathbf{J} - \mathbf{I})||^2$ with a penalty for large entries of the matrix $\mathbf{J}$, so that the cost function to minimize is given by $||\dot{\mathbf{X}} - \mathbf{X}(\mathbf{J} - \mathbf{I})||^2 + \lambda||(\mathbf{J} - \mathbf{I})||^2$, where $\lambda$ is a hyperparameter of the model. We can write the ridge regression optimization problem as:

$$(\mathbf{J} - \mathbf{I})_{\lambda}^* = \underset{\mathbf{J} - \mathbf{I}}{\mathrm{argmin}} \, ||\dot{\mathbf{X}}_{\lambda} - \mathbf{X}_{\lambda}(\mathbf{J} - \mathbf{I})||^2 \tag{23}$$

where we defined $\dot{\mathbf{X}}_{\lambda} = (\dot{\mathbf{X}}, \mathbf{0})$ and $\mathbf{X}_{\lambda} = (\mathbf{X}, \sqrt{\lambda}\mathbf{I})$. The solution to *Equation (23)* is given by:

$$\mathbf{J}_{\lambda}^* = \mathbf{I} + (\mathbf{X}_{\lambda}^T \mathbf{X}_{\lambda})^{-1} \mathbf{X}_{\lambda}^T \dot{\mathbf{X}}_{\lambda}. \tag{24}$$

This procedure was used for fits shown in *Figure 3*, *Figure 3—figure supplement 1*, *Figure 3—figure supplement 3*, and *Figure 4*.

## Reduced-rank ridge regression

Reduced-rank regression aims at minimizing the mean squared error $||\dot{\mathbf{X}} - \mathbf{X}(\mathbf{J} - \mathbf{I})||^2$ under a rank constraint on the matrix $\mathbf{J}$, that is, $\mathrm{rank}\,\mathbf{J} \leq R$, where $R$ is a hyperparameter of the model (*Izenman, 1975*; *Davies and Tso, 1982*). Here we combined reduced-rank and ridge regression in the same framework. The reduced-rank ridge regression optimization problem with hyperparameters $R$ and $\lambda$ can be therefore written as (*Mukherjee et al., 2015*):

$$(\mathbf{J} - \mathbf{I})_{r,\lambda}^* = \underset{\mathrm{rank}\,\mathbf{J} \leq R}{\mathrm{argmin}} \, ||\dot{\mathbf{X}}_{\lambda} - \mathbf{X}_{\lambda}(\mathbf{J} - \mathbf{I})||^2. \tag{25}$$

To solve *Equation (25)* we consider the solution to the ridge regression problem given by *Equation (24)*. If the matrix $\mathbf{X}_{\lambda}\mathbf{J}_{\lambda}^*$ has SVD given by $\mathbf{X}_{\lambda}\mathbf{J}_{\lambda}^* = \mathbf{U}\Sigma\mathbf{V}^T$, then it can be shown that the solution to the reduced-rank ridge regression problem given by *Equation (25)* can be written as:

$$\mathbf{J}_{r,\lambda}^* = \mathbf{J}_{\lambda}^* \sum_{i=1}^{R} \mathbf{V}_i \mathbf{V}_i^T \tag{26}$$

We note that each term in the sum of *Equation (26)* has unit rank, so that the resulting matrix $\mathbf{J}_{R,\lambda}^*$ has rank equal to or less than $R$.

This procedure was used for fits shown in *Figure 5*, *Figure 7*, *Figure 8*, *Figure 3—figure supplement 2*, *Figure 5—figure supplement 1*, and *Figure 5—figure supplement 2*.

## Selection of hyperparameters for ridge regression

To select the value of the hyperparameter $\lambda$, we fitted the network model to the data and computed the goodness of fit as the coefficient of determination $R^2(\lambda)$ using cross-validation. We then selected the value of $\lambda$ which maximized the goodness of fit.

## Selection of hyperparameters for reduced-rank ridge regression

To select the values of the hyperparameters $\lambda$ and $R$, we fitted the network model to the data with hyperparameters $(\lambda, R)$ and computed the goodness of fit as the coefficient of determination $R^2(\lambda, R)$ using cross-validation. We repeated the process for a range of values of $(\lambda, R)$. We observed that, independent of the value of $\lambda$, the goodness of fit as a function of the rank $R$ saturates at a particular value of the rank $R^*$, but does not exhibit a clear maximum. We took the value $R^*$ as the minimal rank hyperparameter, while we defined the best ridge parameter as $\lambda^* = \mathrm{argmax}_\lambda R^2(\lambda, R^*)$ (*Figure 5—figure supplement 1*).

For both ridge and reduced-rank ridge regression, we used $K$-fold cross-validation, with $K = 10$. When fitting multiple stimuli at once, for each stimulus we partitioned the temporal activity into $K$ chunks, resulting in a total of $KC$ chunks. At the $i$-th iteration of the cross-validation procedure, we leave out the $i$-th partition for each stimulus to construct the training set (consisting of $(K-1)C$ chunks) and test the trained model on the remaining $C$ folds.

In *Figure 4A,B*, the temporal window considered for the fit was extended from 350 ms to 600 ms to include the decay to baseline of the OFF responses, thus obtaining stable eigenvalues. The extension of the temporal window was possible only for the 1 s long stimuli ($n = 8$), since the length of the temporal window following stimulus offset for the 2 s stimuli was limited to approximately 380 ms by the length of the neural recordings.

## Control data sets

To evaluate whether the fitted network model captured nontrivial collective dynamics of auditory cortical OFF responses, we followed the approach introduced in *Elsayed and Cunningham, 2017*. We first computed the goodness of fit (as quantified by the coefficient of determination $R^2$), then repeated the model fitting on control data sets which kept no additional structure than the one defined by correlations across time, neurons and stimuli of the original neural responses. We found that the goodness of fit obtained from fitting the model to the original data set was significantly higher than the one obtained from fitting the control data sets (*Figure 3—figure supplement 2*, *Figure 3—figure supplement 3*), confirming that the recurrent network model captured nontrivial collective dynamics in the data beyond the correlations across time, neurons and stimuli.

We generated the control data sets using a recent method based on a maximum entropy model (*Savin and Tkačik, 2017*) and described in *Elsayed and Cunningham, 2017*. This method, termed Tensor Maximum Entropy, allowed us to build surrogate data sets that are maximally random (entropy maximization), but constrained in a way that their marginal means and covariances are equal to the means and covariances of the original data set.

## Marginal means and covariances

Let the temporal activity along all $K$ dimensions for all $C$ stimuli be organized in a tensor $\mathbf{Z} \in \mathbb{R}^{T \times K \times C}$. The mean tensor $\mathbf{M}$ is defined as the tensor that makes all the marginal means of $\mathbf{Z}$ vanish. Specifically, if $\bar{\mathbf{Z}} = \mathbf{Z} - \mathbf{M}$, the tensor $\mathbf{M}$ is such that:

$$\sum_{k=1}^{K}\sum_{c=1}^{C} \bar{\mathbf{Z}}_{tkc} = 0, \quad \sum_{t=1}^{T}\sum_{c=1}^{C} \bar{\mathbf{Z}}_{tkc} = 0, \quad \sum_{t=1}^{T}\sum_{k=1}^{K} \bar{\mathbf{Z}}_{tkc} = 0 \tag{27}$$

The marginal covariances of the tensor $\bar{\mathbf{Z}}$ across times, neural dimensions, and stimuli are therefore defined as:

$$\begin{cases} \Sigma_{ij}^T = \sum_{k=1}^{K} \sum_{c=1}^{C} \bar{\mathbf{Z}}_{ikc} \bar{\mathbf{Z}}_{jkc} \\ \Sigma_{ij}^K = \sum_{t=1}^{T} \sum_{c=1}^{C} \bar{\mathbf{Z}}_{tic} \bar{\mathbf{Z}}_{tjc} \\ \Sigma_{ij}^C = \sum_{t=1}^{T} \sum_{k=1}^{K} \bar{\mathbf{Z}}_{tki} \bar{\mathbf{Z}}_{tkj} \end{cases} \qquad (28)$$

## Tensor maximum entropy method

The method generates the desired number of surrogate data sets $\mathbf{S}^{(i)} \in \mathbb{R}^{T \times K \times C}$. Each of these surrogates is randomly drawn from a probability distribution that assumes a priori no structure apart from that expected from the marginal means and covariances of the original data. Let $\mu$, $\Lambda^T$, $\Lambda^K$, and $\Lambda^C$ be the marginal means and covariances of surrogate $\mathbf{S}$. The method computes the probability $P(\mathbf{S})$ over the surrogates that maximizes the entropy function

$$P(\mathbf{S}) = \operatorname*{argmax}_{y(\mathbf{S})} \left[ -\int_{\mathbf{S}} y(\mathbf{S}) \log y(\mathbf{S}) \mathrm{d}\mathbf{S} \right], \quad \text{with} \int_{\mathbf{S}} P(\mathbf{S}) \mathrm{d}\mathbf{S} = 1 \qquad (29)$$

subject to the constraints

$$\mathbb{E}_P[\mu] = \mathbf{M}, \quad \mathbb{E}_P[\Lambda^T] = \Sigma^T, \quad \mathbb{E}_P[\Lambda^K] = \Sigma^K, \quad \mathbb{E}_P[\Lambda^C] = \Sigma^C, \qquad (30)$$

where $\mathbb{E}_P[\cdot]$ denotes the expectation over the probability density $P$. We used three types of surrogate data sets, denoted as T, TK, and TKC. All the three types of surrogates obey to the first constraint in *Equation (30)* on the marginal means. In addition, surrogates of type T obey the constraint on the time covariances, surrogates of type TK on time and dimension covariance, while surrogates TKC obey all the three covariance constraints.

## Analysis of the transient channels

In this section, we provide details on the analysis of the structure of the transient channels obtained from fitting the network model to OFF responses to individual stimuli using reduce-rank regression (*Figure 7*, see Materials and methods, Section 'Fitting the network model'). Let $\mathbf{J}_{\mathrm{Full}}$ be the connectivity matrix resulting from fitting the network model to the responses to all stimuli at once, and $\mathbf{J}^{(s)}$ the connectivity obtained from fitting the model to the response to stimulus $s$. Before fitting both matrices $\mathbf{J}_{\mathrm{Full}}$ and $\mathbf{J}^{(s)}$, we reduced the dimensionality of the responses to the first 100 principal components. We set the value of the rank parameter to $R = 70$ and $R = 5$ respectively for $\mathbf{J}_{\mathrm{Full}}$ and $\mathbf{J}^{(s)}$ (*Figure 5*). Using the SVD, we can write the matrices $\mathbf{J}^{(s)}$ as:

$$\mathbf{J}^{(s)} = \hat{\mathbf{U}}^{(s)} \Sigma^{(s)} \mathbf{V}^{(s)T}, \qquad (31)$$

where $\hat{\mathbf{U}}^{(s)}$ and $\mathbf{V}^{(s)}$ contain respectively the $R_s$ right and left connectivity vectors as columns (see *Equation (6)*), while $\Sigma^{(s)}$ contains the norms of $\mathbf{U}^{(s)}$ on the diagonal (see *Equation (6)*). The transient dynamics elicited by stimulus $s$ has a strong component along the state-space dimensions specified by the right connectivity vectors $\mathbf{u}^{(r;s)}$ (columns of $\mathbf{U}^{(s)}$; see also *Equation (43)*). We therefore define the overlap between the transient channels (*connectivity overlaps*) corresponding to pairs of stimuli $s_1$ and $s_2$ (*Figure 7A*) as the principal angle between the subspaces defined by $\hat{\mathbf{U}}^{(s_1)}$ and $\hat{\mathbf{U}}^{(s_2)}$ (see *Equations (15 and 16)*). We show that the structure of the connectivity overlaps (*Figure 7A*) matched well with the structure of the subspace overlaps (*Figure 2D*) across pairs of stimuli in *Figure 7B*.

To test whether the connectivity fitted on all stimuli at once $\mathbf{J}_{\mathrm{Full}}$ consisted of the sum of low-rank transient coding channels, we defined the matrix $\mathbf{J}_{\mathrm{Sum}}$ as the sum of the individual transient channels for all stimuli (see *Equation 6*):

$$\mathbf{J}_{\mathrm{Sum}} = \hat{\mathbf{U}}^{(1)} \Sigma^{(1)} \mathbf{V}^{(1)T} + \hat{\mathbf{U}}^{(2)} \Sigma^{(2)} \mathbf{V}^{(2)T} + ... + \hat{\mathbf{U}}^{(P)} \Sigma^{(P)} \mathbf{V}^{(P)T} \qquad (32)$$

We then compared the cross-validated goodness of fit of the population OFF responses using the full matrix $\mathbf{J}_{\text{Full}}$ and the matrix $\mathbf{J}_{\text{Sum}}$ (**Figure 7C**).

In **Figure 7**, when fitting individual stimuli, the ridge and rank hyperparameters have been set respectively to $\lambda = 5$, $R = 5$, which correspond to the values that maximize the agreement (coefficient of determination) between the connectivity overlaps (**Figure 7A**) and the subspace overlaps (**Figure 2D**). For these values the coefficient of determination between the transient channel overlaps and the subspace overlaps is >0.7.

## Analysis of the network model

In this section we provide a detailed mathematical analysis of the network model given by **Equation (21)**

$$\tau \dot{r}_i = -r_i + \sum_{j=1}^{N} J_{ij} r_j.$$

Specifically, we derive the conditions on the network components (i.e. connectivity spectra, connectivity vectors, and initial conditions) to produce low-dimensional, transiently amplified OFF responses in networks with low-rank connectivity. We then focus on the specific case of rotational channels in the connectivity.

### Normal and non-normal connectivity matrices

In this section, we summarize the relationship between amplified OFF responses and non-normal connectivity matrices in linear recurrent networks previously reported in *Trefethen and Embree, 2005*; *Hennequin et al., 2012*; *Bondanelli and Ostojic, 2020*. To characterize the amplification of OFF responses we focus on the temporal dynamics of the distance from baseline, defined as the norm of the population activity vector $||\mathbf{r}(t)||$ (*Hennequin et al., 2014*). The network generates an amplified OFF response to the stimulus associated with the initial condition $\mathbf{r}_0$ when the value of $||\mathbf{r}(t)||$ transiently increases before decaying to its asymptotic value $||\mathbf{r}(t \to \infty)|| = 0$. Note that having a transiently increasing value of the distance from baseline implies that the OFF response $r_i(t)$ of at least one unit displays non-monotonic temporal dynamics. Importantly, the transient behavior of $||\mathbf{r}(t)||$ depends on the stimulus through $\mathbf{r}_0$, and on the properties of the connectivity matrix $\mathbf{J}$, in particular on the relationship between its eigenvectors (*Trefethen and Embree, 2005*).

Connectivity matrices for which the eigenvectors are orthogonal to each other are called *normal* matrices and they are formally defined as matrices $\mathbf{J}$ that satisfy $\mathbf{J}\mathbf{J}^T = \mathbf{J}^T\mathbf{J}$. In networks with normal connectivity, any stimulus $\mathbf{r}_0$ evokes an OFF response for which the distance from baseline decays monotonically to zero. Such networks thus cannot produce amplified OFF responses, as defined by a transiently increasing $||\mathbf{r}(t)||$. Note that any symmetric matrix is normal.

On the other hand, connectivity matrices for which some eigenvectors are not mutually orthogonal are called *non-normal* (*Trefethen and Embree, 2005*), and they consist of all connectivity $\mathbf{J}$ for which $\mathbf{J}\mathbf{J}^T \neq \mathbf{J}^T\mathbf{J}$. It is well known that non-normal networks can lead to transiently increasing values of $||\mathbf{r}(t)||$, therefore producing amplified OFF responses. However, the non-normality of the network connectivity $\mathbf{J}$ constitutes only a necessary, but not a sufficient condition for the generation of amplified responses.

### Sufficient condition for amplified OFF responses

In this section we identify the necessary and sufficient condition for the generation of amplified OFF responses in linear recurrent networks. To find the necessary and sufficient condition for amplified responses, we start by writing the differential equation for the dynamics of the distance from baseline as (*Neubert and Caswell, 1997*; *Bondanelli and Ostojic, 2020*):

$$\frac{1}{||\mathbf{r}||}\frac{\mathrm{d}||\mathbf{r}||}{\mathrm{d}t} = \frac{\mathbf{r}^T(\mathbf{J}_S - \mathbf{I})\mathbf{r}}{||\mathbf{r}||^2}, \qquad \mathbf{J}_S = \frac{\mathbf{J} + \mathbf{J}^T}{2}, \tag{33}$$

where $\mathbf{J}_S$ denotes the symmetric part of the connectivity $\mathbf{J}$. A linear recurrent network exhibits amplified responses when the rate of change of the distance from baseline, $\mathrm{d}||\mathbf{r}||/\mathrm{d}t$, takes positive values at time $t = 0$. The right-hand side of **Equation (33)** takes its largest value when the initial condition $\mathbf{r}_0$

is aligned with the eigenvector of $\mathbf{J}_S$ associated with the largest eigenvalue $\lambda_{\max}(\mathbf{J}_S)$. In this case, the rate of change of the distance from baseline at time $t = 0$ takes the value $\lambda_{\max}(\mathbf{J}_S) - 1$. From *Equation (33)* it can be shown that the necessary and sufficient condition for the generation of amplified responses in a recurrent networks with connectivity $\mathbf{J}$ is given by

$$\lambda_{\max}(\mathbf{J}_S) > 1. \tag{34}$$

This criterion defines two classes of networks based on the properties of the connectivity matrix: networks in which amplified responses cannot be evoked by any stimulus, and networks able to generate amplified responses to at least one stimulus.

### Low-rank networks

In the following section we examine OFF dynamics in networks with low-rank connectivity of the form given by *Equation (3)*:

$$\mathbf{J} = \mathbf{u}^{(1)}\mathbf{v}^{(1)T} + \mathbf{u}^{(2)}\mathbf{v}^{(2)T} + ... + \mathbf{u}^{(R)}\mathbf{v}^{(R)T}. \tag{35}$$

We first show that such connectivity directly constraints the network dynamics to a low-dimensional subspace. We then derive the conditions for the stability and amplification of OFF responses. Finally we apply these results to the specific case of low-rank rotational channels as in *Equation (5)*.

### Low-dimensional dynamics

Here we study the dynamics of the population activity vector for a unit-rank network ($R = 1$) and for a general network with rank-$R$ connectivity structure. We consider low-rank networks in which the initial state is set to $\mathbf{r}(0) = \mathbf{r}_0$ and no external input acts on the system at later times $t > 0$. By construction, the autonomous dynamics generated by low-rank networks are constrained by the rank $R$ of the connectivity matrix and are therefore low-dimensional when $R \ll N$.

We first illustrate the linear dynamics in the case of a unit-rank connectivity ($R = 1$), given by

$$\mathbf{J} = \mathbf{u}^{(1)}\mathbf{v}^{(1)T}, \tag{36}$$

with $\mathbf{v}^{(1)T}$ of unit norm. The vectors $\mathbf{u}^{(1)}$ and $\mathbf{v}^{(1)}$ are respectively the right and left eigenvectors of $\mathbf{J}$, and in the following analysis we refer to them respectively as the right and left connectivity vectors. In this case, the dynamics following the initial state $\mathbf{r}_0$ can be explicitly computed as the product between the time-dependent propagator of the dynamics, given by $\mathbf{P}_t = \exp(t(\mathbf{J} - \mathbf{I}))$, and the initial condition $\mathbf{r}_0$ (*Arnold, 1973*; *Bondanelli and Ostojic, 2020*):

$$\mathbf{r}(t) = \mathbf{P}_t \mathbf{r}_0 = \exp(t(\mathbf{u}^{(1)}\mathbf{v}^{(1)T} - \mathbf{I}))\mathbf{r}_0. \tag{37}$$

By expanding the exponential matrix $\exp(t(\mathbf{J} - \mathbf{I}))$ in power series, we can write the propagator for the unit-rank dynamics as:

$$
\begin{aligned}
\exp(t(\mathbf{u}^{(1)}\mathbf{v}^{(1)T} - \mathbf{I})) \quad &= e^{-t}\sum_{k=0}^{\infty}\frac{\left(t(\mathbf{u}^{(1)}\mathbf{v}^{(1)T})\right)^k}{k!} \\
&= e^{-t}\left[\mathbf{I} + \frac{\mathbf{u}^{(1)}\mathbf{v}^{(1)T}}{\lambda}\left(1 + \lambda t + \frac{1}{2}\lambda^2 t^2 + ... - 1\right)\right] \\
&= e^{-t}\left[\mathbf{I} + \frac{e^{\lambda t} - 1}{\lambda}\mathbf{u}^{(1)}\mathbf{v}^{(1)T}\right],
\end{aligned}
\tag{38}
$$

where $\lambda$ is the only nonzero eigenvalue of $\mathbf{J}$, given by

$$\lambda = ||\mathbf{u}^{(1)}||\cos\theta, \quad \cos\theta = \frac{\mathbf{u}^{(1)} \cdot \mathbf{v}^{(1)}}{||\mathbf{u}^{(1)}||}. \tag{39}$$

As a result, the full dynamics in the case of rank-1 connectivity structure can be written as:

$$\mathbf{r}(t) = e^{-t}\mathbf{r}_0 + e^{-t}\mathbf{u}^{(1)}a(t), \tag{40}$$

where

$$a(t) = (\mathbf{v}^{(1)T}\mathbf{r}_0)\,(e^{\lambda t} - 1)/\lambda. \tag{41}$$

Since $\mathbf{u}^{(1)}$ is the right eigenvector of $\mathbf{J}$ corresponding to $\lambda$, from *Equation (40)* we note that, when $\mathbf{r}_0$ is fully aligned with $\mathbf{u}^{(1)}$ the dynamics are one-dimensional and exhibit a monotonic decay along the same dimension. Instead, when the initial state is not fully aligned with $\mathbf{u}^{(1)}$, the dynamics are confined to the plane defined by $\mathbf{r}_0$ and $\mathbf{u}^{(1)}$. In this case, while the component of the dynamics along $\mathbf{r}_0$ decays exponentially as a function of time, the component along the direction of $\mathbf{u}^{(1)}$ increases initially in proportion to the value of its norm, $||\mathbf{u}^{(1)}||$ (since at time $t = 0$, $\mathrm{d}e^{-t}a(t)/\mathrm{d}t = ||\mathbf{u}^{(1)}||$). Eventually, the activity decays to its asymptotic value given by $\mathbf{r}(t \to \infty) = \mathbf{0}$. Therefore in a unit-rank network the dynamics draw a neural trajectory that explores at most two-dimensions in the state space.

These observations can be extended to the general case of rank-$R$ connectivity matrices. For simplicity we rewrite *Equation (3)* as

$$\mathbf{J} = \mathbf{U}\mathbf{V}^T, \tag{42}$$

where the matrices $\mathbf{U}$ and $\mathbf{V}$ contain respectively the $R$ right and left connectivity vectors as columns. Writing the connectivity matrix in this form is always possible by applying the SVD on $\mathbf{J}$. The SVD allows us to write the connectivity matrix as $\mathbf{J} = \hat{\mathbf{U}}\mathbf{S}\mathbf{V}^T$, where $\mathbf{U} = \hat{\mathbf{U}}\mathbf{S}$, and $\hat{\mathbf{U}}^T\hat{\mathbf{U}} = \mathbf{V}^T\mathbf{V} = \mathbf{I}$. In particular, this implies the norm of each left connectivity vector $\mathbf{v}^{(r)}$ is unity, while the right connectivity vectors $\mathbf{u}^{(r)}$ are not normalized.

Following steps similar to *Equation (38)*, the linear dynamics evoked by the initial state $\mathbf{r}_0$ can be written as

$$
\begin{aligned}
\mathbf{r}(t) = \exp(t(\mathbf{J} - \mathbf{I}))\mathbf{r}_0 \; &= e^{-t}\left(\mathbf{I} + \sum_{m=1}^{\infty}\frac{(\mathbf{U}\mathbf{V}^T)^m}{m!}\right)\mathbf{r}_0 \\
&= e^{-t}\mathbf{r}_0 + e^{-t}\mathbf{U}\,\mathbf{a}(t),
\end{aligned}
\tag{43}
$$

where we defined the $R$-dimensional column vector

$$\mathbf{a}(t) = \left(\mathbf{V}^T\mathbf{U}\right)^{-1}\left[\exp(t\mathbf{V}^T\mathbf{U}) - \mathbf{I}\right]\left(\mathbf{V}^T\mathbf{r}_0\right), \tag{44}$$

in analogy with *Equation (40)*. Therefore, in the case of rank-$R$ connectivity matrix, the dynamics evolve in a $(R+1)$-dimensional space determined by the initial state $\mathbf{r}_0$ and the $R$ right connectivity vectors $\mathbf{u}^{(r)}$ (columns of $\mathbf{U}$). *Equation (44)* shows that the dynamics of a rank-$R$ system are determined by the matrix of scalar products between left and right connectivity vectors, which we refer to as the *overlap matrix* (*Schuessler et al., 2020*; *Beiran et al., 2020*)

$$\mathbf{J}^{ov} = \mathbf{V}^T\mathbf{U}. \tag{45}$$

We conclude that low-rank connectivity matrices of the form given by *Equation (3)* with $R \ll N$ generate low-dimensional dynamics that explore at most $R+1$ dimensions.

## Conditions for stability and amplification of OFF responses in low-rank networks

In this paragraph we examine the conditions required to generate stable amplified dynamics in networks with low-rank connectivity $\mathbf{J}$ and initial state $\mathbf{r}_0$. Specifically, two sets of conditions need to be satisfied. The first set of conditions is directly derived by applying the general criterion for stable amplified dynamics given by *Equation (34)* to low-rank networks, which effectively constrains the connectivity $\mathbf{J}$ through the relative arrangement of the connectivity vectors $\mathbf{u}^{(r)}$ and $\mathbf{v}^{(r)}$ ($r = 1, ..., R$). When this criterion is satisfied, amplified dynamics can be generated only if the initial condition $\mathbf{r}_0$ is aligned with specific directions in the state-space. We thus examine a second set of conditions on the initial state $\mathbf{r}_0$ for which amplified trajectories can be evoked, and express these conditions in terms of relationship between $\mathbf{r}_0$ and the modes $\mathbf{u}^{(r)}$-$\mathbf{v}^{(r)}$. As before, without loss of generality, we assume that the norm of the vectors $\mathbf{v}^{(r)}$ is equal to one, while the norm of the vectors $\mathbf{u}^{(r)}$ can vary.

## Conditions on the modes $\mathbf{u}^{(r)}$-$\mathbf{v}^{(r)}$

We first consider a network with unit-rank connectivity $\mathbf{J} = \mathbf{u}^{(1)}\mathbf{v}^{(1)T}$, with $\mathbf{v}^{(1)}$ of unit norm. In unit-rank networks, the dynamics is stable only if the nonzero eigenvalue $\lambda$ is smaller than one. From *Equation (39)* this yields the stability condition

$$\cos\theta < \frac{1}{||\mathbf{u}^{(1)}||}. \tag{46}$$

The condition for the generation of amplified responses given by *Equation (34)* can be derived in terms of $\mathbf{u}^{(1)}$ and $\mathbf{v}^{(1)}$ by computing the eigenvalues of the symmetric part of the connectivity, $\mathbf{J}_S = (\mathbf{u}^{(1)}\mathbf{v}^{(1)T} + \mathbf{v}^{(1)}\mathbf{u}^{(1)T})/2$. The matrix $\mathbf{J}_S$ is of rank two, and has in general two nonzero eigenvalues, given by:

$$\lambda_{\mathrm{max,min}}(\mathbf{J}_S) = \frac{||\mathbf{u}^{(1)}||(\cos\theta \pm 1)}{2} \tag{47}$$

Therefore the condition for amplified OFF responses *Equation (34)* can be written in terms of the length of the vector $\mathbf{u}^{(1)}$ as

$$||\mathbf{u}^{(1)}|| > \frac{2}{\cos\theta + 1}. \tag{48}$$

Previous work has shown that amplification in a unit-rank network can take arbitrarily large values only if $0 \leq \cos\theta < 1/||\mathbf{u}^{(1)}||$ (*Bondanelli and Ostojic, 2020*). As a result, stable and unbounded amplified dynamics can be generated in a unit-rank network if and only if the norm of $\mathbf{u}^{(1)}$ is sufficiently large (*Equation (48)*) and the correlation between the connectivity vectors is positive and sufficiently small (*Equation (46)*).

The conditions for stability and amplification derived for a rank-1 network can be implicitly generalized to a rank-$R$ network model, using the fact that the nonzero eigenvalues $\lambda$ of the low-rank connectivity matrix $\mathbf{J} = \mathbf{U}\mathbf{V}^T$ (where $\mathbf{J}$ is a $N \times N$ matrix) are equal to the nonzero eigenvalues of its overlap matrix $\mathbf{J}^{ov} = \mathbf{V}^T\mathbf{U}$ (where $\mathbf{J}^{ov}$ is a $R \times R$ matrix) (*Nakatsukasa, 2019*):

$$\mathrm{eig}_{\lambda \neq 0}(\mathbf{J}) = \mathrm{eig}_{\lambda \neq 0}(\mathbf{J}^{ov}) \tag{49}$$

Therefore, the condition for stability can be implicitly written in terms of the nonzero eigenvalues $\lambda$ of $\mathbf{J}^{ov}$ as:

$$\begin{cases} \det(\mathbf{V}^T\mathbf{U} - \lambda\mathbf{I}) = 0 \\ \quad\quad \lambda < 1. \end{cases} \tag{50}$$

To derive the conditions for transient amplification, we need that the eigenvalues of the symmetric part of the connectivity be larger than unity, where the symmetric part $\mathbf{J}_S$ is a matrix of rank $2R$ given by:

$$\mathbf{J}_S = \frac{1}{2}(\mathbf{U}\mathbf{V}^T + \mathbf{V}\mathbf{U}^T) = \frac{1}{2}[\mathbf{U}, \mathbf{V}]\begin{bmatrix} \mathbf{V}^T \\ \mathbf{U}^T \end{bmatrix} \tag{51}$$

Using *Equation (49)* (*Nakatsukasa, 2019*), we can write the nonzero eigenvalues $\lambda_S$ of $\mathbf{J}_S$ as:

$$\mathrm{eig}_{\lambda_S \neq 0}(\mathbf{J}_S) = \frac{1}{2}\mathrm{eig}_{\lambda_S \neq 0}\begin{bmatrix} \mathbf{V}^T \\ \mathbf{U}^T \end{bmatrix}[\mathbf{U}, \mathbf{V}] = \frac{1}{2}\mathrm{eig}_{\lambda_S \neq 0}\begin{pmatrix} \mathbf{V}^T\mathbf{U} & \mathbf{I} \\ \mathbf{S}^2 & \mathbf{U}^T\mathbf{V} \end{pmatrix}, \tag{52}$$

where we used the fact that $\mathbf{V}^T\mathbf{V} = \mathbf{I}$, and defined $\mathbf{S}^2 = \mathrm{diag}(||\mathbf{u}^{(1)}||, ||\mathbf{u}^{(2)}||, ...)$. By definition, the eigenvalues of the symmetric part multiplied by 2, that is, $2\lambda_S$, should satisfy:

$$0 = \det(\mathbf{V}^T\mathbf{U} - 2\lambda_S\mathbf{I})\det\left(\mathbf{U}^T\mathbf{V} - 2\lambda_S\mathbf{I} - \mathbf{S}^2(\mathbf{U}^T\mathbf{V} - 2\lambda_S\mathbf{I})^{-1}\right), \tag{53}$$

where we used the expression of the determinant of a $2 \times 2$ block matrix (*Horn and Johnson, 2012*).

Since for two square matrices $\mathbf{A}$ and $\mathbf{B}$ we have that $\det(\mathbf{AB}) = \det(\mathbf{A})\det(\mathbf{B})$, $\det(\mathbf{A}^T) = \det(\mathbf{A})$, and $\det(\mathbf{A}^{-1}) = 1/\det(\mathbf{A})$, we can write:

$$
\begin{aligned}
0 &= \det\left(\mathbf{V}^T\mathbf{U} - 2\lambda_S\mathbf{I}\right)\det\left\{\left[(\mathbf{U}^T\mathbf{V} - 2\lambda_S\mathbf{I})^2 - \mathbf{S}^2\right](\mathbf{U}^T\mathbf{V} - 2\lambda_S\mathbf{I})^{-1}\right\} \\
&= \det\left(\mathbf{V}^T\mathbf{U} - 2\lambda_S\mathbf{I}\right)\det\left[(\mathbf{U}^T\mathbf{V} - 2\lambda_S\mathbf{I})(\mathbf{V}^T\mathbf{U} - 2\lambda_S\mathbf{I}) - \mathbf{S}^2\right]\det\left(\mathbf{V}^T\mathbf{U} - 2\lambda_S\mathbf{I}\right)^{-1} \\
&= \det\left[(\mathbf{U}^T\mathbf{V} - 2\lambda_S\mathbf{I})(\mathbf{V}^T\mathbf{U} - 2\lambda_S\mathbf{I}) - \mathbf{S}^2\right].
\end{aligned}
\tag{54}
$$

Thus, the condition for amplification becomes:

$$
\begin{cases}
\det\left[\mathbf{B}(\lambda_S)\mathbf{B}(\lambda_S)^T - \mathbf{S}^2\right] = 0, \quad \text{where} \quad \mathbf{B}(\lambda_S) = (\mathbf{U}^T\mathbf{V} - 2\lambda_S\mathbf{I}) \\
\lambda_S > 1
\end{cases}
\tag{55}
$$

Thus, in the general case of a rank-$R$ network, the condition for stable and amplified dynamics is given by *Equations (50) and (55)*.

## Dynamics of a low-rank rotational channel

In this section we describe the structure of the low-rank connectivity obtained by fitting the network model to the OFF responses to individual stimuli (*Figure 5B,C*) and analyze the resulting dynamics. When fitting a low-rank network model to the OFF responses to individual stimuli, we observed a specific structure in the pattern of correlations between right and left connectivity vectors $\mathbf{u}^{(i)}$ and $\mathbf{v}^{(j)}$ of the fitted connectivity $\mathbf{J}$. This pattern exhibited low values of the correlation for almost all pairs of connectivity vectors, except for pairs of left and right vectors coupled across nearby modes, e.g. $\mathbf{u}^{(1)}$-$\mathbf{v}^{(2)}$, $\mathbf{u}^{(2)}$-$\mathbf{v}^{(1)}$, $\mathbf{u}^{(3)}$-$\mathbf{v}^{(4)}$, $\mathbf{u}^{(4)}$-$\mathbf{v}^{(3)}$, and so forth. This structures gives rise to independent rank-2 channels grouping pairs of modes of the form $\mathbf{J}_1 = \mathbf{u}^{(1)}\mathbf{v}^{(1)T} + \mathbf{u}^{(2)}\mathbf{v}^{(2)T}$. Within a single channel, the values of the correlation was high, and opposite in sign, for different pairs of vectors, that is, $\hat{\mathbf{u}}^{(1)} \cdot \mathbf{v}^{(2)} \approx 1$ and $\hat{\mathbf{u}}^{(2)} \cdot \mathbf{v}^{(1)} \approx -1$. As a result, each of these channels can be cast in the form:

$$
\mathbf{J}_2 = \Delta_1\mathbf{v}^{(2)}\mathbf{v}^{(1)T} - \Delta_2\mathbf{v}^{(1)}\mathbf{v}^{(2)T},
\tag{56}
$$

where we set

$$
\begin{aligned}
\mathbf{u}^{(1)} &= \Delta_1\mathbf{v}^{(2)} \\
\mathbf{u}^{(2)} &= -\Delta_2\mathbf{v}^{(1)}
\end{aligned}
\tag{57}
$$

with $\Delta_1$, $\Delta_2$ two positive scalars, and the vectors $\mathbf{v}$'s are of unit norm. *Scheme 1* illustrates the structure of the rank-2 channel in terms of the left and right connectivity vectors, and the dynamics evoked by an initial condition along $\mathbf{v}^{(1)}$.

For a rank-2 connectivity as in *Equation (5)*, the 2x2 overlap matrix $\mathbf{J}^{ov}$ is therefore given by

$$
\mathbf{J}^{ov} = \mathbf{V}^T\mathbf{U} = \begin{pmatrix} 0 & -\Delta_2 \\ \Delta_1 & 0 \end{pmatrix} = \mathbf{V}^T\mathbf{J}_2\mathbf{V} \equiv \tilde{\mathbf{J}}_2. \tag{58}
$$

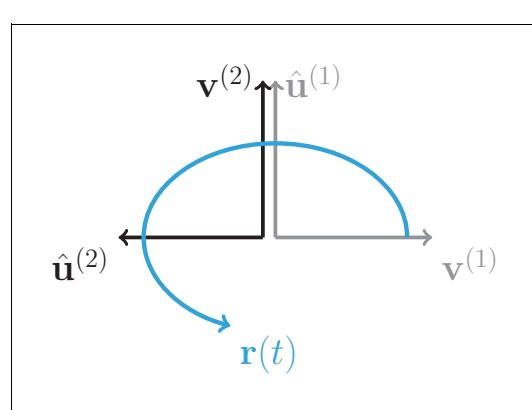

**Scheme 1.** Schematics of the rank-2 rotational channel.

Note that the overlap matrix also corresponds to the connectivity matrix $\mathbf{J}_2$ projected on the basis of the left connectivity vectors $[\mathbf{v}^{(1)}, \mathbf{v}^{(2)}]$ (third equality in *Equation (58)*).

To derive the necessary and sufficient conditions for the channel to exhibit stable and amplified dynamics in response to at least one initial condition $\mathbf{r}_0$, we need to compute the nonzero eigenvalues of the connectivity matrix and of its symmetric part (equal to the eigenvalues of $\tilde{\mathbf{J}}_2$ and of its symmetric part). The eigenvalues of $\tilde{\mathbf{J}}_2$ are purely imaginary for all values of $\Delta_1$ and $\Delta_2$ and are given by $\lambda_{1,2} = \pm i\sqrt{\Delta_1\Delta_2} \equiv \pm i\omega$. The dynamics are therefore always stable. Next, we

compute the eigenvalues of the symmetric part of the connectivity, which reads:

$$\tilde{\mathbf{J}}_{2,S} = \frac{1}{2}(\tilde{\mathbf{J}}_2 + \tilde{\mathbf{J}}_2^T) = \begin{pmatrix} 0 & (\Delta_1 - \Delta_2)/2 \\ (\Delta_1 - \Delta_2)/2 & 0 \end{pmatrix}. \tag{59}$$

The eigenvalues of $\tilde{\mathbf{J}}_{2,S}$ are $\pm|\Delta_2 - \Delta_1|/2$, so that the dynamics are amplified when

$$|\Delta_2 - \Delta_1| > 2 \tag{60}$$

We next derive the full dynamics of the rank-2 connectivity matrix $\mathbf{J}_2$ in *Equation (56)*. To this end, we use the general expression of the propagator for a rank-R network given by *Equations (43), (44)* in terms of the overlap matrix $\mathbf{J}^{ov}$ (*Equation (58)*). To compute the inverse and the exponential of the overlap matrix, we start by diagonalizing $\mathbf{J}^{ov}$. Its eigenvalues are given by $\lambda_{1,2} = \pm i\omega$, while the corresponding eigenvectors are specified in the columns of the matrix $\mathbf{E}$, where $\mathbf{E}$ and $\mathbf{E}^{-1}$ read:

$$\mathbf{E} = \frac{1}{C}\begin{pmatrix} 1 & 1 \\ -i\sqrt{\frac{\Delta_1}{\Delta_2}} & i\sqrt{\frac{\Delta_1}{\Delta_2}} \end{pmatrix}, \quad \mathbf{E}^{-1} = C\begin{pmatrix} 1 & i\sqrt{\frac{\Delta_2}{\Delta_1}} \\ 1 & -i\sqrt{\frac{\Delta_2}{\Delta_1}} \end{pmatrix}, \quad C = \sqrt{1 + \Delta_1/\Delta_2} \tag{61}$$

The inverse and exponential of the overlap matrix can be therefore computed as:

$$
\begin{aligned}
(\mathbf{V}^T\mathbf{U})^{-1} &= \mathbf{E}\begin{pmatrix} -\frac{i}{\omega} & 0 \\ 0 & \frac{i}{\omega} \end{pmatrix}\mathbf{E}^{-1} \\
\exp(\mathbf{V}^T\mathbf{U}) &= \mathbf{E}\begin{pmatrix} \exp(i\omega) & 0 \\ 0 & \exp(-i\omega) \end{pmatrix}\mathbf{E}^{-1},
\end{aligned}
\tag{62}
$$

so that we obtain the term $\mathbf{a}(t)$ in *Equation (44)* as:

$$\mathbf{a}(t) = \begin{pmatrix} \frac{1}{\omega}\sin\omega t & \frac{1}{\Delta_1}(\cos\omega t - 1) \\ -\frac{1}{\Delta_2}(\cos\omega t - 1) & \frac{1}{\omega}\sin\omega t \end{pmatrix}(\mathbf{V}^T\mathbf{r}_0). \tag{63}$$

From *Equations (43), (44), (57), and (63)*, we can write the expression for the dynamics of the transient channel $\mathbf{J}_2$ with initial condition given by $\mathbf{r}_0 = \alpha_1\mathbf{v}^{(1)} + \alpha_2\mathbf{v}^{(2)} + \beta\mathbf{z}^{(\perp)}$ (with $\mathbf{z}^{(\perp)}$ a vector orthogonal to both $\mathbf{v}^{(1)}$ and $\mathbf{v}^{(2)}$) as:

$$\mathbf{r}(t) = e^{-t}\beta\mathbf{z}^{(\perp)} + e^{-t}\left[\mathbf{v}^{(1)}\left(\alpha_1\cos\omega t - \sqrt{\frac{\Delta_2}{\Delta_1}}\alpha_2\sin\omega t\right) + \mathbf{v}^{(2)}\left(\alpha_2\cos\omega t + \sqrt{\frac{\Delta_1}{\Delta_2}}\alpha_1\sin\omega t\right)\right]. \tag{64}$$

Therefore, the squared distance from baseline of the trajectory reads:

$$
\begin{aligned}
||\mathbf{r}(t)||^2 = e^{-2t}\beta^2 \quad &+ e^{-2t}\left[\alpha_1^2\cos^2\omega t + \frac{\Delta_2}{\Delta_1}\alpha_2^2\sin^2\omega t - 2\alpha_1\alpha_2\frac{\Delta_2}{\Delta_1}\cos\omega t\sin\omega t\right. \\
&\left. \alpha_2^2\cos^2\omega t + \frac{\Delta_1}{\Delta_2}\alpha_1^2\sin^2\omega t + 2\alpha_1\alpha_2\frac{\Delta_1}{\Delta_2}\cos\omega t\sin\omega t\right].
\end{aligned}
\tag{65}
$$

Note that when $\Delta_2 = \Delta_1$, the connectivity matrix is purely anti-symmetric (and therefore normal) and cannot produce amplified dynamics. In fact, in this case, the evolution of the distance from baseline results in an exponential decay $||\mathbf{r}(t)|| = e^{-t}$. Amplification therefore requires that $\Delta_1 \neq \Delta_2$ and specifically that *Equation (60)* holds.

We observe that while an anti-symmetric matrix (therefore normal, and with purely imaginary eigenvalues) cannot produce amplified dynamics, nonetheless a matrix with purely imaginary eigenvalues is not necessarily normal, and can thus produce amplified dynamics. Qualitatively, this is reflected in the dynamics tracing elliptical rotations (as opposed to circular rotations in the case of an anti-symmetric matrix) which determine oscillations (and therefore rising and decaying phases) in the distance from baseline. This analysis can be extended to the case of a superposition of K rank-2 mutually orthogonal channels with connectivity matrix given by:

$$\mathbf{J}_{2K} = \sum_{k=1}^{K} \Delta_1^{(k)} \mathbf{v}^{(2;k)} \mathbf{v}^{(1;k)T} - \Delta_2^{(k)} \mathbf{v}^{(1;k)} \mathbf{v}^{(2;k)T}, \tag{66}$$

where each channel $k$ is defined by the connectivity vectors $\mathbf{v}^{(1;k)}$, $\mathbf{v}^{(2;k)}$ (*Figure 5B*), defining together a rank-2K rotational channel. If we write the initial condition along the connectivity vectors of the different K channels as

$$\mathbf{r}_0 = \sum_{k=1}^{K} \left( \alpha_1^{(k)} \mathbf{v}^{(1;k)} + \alpha_2^{(k)} \mathbf{v}^{(2;k)} \right) + \beta \mathbf{z}^{(\perp)} = \sum_{k=1}^{K} \mathbf{r}_0^{(k)} + \beta \mathbf{z}^{(\perp)} \tag{67}$$

with $\sum_{k=1}^{K} \left( \alpha_1^{(k)2} + \alpha_2^{(k)2} \right) + \beta^2 = 1$, we can write the resulting dynamics as the sum of the dynamics within each channel as:

$$\mathbf{r}_{2K}(t) = \sum_{k=1}^{K} \mathbf{r}(t; \mathbf{r}_0^{(k)}) + e^{-t} \beta \mathbf{z}^{(\perp)}, \tag{68}$$

where the terms $\mathbf{r}(t; \mathbf{r}_0^{(k)})$ have the same form of the second term of the right-hand side of *Equation (64)*.

## Conditions on the initial state $\mathbf{r}_0$

In this paragraph we consider the rotational channel examined in Section 'Dynamics of a low-rank rotational channel' and study the conditions on the relationship between the initial condition $\mathbf{r}_0$ and the connectivity vectors that ensure that the evoked dynamics be amplified. We first examine the case of a rank-2 rotational channel, and then generalize to a superposition of orthogonal rank-2 channels.

We consider the equation for the distance from baseline of the dynamics for a rank-2 channel (*Equation (65)*) and assume, without loss of generality, that $\Delta_2 > \Delta_1$. We observe that when $\Delta_2 > \Delta_1$, an initial condition along $\mathbf{v}^{(1)}$ does not evoke amplified dynamics. On the other hand, an initial condition along $\mathbf{v}^{(2)}$ can evoke amplified dynamics, depending on the values of $\Delta_1$ and $\Delta_2$. In fact, we have:

$$\begin{aligned} ||\mathbf{r}(t; \mathbf{r}_0 = \mathbf{v}^{(1)})||^2 &= e^{-2t} \left[ 1 + \left( \frac{\Delta_1}{\Delta_2} - 1 \right) \sin^2 \omega t \right] < e^{-2t} \; \forall t \\ ||\mathbf{r}(t; \mathbf{r}_0 = \mathbf{v}^{(2)})||^2 &= e^{-2t} \left[ 1 + \left( \frac{\Delta_2}{\Delta_1} - 1 \right) \sin^2 \omega t \right] \end{aligned} \tag{69}$$

Therefore, when $\Delta_2 > \Delta_1$ (resp. $\Delta_1 > \Delta_2$), the initial condition $\mathbf{r}_0$ needs to have a substantial component along the vector $\mathbf{v}^{(2)}$ (resp. $\mathbf{v}^{(1)}$). To formalize this observation, we compute the peak time $t^*$ when $||\mathbf{r}(t; \mathbf{r}_0 = \mathbf{v}^{(2)})||^2$ takes the largest value. If *Equation (60)* holds and $\Delta_2 \gg \Delta_1$, the peak time can be approximated by $\omega t^* \approx \pi/2$.

We then examine the value of the distance from baseline $||\mathbf{r}(t^*; \mathbf{r}_0)||^2$ when the initial condition $\mathbf{r}_0$ is a linear combination of the vectors $\mathbf{v}^{(1)}$, $\mathbf{v}^{(2)}$ and a vector orthogonal to both $\mathbf{v}$'s, that is, $\mathbf{r}_0 = \alpha_1 \mathbf{v}^{(1)} + \alpha_2 \mathbf{v}^{(2)} + \beta \mathbf{z}^{(\perp)}$, with $\alpha_1^2 + \alpha_2^2 + \beta^2 = 1$. At the peak time we can write the distance from baseline as:

$$||\mathbf{r}(t^*; \mathbf{r}_0)||^2 = e^{-2t^*} \left[ \frac{\Delta_1}{\Delta_2} \alpha_1^2 + \frac{\Delta_2}{\Delta_1} \alpha_2^2 + \beta^2 \right]. \tag{70}$$

By requiring that, at the peak time, the distance from baseline is larger than unity we can derive the following sufficient conditions on the component of the initial conditions along the vectors $\mathbf{v}^{(1)}$, $\mathbf{v}^{(2)}$ and $\mathbf{z}^{(\perp)}$:

$$\frac{\Delta_1}{\Delta_2} \alpha_1^2 + \frac{\Delta_2}{\Delta_1} \alpha_2^2 + \beta^2 > e^{2t^*}. \tag{71}$$

When the component of the initial condition along $\mathbf{z}^{(\perp)}$ is zero ($\beta = 0$), *Equation (71)* is satisfied when

$$\alpha_2^2 > \left(\frac{\Delta_2}{\Delta_1}e^{2t^*} - 1\right) / \left(\frac{\Delta_2^2}{\Delta_1^2} - 1\right). \tag{72}$$

*Equation (71)* shows that in an amplified rotational channel the initial state $\mathbf{r}_0$ may have a significant component orthogonal to the vector $\mathbf{v}^{(2)}$ and yet be able to generate amplified dynamics. However, from *Equation (70)* we observe that, for a fixed value of $\Delta_1$ and $\Delta_2$ (with $\Delta_2 > \Delta_1$), the amplification decreases when the component along $\mathbf{v}^{(2)}$, that is, $\alpha_2$, decreases. Therefore, to have amplification the component of $\mathbf{r}_0$ along $\mathbf{v}^{(2)}$ should be sufficiently strong.

The above condition *Equation (71)* can be generalized to the case of a superposition of orthogonal rank-2 rotational channels (see *Equation (68)*) if we assume that the peak time is approximately the same when considering separately the dynamics within each rank-2 channel. Under these assumptions, the distance from baseline at the peak time can be written as

$$||\mathbf{r}(t^*; \mathbf{r}_0)||^2 = e^{-2t^*} \sum_{k=1}^{K} \left[\frac{\Delta_1^{(k)}}{\Delta_2^{(k)}}\alpha_1^{(k)2} + \frac{\Delta_2^{(k)}}{\Delta_1^{(k)}}\alpha_2^{(k)2}\right] + e^{-2t^*}\beta^2, \tag{73}$$

and the condition for the relationship between initial condition and connectivity vectors in a rank-2K channel becomes:

$$\sum_{k=1}^{K} \frac{\Delta_1^{(k)}}{\Delta_2^{(k)}}\alpha_1^{(k)2} + \frac{\Delta_2^{(k)}}{\Delta_1^{(k)}}\alpha_2^{(k)2} + \beta^2 > e^{2t^*}. \tag{74}$$

When the component of the initial condition along $\mathbf{z}^{(\perp)}$ is zero ($\beta = 0$) and the $\Delta_i^{(k)}$ do not depend on the channel $k$, *Equation (74)* is satisfied when

$$\sum_{k=1}^{K} \alpha_2^{(k)2} > \left(\frac{\Delta_2}{\Delta_1}e^{2t^*} - 1\right) / \left(\frac{\Delta_2^2}{\Delta_1^2} - 1\right) \tag{75}$$

where the left-hand side represents the component of the initial condition on all the vectors $\mathbf{v}^{(2;k)}$'s. From *Equation (73)* we note that, for fixed values of the $\Delta_1^{(k)}$ and $\Delta_2^{(k)}$ (assuming $\Delta_2^{(k)} > \Delta_1^{(k)}$ for all $k$), amplification decreases by decreasing the components of the initial condition on the vectors $\mathbf{v}^{(2;k)}$'s. Therefore, to generate amplified dynamics, the components of the initial condition $\mathbf{r}_0$ on the vectors $\mathbf{v}^{(2;k)}$'s should be sufficiently strong.

## Correlation between initial and peak state

In this section we derive the expression for the correlation between the initial state $\mathbf{r}_0$ and the state at the peak of the transient dynamics, for a rank-2K rotational channel consisting of K rotational rank-2 channels (see Section 'Dynamics of a low-rank rotational channel'). We set the initial condition to $\mathbf{r}_0 = \sum_{k=1}^{K} \alpha_1^{(k)}\mathbf{v}^{(1;k)} + \sum_{k=1}^{K} \alpha_2^{(k)}\mathbf{v}^{(2;k)} + \beta\mathbf{z}^{(\perp)}$ and evaluate the state at the peak at time $t^*$ (corresponding to the peak time when $\mathbf{r}_0$ has a strong component on the $\mathbf{v}^{(2;k)}$'s and $\Delta_2^{(k)} \gg \Delta_1^{(k)}$). The initial condition and the peak state are therefore given by:

$$\mathbf{r}(0) = \sum_{k=1}^{K} \alpha_1^{(k)}\mathbf{v}^{(1;k)} + \sum_{k=1}^{K} \alpha_2^{(k)}\mathbf{v}^{(2;k)} + \beta\mathbf{z}^{(\perp)}$$

$$\mathbf{r}(t^*) = \frac{1}{e^{t^*}} \sum_{k=1}^{K} \left[-\mathbf{v}^{(1;k)}\sqrt{\frac{\Delta_2^{(k)}}{\Delta_1^{(k)}}}\alpha_2^{(k)} + \mathbf{v}^{(2;k)}\sqrt{\frac{\Delta_1^{(k)}}{\Delta_2^{(k)}}}\alpha_1^{(k)}\right] + \frac{1}{e^{t^*}}\beta\mathbf{z}^{(\perp)}. \tag{76}$$

From the mutual orthogonality between $\mathbf{v}^{(1;k)}$, $\mathbf{v}^{(2;k)}$, and $\mathbf{z}^{(\perp)}$ it follows that

$$\|\mathbf{r}(0)\|^2 = 1$$

$$\|\mathbf{r}(t^*)\|^2 = \frac{1}{e^{2t^*}} \sum_{k=1}^{K} \left[ \frac{\Delta_2^{(k)}}{\Delta_1^{(k)}} \alpha_2^{(k)2} + \frac{\Delta_1^{(k)}}{\Delta_2^{(k)}} \alpha_1^{(k)2} \right] + \frac{1}{e^{2t^*}} \beta^2. \tag{77}$$

Thus, we can write the correlation between the initial condition and the peak state as

$$\frac{\mathbf{r}(0) \cdot \mathbf{r}(t^*)}{\|\mathbf{r}(0)\|\,\|\mathbf{r}(t^*)\|} = \frac{\sum_{k=1}^{K} \left( \sqrt{\frac{\Delta_1^{(k)}}{\Delta_2^{(k)}}} - \sqrt{\frac{\Delta_2^{(k)}}{\Delta_1^{(k)}}} \right) \alpha_1^{(k)} \alpha_2^{(k)} + \beta \mathbf{z}^{(\perp)}}{e^{t^*} \|\mathbf{r}(t^*)\|}. \tag{78}$$

If the initial condition has a strong component on the vectors $\mathbf{v}^{(2)}$'s (and the component on $\mathbf{z}^{(\perp)}$ can be neglected, that is, $\beta = 0$), so that $\alpha_1^{(k)2} = \epsilon^2$ and $\alpha_2^{(k)2} = 1 - \epsilon^2$, then the correlation between initial and peak state satisfies

$$\frac{\mathbf{r}(0) \cdot \mathbf{r}(t^*)}{\|\mathbf{r}(0)\|\,\|\mathbf{r}(t^*)\|} \propto \sum_{k=1}^{K} \alpha_1^{(k)} \alpha_2^{(k)} = K\epsilon, \tag{79}$$

and equal to zero when the initial condition is a linear combination of the vectors $\mathbf{v}^{(2)}$'s.

## Single-trial analysis of population OFF responses

In this section we focus on the single-trial structure of the dynamics generated by the network and single-cell models. In particular, we consider a setting where dynamics are amplified and examine the amount of single-trial variability along specific directions in the state space: the direction corresponding to the maximum amplification of the dynamics, denoted by $\mathbf{r}_{\mathrm{ampl}}$, and a random direction $\mathbf{r}_{\mathrm{rand}}$. Variability along a given direction $\mathbf{z}_0$ at time $t$ can be written as a function of the covariance matrix $\mathbf{C}(t)$ of the population activity at time $t$ as:

$$\mathrm{var}(\mathbf{z}_0; t) = \mathbf{z}_0^T \mathbf{C}(t) \mathbf{z}_0. \tag{80}$$

In particular, for a given direction $\mathbf{z}_0$, we examine the ratio between the variability at the time of maximum amplification $t = t^*$ and the variability at time $t = 0$ (corresponding to the end of stimulus presentation), which we termed variability amplification (VA), defined as:

$$\mathrm{VA}(\mathbf{z}_0) = \frac{\mathrm{var}(\mathbf{z}_0; t^*)}{\mathrm{var}(\mathbf{z}_0; 0)} \tag{81}$$

We show that the network model predicts that the variability should increase from $t = 0$ to $t = t^*$ along the amplified direction $\mathbf{r}_{\mathrm{ampl}}$, but not along a random direction $\mathbf{r}_{\mathrm{rand}}$, resulting in the relationship $VA(\mathbf{r}_{\mathrm{ampl}}) > VA(\mathbf{r}_{\mathrm{rand}})$. In contrast, in the single-cell model the presence of VA depends on the shapes of the single-cell responses. Further, we show that changes in VA between the amplified and random directions are due to distinct mechanisms in the network and single-cell models: in the network model they are due to the inter-neuron correlations generated by network recurrency; in the single-cell model they directly stem from changes in the variance of single neurons in the population.

## Structure of single-trial responses in the network model

In the following we examine the structure of single-trial population OFF responses generated by a general linear network model. We show that the network model is able to reproduce the experimental observations on VA under the minimal assumption that the variability in the single-trial responses is due to the noise in the initial condition $\mathbf{r}_0$ at stimulus offset. The analysis can nevertheless be extended to the case where external input noise is fed into the network without affecting the key results.

We consider a linear network model with connectivity $\mathbf{J}$ and a single stimulus $s$ modeled by the trial-averaged initial condition $\mathbf{r}_0$. We assume that the trial-to-trial noise in the initial condition is Gaussian, so that for trial $\nu$ the corresponding initial condition $\mathbf{r}_0^{(\nu)}$ is given by

$$\mathbf{r}_0^{(\nu)} = \mathbf{r}_0 + \eta^{(\nu)}, \tag{82}$$

where $\langle \eta_i^{(\nu)} \rangle = 0$ and $\langle \eta_i^{(\nu)} \eta_j^{(\nu)} \rangle = s^2 \delta_{ij}$, so that $\langle \mathbf{r}_0^{(\nu)} \rangle = \mathbf{r}_0$. The solution of the linear system can thus be analytically expressed as:

$$\begin{aligned} \mathbf{r}^{(\nu)}(t) &= \mathbf{P}_t \mathbf{r}_0^{(\nu)} \\ &= \exp(t(\mathbf{J} - \mathbf{I}))(\mathbf{r}_0 + \eta^{(\nu)}), \end{aligned} \tag{83}$$

where the time-dependent matrix $\mathbf{P}_t$ is the propagator of the system (**Arnold, 1973**).

We start by computing the covariance matrix of the population activity at an arbitrary time $t$ as defined by:

$$\mathbf{C}(t) = \langle \left( \mathbf{r}^{(\nu)}(t) - \langle \mathbf{r}^{(\nu)}(t) \rangle \right) \left( \mathbf{r}^{(\nu)}(t) - \langle \mathbf{r}^{(\nu)}(t) \rangle \right)^T \rangle. \tag{84}$$

Using **Equations (83) and (84)**, we can write the covariance matrix as (**Farrell and Ioannou, 1996**; **Farrell and Ioannou, 2001**):

$$\begin{aligned} \mathbf{C}(t) &= \langle \mathbf{P}_t \eta^{(\nu)} \eta^{(\nu)T} \mathbf{P}_t^T \rangle = \mathbf{P}_t \langle \eta^{(\nu)} \eta^{(\nu)T} \rangle \mathbf{P}_t^T \\ &= s^2 \mathbf{P}_t \mathbf{P}_t^T, \end{aligned} \tag{85}$$

so that $\mathbf{C}(0) = s^2 \mathbf{I}$.

At a given time $t = t^*$, let the SVD of the propagator is given by

$$\mathbf{P}_{t^*} = \sum_{i=1}^{N} \sigma_i \mathbf{m}^{(i)} \mathbf{n}^{(i)T} = \mathbf{M} \mathbf{\Sigma} \mathbf{N}^T, \tag{86}$$

where the singular values $\sigma_i$ are positive numbers. If none of the singular values $\sigma_i$ is larger than unity, no initial condition $\mathbf{r}_0$ can lead to amplified dynamics at time $t^*$, meaning that for all initial conditions $||\mathbf{r}(t^*)|| < 1$ (**Bondanelli and Ostojic, 2020**). Instead, suppose that the first $K$ singular values $\{\sigma_k\}_{k=1}^{K}$ are larger than unity, while the remaining $N - K$ singular values are smaller than one (with arbitrary $K$). Under this condition, any initial condition $\mathbf{r}_0$ consisting of a linear combination of the first K vectors $\mathbf{n}^{(k)}$ (i.e. $\mathbf{r}_0 = \sum_{k=1}^{K} \alpha_k \mathbf{n}^{(k)}$) will be amplified, so that the norm of the population vector at time $t^*$ is larger than one (**Bondanelli and Ostojic, 2020**):

$$\mathbf{r}(t^*) = \mathbf{P}_{t^*} \mathbf{r}_0 = \sum_{k=1}^{K} \alpha_k \sigma_k \mathbf{m}^{(k)} \tag{87}$$

$$||\mathbf{r}(t^*)|| > 1$$

In the following analysis, we explore this last scenario, where the first $K$ singular values are larger than unity. We make the following two assumptions. First, we assume that the initial condition is a linear combination of the first $K$ vectors $\mathbf{n}^{(k)}$'s, so that the amplified direction is given by $\mathbf{r}_{ampl} = \mathbf{r}(t^*) / ||\mathbf{r}(t^*)||$ (see **Equation (87)**). Notably, for any specified initial condition $\mathbf{r}_0$ defined in this way, the following analysis holds for all times $t^*$ for which $||\mathbf{r}(t^*)|| > 1$, including the peak time where the distance from baseline is maximum. Second, we assume that averages of the first or second powers of the singular values are of order (at most) one, that is, $\sum_{i=1}^{M} \sigma_i / M, \sum_{i=1}^{M} \sigma_i^2 / M = O(1)$ for any integer $M$. This assumption holds for typically studied network topologies (e.g. Gaussian random, low-rank networks), where a few amplified dimensions (corresponding to $\sigma_i > 1$) coexist with many non-amplified ones (corresponding to $\sigma_i < 1$) (see **Bondanelli and Ostojic, 2020**).

Using **Equation (84)** and the SVD of the propagator $\mathbf{P}_{t^*}$ (**Equation (86)**), we can write the covariance matrix at time $t^*$ as

$$\mathbf{C}(t^*) = s^2 \mathbf{M} \mathbf{\Sigma}^2 \mathbf{M}^T \tag{88}$$

Thus, we can compute the amount of variability along the amplified direction $\mathbf{r}_{ampl}$ (**Equation (87)**) and a random direction $\mathbf{r}_{rand}$ (with $\mathbf{r}_{rand} = \sum_{i=1}^{N} \beta_i \mathbf{m}^{(i)}$, with $\beta_i \sim \mathcal{N}(0, 1/N)$), using **Equation (80)** as:

$$\begin{aligned}
\mathrm{var}(\mathbf{r}_{\mathrm{ampl}};0) \quad &= s^2 \\
\mathrm{var}(\mathbf{r}_{\mathrm{rand}};0) \quad &= s^2
\end{aligned}$$

$$\mathrm{var}(\mathbf{r}_{\mathrm{ampl}};t^*) \quad = s^2 \sum_{k=1}^{K} \alpha_k^2 \sigma_k^4 \Big/ \sum_{k=1}^{K} \alpha_k^2 \sigma_k^2 \gg s^2 \tag{89}$$

$$[\mathrm{var}(\mathbf{r}_{\mathrm{rand}};t^*)]_{\mathbf{r}_{\mathrm{rand}}} = s^2 \left[ \sum_{i=1}^{N} \beta_i^2 \sigma_i^2 \right] = s^2 \sum_{i=1}^{N} [\beta_i^2]\,\sigma_i^2 = \frac{s^2}{N} \sum_{i=1}^{N} \sigma_i^2 = O(s^2),$$

where in the third equation the sum runs only over the singular values larger than one, while in the fourth equation the sum runs over all singular values (both larger and smaller than unity), and brackets denote average over realizations of $\mathbf{r}_{\mathrm{rand}}$. *Equation (89)* shows that in a linear network model the variability along the amplified direction $\mathbf{r}_{\mathrm{ampl}}$ increases from the initial time $t=0$ to the peak time $t=t^*$, while the variability along a random direction does not, and instead decreases. We quantified these observations by computing the VA, defined in *Equation (81)* (*Figure 8*). Using *Equation (89)* we obtain:

$$\begin{aligned}
\mathrm{VA}(\mathbf{r}_{\mathrm{rand}}) &= O(1) \\
\mathrm{VA}(\mathbf{r}_{\mathrm{ampl}}) &\gg 1,
\end{aligned} \tag{90}$$

thus showing that, for a linear network model, VA is larger when computed for the amplified direction than a random direction.

## Shuffled responses

In the following we compare the VA of the original responses with the VA obtained from the responses where the trial labels have been shuffled independently for each cell. The covariance matrix of the shuffled responses $\mathbf{C}^{(\mathrm{sh})}(t)$ is obtained by retaining only the diagonal elements of the real covariance matrix $\mathbf{C}(t)$:

$$\mathbf{C}_{ij}^{\mathrm{sh}}(t) = \mathbf{C}_{ij}(t)\delta_{ij} = \sum_{l=1}^{N} \mathbf{m}_i^{(l)} \sigma_l^2 \mathbf{m}_j^{(l)} \delta_{ij}. \tag{91}$$

The first two quantities in *Equation (89)* computed using $\mathbf{C}^{\mathrm{sh}}$ instead of $\mathbf{C}$ do not change, since $\mathbf{C}(0)$ is already diagonal. Instead, for shuffled responses the variability along both directions at the peak time $t^*$ is of order $O(s^2)$, since:

$$\begin{aligned}
[\mathrm{var}(\mathbf{r}_{\mathrm{rand}};t^*;\mathrm{shuffled})]_{\mathbf{r}_{\mathrm{rand}},\{\mathbf{m}^{(k)}\}_k} &= s^2 \left[ \sum_{i,j,l=1}^{N} \mathbf{r}_{\mathrm{rand},i} \mathbf{m}_i^{(l)} \sigma_l^2 \mathbf{m}_j^{(l)} \delta_{ij} \mathbf{r}_{\mathrm{rand},j} \right] \\
&= s^2 \sum_{i,j,l=1}^{N} \sigma_l^2 \delta_{ij} [\mathbf{r}_{\mathrm{rand},i} \mathbf{r}_{\mathrm{rand},j}][\mathbf{m}_i^{(l)} \mathbf{m}_j^{(l)}] = \frac{s^2}{N^2} \sum_{i,j,l=1}^{N} \sigma_l^2 \delta_{ij} = \frac{s^2}{N} \sum_{l=1}^{N} \sigma_l^2 = O(s^2).
\end{aligned} \tag{92}$$

Following the same steps it is possible to show that $[\mathrm{var}(\mathbf{r}_{\mathrm{ampl}};t^*;\mathrm{shuffled})]_{\mathbf{r}_{\mathrm{ampl}},\{\mathbf{m}^{(k)}\}_k} = O(s^2)$. In fact if we set, according to *Equation (87)*, $\mathbf{r}_{\mathrm{ampl}} = \sum_{k=1}^{K} \alpha_k \sigma_k \mathbf{m}^{(k)}/C$, where $C = \sum_{k=1}^{K} \alpha_k^2 \sigma_k^2 = O(1)$, we have:

$$\begin{aligned}
[\mathrm{var}(\mathbf{r}_{\mathrm{ampl}};t^*;\mathrm{shuffled})]_{\mathbf{r}_{\mathrm{ampl}},\{\mathbf{m}^{(k)}\}_k} &= \frac{s^2}{C^2} \sum_{i,j,l=1}^{N} \sum_{k_1,k_2=1}^{K} \sigma_{k_1} \sigma_{k_2} \sigma_l^2 [\alpha_{k_1} \alpha_{k_2}][\mathbf{m}_i^{(k_1)} \mathbf{m}_j^{(k_1)} \mathbf{m}_i^{(l)} \mathbf{m}_i^{(l)}]\delta_{ij} \\
&= \frac{s^2}{C^2} \sum_{i,l=1}^{N} \sum_{k_1,k_2=1}^{K} \sigma_{k_1} \sigma_{k_2} \sigma_l^2 [\alpha_{k_1} \alpha_{k_2}] \left( \delta_{lk_1} \delta_{k_1 k_2} \frac{1}{N^2} + \delta_{k_1 k_2} \frac{1}{N^2} \right) \\
&= \frac{s^2}{C^2} (O(1/N) + O(1)) = O(s^2),
\end{aligned} \tag{93}$$

where in the last equality we assumed that $[\alpha_k^2] \approx 1/K$ if $\alpha_k \sim \mathcal{N}(0,1/K)$. To summarize, for shuffled responses we have:

$$\text{var}(\mathbf{r}_{\text{ampl}};0;\text{shuffled}) = s^2$$
$$\text{var}(\mathbf{r}_{\text{rand}};0;\text{shuffled}) = s^2, \tag{94}$$

so that the VA along both directions is given by:

$$\text{VA}(\mathbf{r}_{\text{rand}}) = O(1)$$
$$\text{VA}(\mathbf{r}_{\text{ampl}}) = O(1), \tag{95}$$

implying a weaker modulation of the VA between random and amplified direction for shuffled responses with respect to the original responses. These analyses show that in the network model, the difference in the VA between the amplified and random directions is a result of the recurrent interactions within the network, reflected in the off-diagonal elements of the covariance matrix $\mathbf{C}(t)$, and it therefore cannot be explained solely by changes in the variability of single units.

## Structure of single-trial responses in the single-cell model

Here we examine the structure of the single-trial population responses generated by the single-cell model (*Equation (1)*). We consider single-trial responses $\mathbf{r}^{(\nu)}(t)$ to a single stimulus generated by perturbing the initial state of the trial averaged responses $\mathbf{r}_0$, yielding

$$\begin{aligned} r_i^{(\nu)}(t) &= r_{0,i}^{(\nu)} L_i(t) \\ &= (r_{0,i} + \eta_i^{(\nu)}) L_i(t), \end{aligned} \tag{96}$$

where $\eta^{(\nu)}$ has zero mean and variance equal to $s^2$ ($\langle \eta_i^{(\nu)} \rangle = 0$ and $\langle \eta_i^{(\nu)} \eta_j^{(\nu)} \rangle = s^2 \delta_{ij}$). From *Equation (96)* we can compute the covariance matrix of the population response for the single-cell model as:

$$\begin{aligned} \mathbf{C}_{ij}(t) &= L_i(t) \langle \eta_i^{(\nu)} \eta_j^{(\nu)} \rangle L_j(t) \\ &= s^2 L_i^2(t) \delta_{ij}. \end{aligned} \tag{97}$$

We can obtain the variability along a given direction $\mathbf{z}_0$ as:

$$\text{var}(\mathbf{z}_0;t) = s^2 \sum_{i=1}^{N} \mathbf{z}_{0,i}^2 L_i^2(t). \tag{98}$$

Notably, the term on the right-hand side is proportional to the norm of the dynamics that would be evoked by the initial condition $\mathbf{z}_0$. In the single-cell model, the variability is maximally amplified along the initial condition that leads to largest amplification of the trial averaged dynamics. In contrast, in the network model, the variability is maximally amplified along the amplified direction of the trial-averaged dynamics, which is in general orthogonal to the corresponding initial condition (see Materials and methods, Section 'Conditions on the initial state $\mathbf{r}_0$'; *Figure 6C,E*). In general, the relationship between the modulation of the variability along the amplified direction and modulation along a random direction depends on the specific shape of the temporal filters $L_i(t)$, and we used simulations of the fitted model to determine it. We can however contrast the single-trial structure generated by the single-cell model with the structure generated by the recurrent model by noting that the covariance matrix given by *Equation (97)* is diagonal, and is therefore equal to the covariance matrix of the shuffled responses, $\mathbf{C}_{ij}(t) = \mathbf{C}_{ij}^{\text{sh}}(t)$. This implies that no difference is observed when computing the VA for the amplified and random directions for the real and shuffled responses (i.e. $\text{VA}(\mathbf{r}_{\text{rand}}) = \text{VA}(\mathbf{r}_{\text{rand}};\text{shuffled})$ and $\text{VA}(\mathbf{r}_{\text{ampl}}) = \text{VA}(\mathbf{r}_{\text{ampl}};\text{shuffled})$), indicating that changes in the trial-to-trial variability can be fully explained in terms of changes in the variability of single units.

In *Figure 8E,G,F,H*, for both network and single-cell models, single-trial population responses were generated by drawing the single-trial initial conditions from a random distribution with mean $\mathbf{r}_0$ and covariance matrix computed from the single-trials initial conditions of the original data (across neurons for the single-cell model, across PC dimensions for the network model). The results do not substantially change if we draw the single-trial initial conditions from a random distribution with

mean $\mathbf{r}_0$ and isotropic covariance matrix ($\mathbf{r}_0^{(\nu)} \sim \mathcal{N}(\mathbf{r}_0, \mathbf{C}(0))$, with $\mathbf{C}(0) = s^2\,\mathbf{I}$), as assumed in the analysis above (not shown).

## Software

Numerical simulations and data analyses were done using Python 3.6 (using NumPy [*Harris et al., 2020*], SciPy [*Virtanen et al., 2020*], and the scikit-learn package [*Pedregosa et al., 2011*]). Code is available at https://github.com/gbondanelli/OffResponses; *Bondanelli, 2021*; copy archived at swh: 1:rev:2438e688ad719eb9870af8c032803a7367fe1140. Surrogate data sets for model validation (see Materials and methods, Section 'Control data sets', *Figure 3*, *Figure 3—figure supplement 2*, *Figure 3—figure supplement 3*) were generated using the Matlab code provided by *Elsayed and Cunningham, 2017* available at https://github.com/gamaleldin/TME.

# Additional information

## Competing interests

Brice Bathellier: Reviewing editor, *eLife*. The other authors declare that no competing interests exist.

## Funding

| Funder | Grant reference number | Author |
| --- | --- | --- |
| Agence Nationale de la Recherche | ANR-16-CE37-0016 | Giulio Bondanelli Srdjan Ostojic |
| Agence Nationale de la Recherche | ANR-17-EURE-0017 | Giulio Bondanelli Srdjan Ostojic |

The funders had no role in study design, data collection and interpretation, or the decision to submit the work for publication.

## Author contributions

Giulio Bondanelli, Conceptualization, Software, Formal analysis, Validation, Investigation, Visualization, Methodology, Writing - original draft, Writing - review and editing; Thomas Deneux, Data curation, Methodology; Brice Bathellier, Conceptualization, Resources, Data curation, Investigation, Methodology, Writing - review and editing; Srdjan Ostojic, Conceptualization, Formal analysis, Supervision, Funding acquisition, Validation, Investigation, Methodology, Project administration, Writing - review and editing

## Author ORCIDs

Giulio Bondanelli ⓘD https://orcid.org/0000-0001-6781-4939
Thomas Deneux ⓘD http://orcid.org/0000-0002-9330-7655
Brice Bathellier ⓘD http://orcid.org/0000-0001-9211-1960
Srdjan Ostojic ⓘD https://orcid.org/0000-0002-7473-1223

## Decision letter and Author response

Decision letter https://doi.org/10.7554/eLife.53151.sa1
Author response https://doi.org/10.7554/eLife.53151.sa2

# Additional files

## Supplementary files

• Transparent reporting form

## Data availability

Python code and data are available at https://github.com/gbondanelli/OffResponses copy archived at https://archive.softwareheritage.org/swh:1:rev:2438e688ad719eb9870af8c032803a7367fe1140/.

The following datasets were generated:

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
