## [Decision Letter]

**Acceptance summary:**

This paper explores the hypothesis that OFF responses are single-cell signatures of network dynamics, and shows convincingly that a model based on recurrent interactions does a much better job explaining the data than the de- facto standard model, which is based purely on single cell dynamics. In addition, the proposed recurrent network model generates transient OFF responses very similar to what is seen in experiments.

**Decision letter after peer review:**

Thank you for submitting your article "Population coding and network dynamics during OFF responses in auditory cortex" for consideration by *eLife*. Your article has been reviewed by three peer reviewers, one of whom is a member of our Board of Reviewing Editors, and the evaluation has been overseen by Barbara Shinn-Cunningham as the Senior Editor. The following individual involved in review of your submission has agreed to reveal their identity: Guillaume Hennequin (Reviewer #2).

The reviewers have discussed the reviews with one another and the Reviewing Editor has drafted this decision to help you prepare a revised submission. Please aim to submit the revised version within two months.

Summary:

In this paper, Bondanelli and colleagues study the spatiotemporal structure of OFF responses in A1. They propose that these responses arise from a simple linear mechanism involving a certain form of nonnormal population dynamics. They fit such a network model to A1 data and find that it explains several important aspects of OFF responses, some, but importantly, not all, of which are also captured by a single-neuron mechanism.

Essential revisions:

We're very sorry for the long delay. That's in part because at least one of use found the paper very hard to read, and it took a long time to figure out what was going on. And as you'll see, it's not clear if we did that very well. We think the problem is that you're missing quantitative analysis; in many cases it seems that we're supposed to understand why a particular piece of analysis supported your model, but it simply wasn't clear to us why that was true. Explicit examples of this will be given below.

A major issue for us has been that you begin with a discussion of special cases of Equation (2) (last two paragraphs of page 6) but then go on to test predictions of THE model without telling us which of these special assumptions you are actually making (if any) in the production of your results (Figure 4 specifically). We believe that the first paragraph of "Testing the network mechanisms on the data" should begin with a clear description of the actual model considered (including whether your transient channels are mutually orthogonal, and how initial conditions relate to those planes).

More specifically, to summarize our current understanding: under the assumption that the v's are orthonormal, the u's are orthogonal, and u_k_ dot v_l_ = 0 for all k \ne l (not strictly true, but close enough in high dimensions), the most general solution (a generalization of Eq. 34 to an arbitrary number of modes) is

r(t) = [uperp + sum_k_ u_k_ (a_k_ + v_k_ dot u_k_ t)] exp(-t)

where uperp is perpendicular to all the u_k_ and the a_k_ are arbitrary.

This appears late in Methods. But it's important, because there seem to be hidden assumptions about uperp and the a_k_. Namely,

- the a_k_ are all set to zero

- uperp = u_k_ for only one k, with k dependent on the stimulus.

Assuming this is correct, it's a critical part of the model. And it is, in fact, an extra assumption.

Given that setup, we believe there were several predictions:

a. Lines 216-228: There's a prediction about the correlational structure that we didn't fully understand. Equations are needed, in the main text, that make the predictions explicit.

b. Lines 229-238: Correlations between the beginning and the peak of the off response should small. Again, equations are needed; this should be quantified.

c. Lines 239-253:

"To further examine to which extent the population dynamics during OFF responses could be accounted for by a non-normal linear dynamical system, we next fitted our network model population OFF responses to all stimuli at once using reduced-rank regression (Figure S1; see Methods). Qualitatively, we found that the fit reproduced well the low-dimensional population trajectories (Figure 4C)."

As far as we could tell, this just tells us that the fit is good; it doesn't tell us anything about whether the OFF responses could be accounted for by a non-normal linear dynamical system. Is that correct, or did we miss something? If that is correct, it seems like this paragraph is mainly a distractor.

d. Lines 254-262: The fitted connectivity, J, had eigenvalues less than 1 and the symmetrized version had eigenvalues greater than 1. That's consistent with the low rank, non-normal model. However, couldn't such a J could come from a high rank, non-normal model? If so, it's not strong evidence for your model. This should be clarified.

e. Lines 263-283: There were several results here, but it's not clear how much they supported the model. For example,

"We computed the overlap between the subspaces spanned by the transient channels for each pair of stimuli s_1_ and s_2_ , analogous to the overlap between the vectors u^(s1)^ and u^(s2)^ in Eq. (2) (see Methods). We found that the overlap between the transient channels matched the overlap between the response dynamics and population states at the end of stimulus presentation (Figure 4F, right panel), consistent with the interpretation that individual OFF responses may be generated through transient coding channels of the form given by Eq. (2)."

It's not clear what we should expect here. Is this strong or weak evidence for the model? You need to quantify what we should expect, and be clear whether this is only consistent with the model, or provides evidence for it.

"While the fraction of variance explained when using the matrix J_Sum_ was necessarily lower than the one computed using J_Full_ , the model with connectivity J_Sum_ could still explain a consistent fraction of the total variance, and yielded values of R^2^ significantly higher than the ones obtained from the shuffles."

Again, it's not clear what we should expect. Is this strong or weak evidence for the model?

f. Lines 328-331: for the single cell model in Eq. 3, the correlations between the beginning and peak of the off response are high. This seems easily fixable with a very minor tweak. Suppose there are two kinds of neurons, some that peak early,

r_i_(t) = a_i_(s) f_i_(t)

and some that peak late,

r_i_(t) = b_i_(s) g_i_(t)

(for instance, f_i_(t) = Theta(t) e^{-t} and g_i_(t) = Theta(t) t*e^{-t}). Different stimuli (s) could activate different sets of neurons.

Alternatively, the single neuron responses could be of the form

r_i_(t) = a_i_(s) f_i_(t) + b_i_(s) g_i_(t)

where the a's and b's are anti correlated (either a is large and b is small or vice versa) and f_i_ and g_i_ are as above.

Are these model ruled out by the data? If not, that seems to weaken the paper (although maybe there are other things wrong with these models). In either case, models like these should be discussed.

g. Lines 332-353:

"A second important difference between the two models is that in the single-cell model the temporal shape of the response of each neuron is the same across all stimuli, while for the recurrent model this is not necessarily the case."

Given this opening sentence, why not just show that for the recurrent network model, the temporal shape of the response is not the same across all stimuli? Instead, there's a lot of analysis that we didn't fully understand. In particular, there's a rather sudden switch from specifying the form of L_i_(t) (Eq. 44) and fitting L_i_(t) (Eq. 48). What's missing is a quantitative prediction of what we should expect, followed by a demonstration that the data is consistent with that prediction.

In addition, we have some specific suggestions.

1. Lines 174-177,

"We focus on two outstanding properties identified in the OFF responses to different stimuli. The first one is the strong amplification of OFF responses, as quantified by the transient increase of the distance from baseline ||r(t)||. The second important feature is the low-dimensionality of OFF response trajectories and their global structure across different stimuli."

It seems that this is all you're going to focus on, but it's not. It would be nice to know, at some point early in the manuscript, exactly what you're going to do.

2. What's the difference between Figures 2D and 4A (left panel).

3. Lines 329-31,

"The single-cell model predicts that these two states are only weakly decorrelated and essentially lie along a single dimension (Figure 5E and S7), which is inconsistent with experimental observations"

Which experimental observations?

4. Throughout the paper we found it unclear how initial conditions were chosen for the model, e.g. in Figure 3. From line 167, we are led to think that they are picked from the actual data, "we model the firing activity at the end of stimulus presentation as the initial condition of the network dynamics" -- since the model doesn't explicitly models the stimulus period as far as we understand (?), we infer this activity is taken from the data? But does the model even have the same number of neurons?

5. The purpose of Figure 2 is to convey that different stimuli evoke orthogonal OFF responses, and these OFF responses are low dimensional. By my calculation, for each conditional mean (over 20 trials), there are 400ms * 31.5 frame/s ~= 13 samples used to estimate the PCs and variance explained resulting in a maximum dimensionality (PCs to get to 80%) of 13. If this is the case, a more careful comparison to the cross-condition dimensionality is warranted, which accounts for sample size.

Also, we would expect there to be fewer time points on the traces in Figure 2B. How were the trial-averaged responses interpolated?

What steps were taken to insure that the same unit wasn't double counted during pooling across sessions. Mistakes in this process can contribute to inflated variance along primary PC's creating the illusion of low dimensionality.

6. Statements of relative degree of subspace overlap should be verified with a hypothesis test. For data, this can be done by resampling neurons.

7. You should justify why LDA (naive covariance assumption) is more suitable for assessing linear separability in Figure 1C than an SVM. Absent justification, an SVM should be used to get the clearest assessment of linear separability.

[Editors' note: further revisions were suggested prior to acceptance, as described below.]

Thank you for resubmitting your work entitled "Network dynamics underlying OFF responses in the auditory cortex" for further consideration by *eLife*. Your revised article has been evaluated by Barbara Shinn-Cunningham (Senior Editor) and a Reviewing Editor.

The manuscript has been improved but there are some remaining issues that need to be addressed before acceptance, as outlined below:

The good news is that the paper is much improved. The writing is more clear, and the analysis of dimensions of amplified trial-to-trial variability is compelling. The less good news is that now that we understand it better, additional issues have come up. All can, we believe, be dealt with relatively easily.

Let's start with the data you're are trying to explain. Basically, after the offset of an auditory stimulus, there is transient amplification, with different stimuli activating nearly orthogonal subspaces.

To model this, low rank linear dynamics is used, of the form

dr/dt = sum_{k=1}^R u_k_ v_k_ r – r,

where r (=r1, r2,.…, rN) is the trial averaged activity. The network had a few additional properties,

a. The rank, R, is much less than the number of neurons, N.

b. The u's and v's are nearly orthogonal.

c. The symmetric part of the connectivity matrix must have eigenvalues with real part >1.

d. The model has no external input, so the response to a particular stimulus (after offset) depends only on the value of r immediately after stimulus offset, here denoted r0(s) where s is the stimulus.

e. For any stimulus, s, r0(s) has appreciable overlap with a small number of v_k_ (on the order of 5).

This model did an OK job explaining the data. In particular, Figure 4H, left (the model), is supposed to match Figure 2D (the data). But the match is not so good: the off-diagonal channel overlap in Figure 2D is near zero, whereas it's around 0.5 in Figure 4H. And the full model explains only about half the variance (Figure 4H, Right).

In addition, you claim that the dynamics is autonomous. The test for this is described on lines 251-9:

"We first tested the starting assumption that the OFF response trajectories are consistent with linear, autonomous network dynamics driven only by the initial pattern of activity and network interactions. If that is the case, the OFF response dynamics are fully determined by the initial activity at stimulus offset (*t* = 0). This assumption implies that the geometric structure of the initial activity states r_0_^(s)^ fully predicts the geometric structure of the dynamics r^(s)^(*t*) during the OFF responses (see Materials and methods). In particular, two stimuli s_1_ and s_2_ corresponding to correlated initial states r_0_^(s1)^ and r_0_^(s2)^ will lead to correlated trajectories r^(s1)^(t) and r^(s2)^(t) in the model. Conversely, if two OFF-response trajectories r^(s1)^(t) and r^(s2)^(t) are orthogonal, the initial states r_0_^(s1)^ and r_0_^(s^_2_^)^ are necessarily orthogonal."

We're pretty sure that isn't correct: correlations are preserved only if the dynamics causes pure rotation at constant angular speed, something that doesn't happen in general, even for linear dynamics.

And finally, Figure 4E, right, shows that many u's and v's are not orthogonal -- judging from that figure, on the order of 10-20%. These terms were of the form

|u1| v2 v1 – |u2| v1 v2

where v1 and v2 are nearly orthogonal. You claim that terms like this can't lead to transient amplification. That's true if |u_1|=|u_2|. But if |u_1| and |u_2| are sufficiently different, it's not true. But maybe |u_1| \approx |u_2| in the data? If so, that should be clear. In any case, this seems like a detail -- it's certainly possible to do the analysis with these terms included.

The lack of orthogonality between the u's and v's doesn't seem so serious. But the lack of agreement between Figures 2D and 4H, combined with no strong evidence for autonomous dynamics, seems potentially problematic. Of course, no model ever fits perfectly. But it's hard to figure out exactly what the take-home message is. If there was strong evidence that the dynamics is autonomous, that would help -- at least we would know that the intrinsic dynamics is responsible for the transient amplification.

As a potentially important aside: the eigenvalue spectrum of Figure 4D shows one marginally stable, non-oscillatory mode. Based on past experience of one of the reviewers, it may be that the lack of agreement between Figures 2D and 4H is because of this mode. The reasoning is that if this slow mode is present by a small amount in all initial conditions, it will generate the fairly weak overlap in Figure 2D, but because it sticks, it might end up dominating the late off-responses for each stimulus after all the rest has decayed away. If that's the case, then this mode will be part of the principal subspace extracted for each stimulus. The measure of overlap based on "subspace angle" is very sensitive to this, which could explain the large overlaps in Figure 4H. Importantly, that marginally stable eigenvalue may be an artifact of insufficient regularization. There are a lot of "ifs" in the above reasoning, but if this mode can be eliminated, perhaps by different regularization, agreement may improve. We don't know if that's possible, but it's worth thinking about.

There is a lot of very nice analysis in the paper, and all reviewers agree that it should be published. However, it needs a clear take-home message. Exactly what that is depends, we believe, on how strongly the evidence is for autonomous dynamics. If that's very strong, then at the very least the take-home message is:

"Off cells show transient amplification that is generated by internal dynamics, not external input. That transient dynamics is mildly consistent with low rank connectivity in which dynamics can mainly be reduced to multiple 2-D systems. However, we can't rule out more complex behavior, with transient amplification fundamentally involving several dimensions."

Possibly you can come up with a stronger conclusion, but that's the best we could come up with.

If the evidence for autonomous dynamics is weak, that seems to be a fundamental problem, since the transient amplification may be caused solely by external input. But presumably that can never be ruled out -- a fact that you should probably mention. It's not a plausible scenario, so we don't think it will weaken the paper..

Our main comment, then, is to explore more carefully whether or not dynamics is autonomous, and/or be more careful about the conclusions. One possible way to test for autonomous dynamics is to plot the trajectories and look for intersections, or near intersections -- what Mark Churchland calls tangled trajectories, for which quantification is possible. Absence of near intersections would at least be consistent with autonomous dynamics. But there may be better methods. For instance, you might try LFADS: you can fit both an RNN and a linear dynamical model, and see which one fits better. That will, though, be a huge amount of work, and you should do it only if you want to.

In addition, we suggest exploring a richer single-neuron model. The current model has the form

r_i_^s(t) = r_{0i}^s L_i_(t).

If r_i_ is non-negative, this can be written

dr_i/dt = f_i_(t) r_i_,

with r_i_(t=0) a function of the stimulus. It would be more convincing to consider a model with a nonlinearity; say

dr_i_/dt = f_i_(t) g_i_(r_i_).

The nonlinearity g_i_ could be parameterized with, probably, only a handful of parameters. For instance, it could be written as a sum of basis functions, much the way L_i_ was parameterised. The single-cell model was rejected because it explained only about 20% of the variance, versus 50% for the recurrent network. It would be reassuring if adding a nonlinearity didn't bring this up to 50%. Again, this also is only a suggestion. However, if you stick with the current single cell model, you'll need to frame your comparison less generally than "recurrent versus single cell".

---

## [Author Response]

Essential revisions:We're very sorry for the long delay. That's in part because at least one of use found the paper very hard to read, and it took a long time to figure out what was going on. And as you'll see, it's not clear if we did that very well. We think the problem is that you're missing quantitative analysis; in many cases it seems that we're supposed to understand why a particular piece of analysis supported your model, but it simply wasn't clear to us why that was true. Explicit examples of this will be given below.

We thank the reviewers for a very constructive feedback which has allowed us to significantly improve the manuscript. Following the reviewers’ suggestions we have rewritten large parts of the Results and Methods. Here we first briefly list the main analyses we added. Further below, we explain the new structure of the manuscript, and how the new analyses address the reviewers’ comments.

List of new analyses:

– We theoretically derived quantitative predictions on the structure of the network connectivity for a general l ow-rank connectivity matrix and tested these predictions on the AC data (Figure 4E, Methods Section 2.3, 2.7);

– We derived quantitative predictions for the relationship between the structure of the network connectivity and the population vector at stimulus offset and tested these predictions on AC data (Figure 4F, Methods Section 2.3);

– We derived quantitative prediction for the value of the correlation between the population activity vectors at stimulus offset and at the time of maximum distance from baseline (Figure 4G, Methods Section 2.5);

– We added a method section illustrating the prediction for he correlation between the structure of the population vectors at stimulus offset and dynamical trajectories (Figure 4A) f or a general low-rank network model (Methods Section 2.4);

– We adopted a different approach when comparing the recurrent model with the single-cell model. In the new version of the manuscript, we fit directly the form of the temporal responses of the single units L_i_(t) to the data (we correspondingly updated Figure 5B-C; Methods Section 3.1). We then compare the predictions of the two models on AC data (Figure 5F).

– We added a new (unrequested) analysis comparing the predictions of the recurrent and single-cell model on the population dynamics across single-trials (Figure 6, Methods Section 3.2, 3.3). We show that the two models yield quantitatively different predictions for the structure of the trial-to-trial response variability, and that the AC data i s consistent with the recurrent model and not with the single-cell model. This adds substantial new evidence i n favor of the recurrent model.

– We added controls f or the principal component analysis of the OFF responses to individual stimuli and across stimuli (Figure S1, Methods Section 1.4), and for the values of subspace overlap (i.e. Figure 2D) and correlation between i nitial and peak states (i.e. Figure 4G). These controls are described i n detail i n Methods Sections 1.6 and shown i n Figure S7, S8.

A major issue for us has been that you begin with a discussion of special cases of Equation (2) (last two paragraphs of page 6) but then go on to test predictions of THE model without telling us which of these special assumptions you are actually making (if any) in the production of your results (Figure 4 specifically). We believe that the first paragraph of "Testing the network mechanisms on the data" should begin with a clear description of the actual model considered (including whether your transient channels are mutually orthogonal, and how initial conditions relate to those planes).

Following the reviewers’ suggestion, we have totally rewritten the subsection “Recurrent network mechanism” (and the corresponding Methods) and most of the subsections “Testing the network mechanism” and “Comparison with a single-cell model”.

To help the reviewers navigate the changes, here we provide a summary of the new organization of the manuscript.

First two subsections of Results: we use population analysis of calcium activity identify three main features of OFF responses in the auditory cortex:

F1. OFF responses correspond to transiently amplified trajectories;

F2. responses to individual stimuli lie in low-dimensional subspaces;

F3. responses to different stimuli lie in largely orthogonal subspaces.

Subsection “Recurrent network mechanism”: we propose a recurrent network model f or OFF responses based on only two hypotheses:

H1. dynamics are driven exclusively by recurrent interactions (Eq. 1), and the only effect of each stimulus s is to set the initial state r_0_^s^ that corresponds to activity at stimulus offset;

H2. the recurrent connectivity is of l ow-rank type, i.e. characterized by a set of connectivity vectors u^r^ and v_r_ (Eq. 1).

We then systematically derive a set of sufficient conditions on vectors u^r^, v^r^ and r_0_^s^ to reproduce the observed features F1-F3. These sufficient conditions provide corresponding predictions P1-P3 that can be tested on the data.

Subsection “Testing the network mechanism”: we systematically test on the data the hypotheses H1 and H2, as well as the predictions P1-P3. The structure of this subsection now mirrors the structure of “Recurrent network mechanism”.

Subsection “Comparison with a single-cell model”: we first compare predictions of the fitted models on trial-averaged data; we then compare new predictions f or trial-to-trial variability.

More specifically, to summarize our current understanding: under the assumption that the v's are orthonormal, the u's are orthogonal, and u_k_ dot v_l_ = 0 for all k \ne l (not strictly true, but close enough in high dimensions), the most general solution (a generalization of Eq. 34 to an arbitrary number of modes) isr(t) = [uperp + sum_k_ u_k_ (a_k_ + v_k_ dot u_k_ t)] exp(-t)where uperp is perpendicular to all the u_k_ and the a_k_ are arbitrary.This appears late in Methods. But it's important, because there seem to be hidden assumptions about uperp and the a_k_. Namely,– the a_k_ are all set to zero– uperp = u_k_ for only one k, with k dependent on the stimulus.Assuming this is correct, it's a critical part of the model. And it is, in fact, an extra assumption.

The new version of the text clarifies i n detail the questions raised by the reviewers. In short, we make no a priori assumptions on vectors u^r^, v^r^ and r_0_^s^ (the initial state) other than that the v's are orthonormal, the u's are orthogonal, but there is no loss of generality there (such u’s and v’s can be uniquely specified by an SVD of the connectivity matrix).

We then derive sufficient conditions on u+ and v^r^ (u_k_, v_k_ in the reviewers' notation) and r_0_ to reproduce F1-F3, and find i n particular:

– An upper bound on u^r^ dot v^s^ (always satisfied if u^r^ dot v_s_ = 0 for all r,s) to have transient amplification (F1)

– Conditions on r_0_ with respect to u^r^, v^r^, that correspond to conditions on uperp and a_k_ in the reviewers’ notation

We next fit the network model on the data (without specific constraints on u^r^, v^r^ and r_0_^s^), and check whether the predicted constraints on u^r^, v^r^ and r_0_^s^ hold.

The methods now start with equations equivalent to the general solution put forward by the reviewers (Eq. 33 i n a unit-rank example, and Eq. 35 for arbitrary rank).

Given that setup, we believe there were several predictions:a. Lines 216-228: There's a prediction about the correlational structure that we didn't fully understand. Equations are needed, in the main text, that make the predictions explicit.

The goal was simply to test H1, i.e. that the states of initial activity at stimulus offset determine the ensuing dynamics. We have rewritten that part of the text, and added a detailed section i n the Methods (Section 2.4, "Correlation between structure of initial states and structure of the dynamics").

b. Lines 229-238: Correlations between the beginning and the peak of the off response should small. Again, equations are needed; this should be quantified.

In the new organization of the text, this prediction now appears later. We have now derived a quantitative prediction for the upper bound of the correlation between initial state and peak response (Methods Section 2.5, "Correlation between initial and peak state"), and added a panel (Figure 4G) demonstrating this prediction is quantitatively satisfied.

c. Lines 239-253:"To further examine to which extent the population dynamics during OFF responses could be accounted for by a non-normal linear dynamical system, we next fitted our network model population OFF responses to all stimuli at once using reduced-rank regression (Figure S1; see Methods). Qualitatively, we found that the fit reproduced well the low-dimensional population trajectories (Figure 4C)."As far as we could tell, this just tells us that the fit is good; it doesn't tell us anything about whether the OFF responses could be accounted for by a non-normal linear dynamical system. Is that correct, or did we miss something? If that is correct, it seems like this paragraph is mainly a distractor.

The aim of this paragraph is twofold: (i) test H1, i.e. show an autonomous dynamical system is a good model of the data, using the approach introduced in Elsayed and Cunningham (2017) (ii) determine the effective connectivity matrix, and extract vectors u^r^ and v^r^, which are then used to test the predictions on the relationship between u^r^, v^r^ and r_0_^s^ (Figure 4 F and F). We have now rephrased that part (and reorganized the whole subsection).

d. Lines 254-262: The fitted connectivity, J, had eigenvalues less than 1 and the symmetrized version had eigenvalues greater than 1. That's consistent with the low rank, non-normal model. However, couldn't such a J could come from a high rank, non-normal model? If so, it's not strong evidence for your model. This should be clarified.

The reviewers are correct, the examination of the eigenvalues tests only the predictions that ensue from H1 (autonomous dynamical system), not H2 (low-rank structure). So this test provides only evidence for H1, not H2, and we have clarified that i n the new version of the text. H2 and the ensuing predictions are tested separately.

e. Lines 263-283: There were several results here, but it's not clear how much they supported the model. For example,"We computed the overlap between the subspaces spanned by the transient channels for each pair of stimuli s_1_ and s_2_ , analogous to the overlap between the vectors u^(s1)^ and u^(s2)^ in Eq. (2) (see Methods). We found that the overlap between the transient channels matched the overlap between the response dynamics and population states at the end of stimulus presentation (Figure 4F, right panel), consistent with the interpretation that individual OFF responses may be generated through transient coding channels of the form given by Eq. (2)."It's not clear what we should expect here. Is this strong or weak evidence for the model? You need to quantify what we should expect, and be clear whether this is only consistent with the model, or provides evidence for it.

Here we test the predictions that result from the requirement that the model reproduces F3 (orthogonal trajectories in response to different stimuli). The corresponding prediction has now been made explicit in the “Recurrent network mechanism” subsection. Specifically, we test Eqs. 3-4.

"While the fraction of variance explained when using the matrix J=Sum was necessarily lower than the one computed using J=Full , the model with connectivity J=Sum could still explain a consistent fraction of the total variance, and yielded values of R^2^ significantly higher than the ones obtained from the shuffles."Again, it's not clear what we should expect. Is this strong or weak evidence for the model?

We have now rephrased this part. If the prediction (Eqs.3-4) i s exact, the explained variance should be the same i f the model was f it on the responses to all stimuli at once, or separately on responses to different stimuli. We find that the explained variance i s slightly lower i n the second case, but still much larger than chance (as could be expected, since the responses are orthogonal to many, but not all stimuli). Hence Eq.3 i s not exact, but a reasonable approximation. The strength of the evidence f or the model i s therefore directly quantified by the value of R^2^ when fitting the model separately to different stimuli.

f. Lines 328-331: for the single cell model in Eq. 3, the correlations between the beginning and peak of the off response are high. This seems easily fixable with a very minor tweak. Suppose there are two kinds of neurons, some that peak early,r_i_(t) = a_i_(s) f_i_(t)and some that peak late,r_i_(t) = b_i_(s) g_i_(t)(for instance, f_i_(t) = Theta(t) e^{-t} and g_i_(t) = Theta(t) t*e^{-t}). Different stimuli (s) could activate different sets of neurons.Alternatively, the single neuron responses could be of the formr_i_(t) = a_i_(s) f_i_(t) + b_i_(s) g_i_(t)where the a's and b's are anti correlated (either a is large and b is small or vice versa) and f_i_ and g_i_ are as above.Are these model ruled out by the data? If not, that seems to weaken the paper (although maybe there are other things wrong with these models). In either case, models like these should be discussed.

In the original manuscript, the correlations between initial and peak activity were assessed by assuming a specific form of the single-neuron response, without constraining i t by the data. Rather than explore this quantity for various assumptions on the single-neuron responses, we decided to first fit the single-neuron responses to data, and then compute the resulting correlations between initial and peak activity. Figure 5F shows that these correlations are much higher than on actual activity trajectories. This clearly demonstrates that the single model does not capture the decorrelation present in the data. (note that the second form suggested by the reviewers, r_i_(t) = a_i_(s) f_i_(t) + b_i_(s) g_i_(t), is not consistent with our hypothesis that a single-cell response depends only on the activity of the cell at stimulus offset; it is instead equivalent to a unit-rank network).

We moreover added a new analysis and figure (Figure 6) comparing the predictions of the recurrent and single-cell mechanisms for trial-to-trial fluctuations. The results clearly indicate that the single-cell model is inconsistent with the data, and provide significant new evidence in favor of the network model.

g. Lines 332-353:"A second important difference between the two models is that in the single-cell model the temporal shape of the response of each neuron is the same across all stimuli, while for the recurrent model this is not necessarily the case."Given this opening sentence, why not just show that for the recurrent network model, the temporal shape of the response is not the same across all stimuli? Instead, there's a lot of analysis that we didn't fully understand. In particular, there's a rather sudden switch from specifying the form of L_i_(t) (Eq. 44) and fitting L_i_(t) (Eq. 48). What's missing is a quantitative prediction of what we should expect, followed by a demonstration that the data is consistent with that prediction.

We thank the reviewers for this important comment. In the new version of the manuscript, we have totally removed the part where we assumed a specific form of single-neuron responses. We now fit i t to the data right away.

Following the reviewers’ suggestion, we also show that in the network model the temporal shape of the response is not the same across stimuli (Figure 3E).

In addition, we have some specific suggestions.1. Lines 174-177,"We focus on two outstanding properties identified in the OFF responses to different stimuli. The first one is the strong amplification of OFF responses, as quantified by the transient increase of the distance from baseline ||r(t)||. The second important feature is the low-dimensionality of OFF response trajectories and their global structure across different stimuli."It seems that this is all you're going to focus on, but it's not. It would be nice to know, at some point early in the manuscript, exactly what you're going to do.

As mentioned above, we have now completely reorganized the manuscript to make clear we focus on three features identified i n the data (F1-F3 listed above). These features are now clearly listed in the abstract, introduction and provide the guiding line throughout the results.

2. What's the difference between Figures 2D and 4A (left panel).

Figure 2D displays the correlation matrix between the subspaces explored during the OFF responses to different stimuli. Figure 4A displays the correlation matrix of the initial states (=activity at stimulus offset, not included in the OFF responses). We have now clarified this i n the text and figure legends.

3. Lines 329-31,"The single-cell model predicts that these two states are only weakly decorrelated and essentially lie along a single dimension (Figure 5E and S7), which is inconsistent with experimental observations"Which experimental observations?

We have rephrased this sentence. As mentioned above, in the new version of the manuscript, we don't fix the shape of the single-cell responses, but instead fit them directly to the data. We compute the correlation between initial state and peak state from the fitted responses when fitting the single-cell model to a progressively increasing number of stimuli. We found that the value of the correlation predicted by the single-cell model deviated from the experimental value as more stimuli were included in the fit, in contrast to the recurrent model, which predicted a value of the correlation consistent with data for any number of fitted stimuli (Figure 5F).

4. Throughout the paper we found it unclear how initial conditions were chosen for the model, e.g. in Figure 3. From line 167, we are led to think that they are picked from the actual data, "we model the firing activity at the end of stimulus presentation as the initial condition of the network dynamics" -- since the model doesn't explicitly models the stimulus period as far as we understand (?), we infer this activity is taken from the data? But does the model even have the same number of neurons?

Figure 3 is only an illustration of the model (with rank=1), where the initial conditions r_0_ are picked to satisfy the conditions derived on the relationship between u_r_, v_r_ and r_0_ to reproduce F1-F3. Once we turn to testing the predictions on the data, the initial conditions are taken from the data: for each stimulus they correspond to the calcium activity recorded at stimulus offset, corresponding to 50ms before the termination of the stimulus. Prior to fitting the model, we applied dimensionality reduction (PCA) to the data to avoid overfitting, so that the dimensionality of the fitted model i s equal to the number of principal components. This has been specified i n Section 1.7 of the Methods "Linear dynamical system fit".

5. The purpose of Figure 2 is to convey that different stimuli evoke orthogonal OFF responses, and these OFF responses are low dimensional. By my calculation, for each conditional mean (over 20 trials), there are 400ms * 31.5 frame/s ~= 13 samples used to estimate the PCs and variance explained resulting in a maximum dimensionality (PCs to get to 80%) of 13. If this is the case, a more careful comparison to the cross-condition dimensionality is warranted, which accounts for sample size.

We thank the reviewers for pointing this out. We have now added two types of controls for the principal component analysis, which aim at controlling respectively for sample size and trial-to-trial response variability. To control f or sample size we performed PCA on the trial-averaged responses to individual stimuli using the raw data (13 time points). Next we performed PCA on the responses where we shuffled cell labels independently for each time point, thus preserving the firing rate statistics at each time point, but removing temporal correlations i n the response. We found that the dimensionality of the shuffled response was higher than the dimensionality of the real responses, indicating that the value of the dimensionality obtained for individual stimuli is not the direct consequence of sample size. These results are shown in Figure S1 A.

To account for trial-to-trial variability i n the response we performed a second control which uses cross-validated PCA. This approach (Stringer et al. 2018) provides an unbiased estimate of the signal variance by computing the covariance between a training and a test set of responses (e.g. two different trials) to an identical collection of stimuli. We show that the same number of dimensions obtained with ordinary PCA explains 80% of variance when the trial-to-trial variability of the responses to a fixed stimulus i s taken into account. We describe the details of this procedure in the Methods, Section 1.4 "Cross-validated PCA", and i n Figure S1 B-C.

Also, we would expect there to be fewer time points on the traces in Figure 2B. How were the trial-averaged responses interpolated?

Indeed, the time points were interpolated using a Gaussian filter with standard deviation equal to 32ms. This is specified in Section 1.1 "Neural recordings".

What steps were taken to insure that the same unit wasn't double counted during pooling across sessions. Mistakes in this process can contribute to inflated variance along primary PC's creating the illusion of low dimensionality.

Recordings in different sessions were performed at different depths, typically with a 50 micron difference (never less than 20 microns). Since the soma diameter is 15 microns, this ensured different cells were recorded in different sessions. We now specify this in the Methods (Section 1.1).

6. Statements of relative degree of subspace overlap should be verified with a hypothesis test. For data, this can be done by resampling neurons.

We thank the reviewers for pointing this out. We performed two types of statistical tests for assessing the significance of the orthogonality between responses to different stimuli, reflected in l ow values of the subspace overlap (Figure 2D). The two controls correspond to two different null hypotheses.

The first hypothesis is that low values of the subspace overlap may be due to the high dimensionality of the state-space in which they are embedded (e.g. the correlation between any pair of random vectors in high dimension is likely to be low). To test for this hypothesis we compared the subspace overlap computed f or each pair of stimulis 1 and s2 on the real trial-averaged responses with the subspace overlap computed on the responses where the stimulus labels s1 and s2 had been shuffled across trials, and then averaged over trials. This yielded pairs of responses that are embedded in the same dimensions of the real responses but are otherwise random. We found that f or most of the stimuli the subspace overlap was significantly lower for the real responses than f or the shuffled responses, indicating that l ow values of the subspace overlap were not a direct consequence of the high dimensionality of the state space.

The second null hypothesis is that l ow values of the subspace overlap might be an artifact of the trial-to-trial variability present in the calcium activity data. To test for this possibility, for each pair of stimuli we computed the values of the subspace overlap by computing the trial-averaged activity on only half of the trials (10 trials), subsampling the set of 10 trials multiple times for each pair of stimuli. We then compared these values with the values of the subspace overlap between the trial-averaged responses to the same stimulus, but averaged over two different sets of 10 trials each, over multiple permutations of the trials (if the trial-to-trial variability was l ow, these values should be close to 1 since we compute it on a single stimulus, but should decrease as a function of the variability). We found that for most of the stimuli the subspace overlap was significantly lower for the real responses than f or the controls, indicating that l ow values of the subspace overlap were not a direct consequence of the trial-to-trial response variability.

The same type of controls have been applied to test f or the decorrelation between the initial state and peak state (see Figure 4G). We describe the controls in detail in the Methods, Section 1.6 "Controls for subspace overlaps and initial state-peak correlations", while the results are reported in Figure S7-S8.

7. You should justify why LDA (naive covariance assumption) is more suitable for assessing linear separability in Figure 1C than an SVM. Absent justification, an SVM should be used to get the clearest assessment of linear separability.

We do not make a strong statement on linear separability in Figure 1C. LDA assuming naive covariance was used simply for convenience, and for consistency with previous studies that did not quantify the impact of noise correlations on decoding (Mazor and Laurent 2005, Saha et al. 2014).

[Editors' note: further revisions were suggested prior to acceptance, as described below.]

The manuscript has been improved but there are some remaining issues that need to be addressed before acceptance, as outlined below:The good news is that the paper is much improved. The writing is more clear, and the analysis of dimensions of amplified trial-to-trial variability is compelling. The less good news is that now that we understand it better, additional issues have come up. All can, we believe, be dealt with relatively easily.

We thank the reviewers for detailed and constructive feedback. We have significantly revised the manuscript to address the reviewers’ comments.

A major comment was that the paper lacked a clear take home message. Our main message is in fact very simple: A1 OFF responses across stimuli are better described by a network mechanism than by a single-cell mechanism. This is an important message, because the whole OFF-response literature is based on single-cell mechanisms (see Kopp-Scheinpflug, Sinclair and Linden, Trends in Neuroscience 2018), and a network model has been lacking.

The reviewers’ feedback however made us realize that this message was not communicated in a sufficiently clear manner. We have therefore thoroughly reorganized the manuscript to make this message more prominent.

The new organisation of the paper is the following:

1. The first part is unchanged: we start with model-free analyses of auditory cortex data.

2. We now first discuss the single-cell model, and show that it describes well responses to a single stimulus, but not responses across stimuli.

3. We then introduce the network model, and show that it accounts better for the responses across stimuli than the single-cell model.

4. We next analyze the structure of the network model, which leads to additional predictions we verify on the data. A subsection of this part has also been substantially revised following the reviewers’ feedback (see below).

As a consequence of this reorganization, the figures have also been thoroughly rearranged, and several panels have been added:

– Figure 1: unchanged.

– Figure 2: panel E (previously in former Figure 4) has been added.

– Figure 3 contains the fit of the single-cell and network models (previously in former Figure 5 and Figure 4 respectively). We added one more example neuron to panel E. We added panel G (distance from baseline for the fitted network model) and panel I (subspace overlaps for the fitted network model).

– Figure 4: the spectra of the fitted connectivity matrix J and of its symmetric part J_S_ have been re-computed using ridge regression (without low-rank constraints), and extending the temporal window of the OFF response from 350 to 600 ms to get a stable connectivity matrix. To evaluate uncertainty in the connectivity spectra, we now compute the spectra over multiple neuron subsamplings.

– Figure 5 (former Figure 3) has been updated, and now illustrates the dynamics of orthogonal rank-2 rotational channels.

– Figure 6 illustrates the structure of the connectivity fitted to individual stimuli, and the predictions ensuing from this structure. We added panel A showing the goodness of fit computed using reduced-rank regression normalized by its value computed when using ordinary least squares regression. We added panel C(left) (condition for amplification for orthogonal rotational channels) and removed former Figure 4F(right). Former Figure 5F has been moved to Figure 6E.

– Figure 7 contains the results shown in former Figure 4H. Panel B has been added (correlation between subspace overlap and overlap between transient channels).

– Figure 8 (former Figure 6): the panels have been re-organized: following the structure of the main text, the data panels are now shown before the model panels.

– Former Figure 4C has been removed and is now shown in Figure 3-Supplementary3.

– Figure 3-Supplementary1 has been updated to include up to 16 stimuli.

Below we reply to the details of the reviewers’ comments.

Let's start with the data you're are trying to explain. Basically, after the offset of an auditory stimulus, there is transient amplification, with different stimuli activating nearly orthogonal subspaces.To model this, low rank linear dynamics is used, of the formdr/dt = sum_{k=1}^R u_k_ v_k_ r – r,where r (=r1, r2,.…, rN) is the trial averaged activity. The network had a few additional properties,a. The rank, R, is much less than the number of neurons, N.b. The u's and v's are nearly orthogonal.c. The symmetric part of the connectivity matrix must have eigenvalues with real part >1.d. The model has no external input, so the response to a particular stimulus (after offset) depends only on the value of r immediately after stimulus offset, here denoted r0(s) where s is the stimulus.e. For any stimulus, s, r0(s) has appreciable overlap with a small number of v_k_ (on the order of 5).This model did an OK job explaining the data. In particular, Figure 4H, left (the model), is supposed to match Figure 2D (the data). But the match is not so good: the off-diagonal channel overlap in Figure 2D is near zero, whereas it's around 0.5 in Figure 4H.

We thank the reviewers for pointing out the inconsistency between the former Figure 4H and Figure 2D. We have traced it back to a mismatch between preprocessing parameters, and updated former Figure 4H (now Figure 7A) so that it now matches much better Figure 2D.

As indicated in the Methods (sections "Fitting the network model" and "Analysis of the transient channels"), before fitting the network model, we denoise the data using a PCA. In the original Figure 4H, the total number of dimensions (PC dimensionality) after denoising was set to D=50, which was too low compared to the dimensionality of responses to individual stimuli, and introduced correlations between the fitted low-rank components. Increasing this pre-processing parameter to D=100 and setting the rank parameter to R = 5 instead leads to a high match between Figure 2D and Figure 4H (new Figure 7C, coeff. of determination >0.7).

And the full model explains only about half the variance (Figure 4H, Right).

Our main argument for the network model is not based on the absolute value of explained variance, but on the fact that the explained variance does not increase, in contrast to the single-cell model.

The total variance explained by our model is consistent with previous reports from the literature that used similar methods. Fitting a linear dynamical system to motor cortical population activity yielded R^2^ values between 0.6 and 0.8 (see Elsayed et al. 2017, Lara et al. 2018). In our case, for D=100, R=5 and λ=1 for individual stimuli, we found that the cross-validated (10-fold) goodness of fit, as quantified by the coefficient of determination R^2^, was equal to 0.65. Note also that we used calcium recordings, which are significantly more noisy than electrophysiological recordings used in Elsayed et al. 2017, Lara et al. 2018.

In addition, you claim that the dynamics is autonomous. The test for this is described on lines 251-9:"We first tested the starting assumption that the OFF response trajectories are consistent with linear, autonomous network dynamics driven only by the initial pattern of activity and network interactions. If that is the case, the OFF response dynamics are fully determined by the initial activity at stimulus offset (t = 0). This assumption implies that the geometric structure of the initial activity states r_0_^(s)^ fully predicts the geometric structure of the dynamics r^(s)^(t) during the OFF responses (see Materials and methods). In particular, two stimuli s_1_ and s_2_ corresponding to correlated initial states r_0_^(s1)^ and r_0_^(s2)^ will lead to correlated trajectories r^(s1)^(t) and r^(s2)^(t) in the model. Conversely, if two OFF-response trajectories r^(s1)^(t) and r^(s2)^(t) are orthogonal, the initial states r_0_^(s1)^ and r_0_^(s^_2_^)^ are necessarily orthogonal."We're pretty sure that isn't correct: correlations are preserved only if the dynamics causes pure rotation at constant angular speed, something that doesn't happen in general, even for linear dynamics.

We apologize, this paragraph appears to have caused a lot of confusion, and we have now removed it. Our goal was in fact to simply show that the OFF responses depend on the initial state of the network at the end of stimulus presentation. This is consistent with a dynamical mechanism, as has been argued previously in the motor cortex (Churchland et al. 2012), but does not exclude a contribution from external inputs. Since this aspect is consistent with both single-cell and network models, we have now moved the previous Figure 4A to model-free analysis in Figure 2.

We did not intend to claim to demonstrate that the OFF-response dynamics in the auditory cortex are autonomous. In fact the word “autonomous” was used only twice in the Results (and only in the paragraph copied above), and we have now completely removed it, as technically the single-cell model we use is not an autonomous dynamical system.

A paragraph in the Discussion (“A major assumption of our model…”) clarifies that we cannot exclude contributions from external inputs during the OFF responses, and do not claim the dynamics are purely autonomous.

The reviewers are absolutely correct that linear dynamics do not in general preserve correlations – additional constraints are needed. The relationship between initial conditions and subsequent responses found in the data is non-trivial. We have now added a plot showing that the fitted network model accounts for this observation (new Figure 3I).

And finally, Figure 4E, right, shows that many u's and v's are not orthogonal -- judging from that figure, on the order of 10-20%. These terms were of the form|u1| v2 v1 – |u2| v1 v2where v1 and v2 are nearly orthogonal. You claim that terms like this can't lead to transient amplification. That's true if |u_1|=|u_2|. But if |u_1| and |u_2| are sufficiently different, it's not true. But maybe |u_1| \approx |u_2| in the data?If so, that should be clear. In any case, this seems like a detail -- it's certainly possible to do the analysis with these terms included.

The reviewers are absolutely correct. We have now thoroughly reorganized this part of the results and the corresponding methods. Specifically:

– We do not assume anymore orthogonality between the u’s and v’s.

– Instead, based on the properties of the fitted model, we now test the hypothesis that the network can be approximated as a sum of orthogonal rank-2 channels of the form |u1| v2 v1 – |u2| v1 v2

– We now describe under which conditions such channels lead to amplified dynamics (new Figure 5). In particular, as pointed out by the reviewers, a key condition is | |u1|-|u2| |>2.

– We then test those conditions in the data, in particular in the new panel Figure 5C.

The lack of orthogonality between the u's and v's doesn't seem so serious. But the lack of agreement between Figures 2D and 4H, combined with no strong evidence for autonomous dynamics, seems potentially problematic. Of course, no model ever fits perfectly. But it's hard to figure out exactly what the take-home message is. If there was strong evidence that the dynamics is autonomous, that would help -- at least we would know that the intrinsic dynamics is responsible for the transient amplification.

As pointed out at the beginning of the reply, our claim is not that the dynamics are autonomous. Our main message is that A1 OFF responses across stimuli are better described by a network mechanism than by a single-cell mechanism.

The manuscript has been thoroughly reorganized to better communicate this message.

As a potentially important aside: the eigenvalue spectrum of Figure 4D shows one marginally stable, non-oscillatory mode. Based on past experience of one of the reviewers, it may be that the lack of agreement between Figures 2D and 4H is because of this mode. The reasoning is that if this slow mode is present by a small amount in all initial conditions, it will generate the fairly weak overlap in Figure 2D, but because it sticks, it might end up dominating the late off-responses for each stimulus after all the rest has decayed away. If that's the case, then this mode will be part of the principal subspace extracted for each stimulus. The measure of overlap based on "subspace angle" is very sensitive to this, which could explain the large overlaps in Figure 4H. Importantly, that marginally stable eigenvalue may be an artifact of insufficient regularization. There are a lot of "ifs" in the above reasoning, but if this mode can be eliminated, perhaps by different regularization, agreement may improve. We don't know if that's possible, but it's worth thinking about.

We thank the reviewer suggesting this possibility. As explained above, the disagreement between former Figure 2D and former Figure 4H (new Figure 7A) was due to an incorrect setting in the preprocessing. Changing the PC dimensionality from 50 (as in the previous version of the manuscript) to 100 (new version) fixed the disagreement.

As noted by the reviewers, there was indeed a marginally stable eigenvalue in the full spectrum of J. We traced this back to the fact that, for the longest stimuli, the inter-trial interval is too short, and the OFF responses do not have enough time to decay to baseline. Excluding the longest stimuli, and extending the analyzed window eliminates the marginal eigenvalues (updated Figure 4).

There is a lot of very nice analysis in the paper, and all reviewers agree that it should be published. However, it needs a clear take-home message. Exactly what that is depends, we believe, on how strongly the evidence is for autonomous dynamics. If that's very strong, then at the very least the take-home message is:"Off cells show transient amplification that is generated by internal dynamics, not external input. That transient dynamics is mildly consistent with low rank connectivity in which dynamics can mainly be reduced to multiple 2-D systems. However, we can't rule out more complex behavior, with transient amplification fundamentally involving several dimensions."Possibly you can come up with a stronger conclusion, but that's the best we could come up with.If the evidence for autonomous dynamics is weak, that seems to be a fundamental problem, since the transient amplification may be caused solely by external input. But presumably that can never be ruled out -- a fact that you should probably mention. It's not a plausible scenario, so we don't think it will weaken the paper..

As pointed out at the beginning of the reply, our claim is not that the dynamics are autonomous. While we show evidence that initial conditions play a role in the subsequent dynamics, we cannot exclude that external inputs are present. This is explicitly acknowledged in the Discussion.

Our main message is instead that A1 OFF responses across stimuli are better described by a network mechanism than by a single-cell mechanism. As summarized at the top of the response, we have thoroughly reorganized the manuscript to better communicate this message.

Our main comment, then, is to explore more carefully whether or not dynamics is autonomous, and/or be more careful about the conclusions. One possible way to test for autonomous dynamics is to plot the trajectories and look for intersections, or near intersections -- what Mark Churchland calls tangled trajectories, for which quantification is possible. Absence of near intersections would at least be consistent with autonomous dynamics. But there may be better methods. For instance, you might try LFADS: you can fit both an RNN and a linear dynamical model, and see which one fits better. That will, though, be a huge amount of work, and you should do it only if you want to.

We thank the reviewers for this suggestion. We have however decided not to follow this route, because our main claim is not that the dynamics are purely autonomous, as explained above.

In addition, we suggest exploring a richer single-neuron model. The current model has the formr_i_^s^(t) = r_{0i}^s L_i_(t).If r_i_ is non-negative, this can be writtendr_i_/dt = f_i_(t) r_i_,with r_i_(t=0) a function of the stimulus. It would be more convincing to consider a model with a nonlinearity; saydr_i_/dt = f_i_(t) g_i_(r_i_).The nonlinearity g_i_ could be parameterized with, probably, only a handful of parameters. For instance, it could be written as a sum of basis functions, much the way L_i_ was parameterised. The single-cell model was rejected because it explained only about 20% of the variance, versus 50% for the recurrent network. It would be reassuring if adding a nonlinearity didn't bring this up to 50%. Again, this also is only a suggestion. However, if you stick with the current single cell model, you'll need to frame your comparison less generally than "recurrent versus single cell".

We thank the reviewers for this suggestion. We now explicitly frame our results as comparing *linear*  single-cell and network models (Abstract).

Note that the single-cell model captures about 75% of the variance when fitted to responses to a single stimulus. It is therefore clearly expressive enough to capture the dynamics seen in the data. What it does not capture is dynamics across stimuli, and it is not clear to us how a non-linearity could fix this, while linear network interactions do. The effects of a non-linearity would need to be included both in the single-cell and network models, and this goes beyond the scope of the present study.